# Quantifying Memory Utilization with Effective State-Size

## Abstract

As the space of causal sequence modeling architectures continues to grow, the need to develop a general framework for their analysis becomes increasingly important. With this aim, we draw insights from classical signal processing and control theory, to develop a quantitative measure of *memory utilization*: the internal mechanisms through which a model stores past information to produce future outputs. This metric, which we call **effective state-size** (ESS), is tailored to the fundamental class of systems with *input-invariant* and *input-varying linear operators*, encompassing a variety of computational units such as variants of attention, convolutions, and recurrences. Unlike prior work on memory utilization, which either relies on raw operator visualizations (e.g. attention maps), or simply the total *memory capacity* (i.e. cache size) of a model, our metrics provide highly interpretable and actionable measurements. In particular, we show how ESS can be leveraged to improve initialization strategies, inform novel regularizers and advance the performance-efficiency frontier through model distillation. Furthermore, we demonstrate that the effect of context delimiters (such as end-of-speech tokens) on ESS highlights cross-architectural differences in how large language models utilize their available memory to recall information. Overall, we find that ESS provides valuable insights into the dynamics that dictate memory utilization, enabling the design of more efficient and effective sequence models.

## 1 Introduction

In recent years, the success of autoregressive sequence modeling in the context of deep learning has largely been driven by advancements in highly parallelizable causal architectures, such as the transformer (Vaswani et al., 2023). However, despite their strong performance and hardware efficiency, understanding the inner workings of these neural networks remains a challenging task due to their non-linearity and the diversity of fundamental building blocks used. To this end, we leverage a new class of model abstractions, allowing for the development of a unified framework for the analysis of these computational units.

In particular, we note that the majority of sequence models of practical interest can formally be expressed as either linear systems ($y = Tu$) or systems with **input-varying linear operators** ($y = f_T(u)u$), the latter of which we abbreviate as LIV[1]. LIVs generalize the notion of adaptive, or *data-controlled* operators to a broader class than previously described in Massaroli et al. (2021); Poli et al. (2023). The input-varying linear operator framework decouples the input-varying *featurization* $u \mapsto T := f_T(u)$ and the linear mapping $y = Tu$ required to construct and apply the operator respectively. This decomposition enables a wide array of deep learning primitives to be uniformly formulated as linear systems, including models like convolutions [40; 47; 35; 56], linear state-transition recurrences [15; 18; 27; 61; 29; 26; 75; 49; 16], and attention variants [68; 34; 65].

Current approaches to analyzing the inner workings of LIVs often rely upon simple visualizations of the materialized operator $T$ (or the aggregation of $T$ across multiple layers and residuals) [45; 69; 1; 4; 74; 63]. However, these visualizations alone often fail to highlight critical properties that explain

---

[1]We abbreviate them as LIVs instead of IVLs to conform to the classical convention, where linear time-varying and time-invariant systems are abbreviated as LTVs and LTIs respectively, as well as recent works such as Zancato et al. (2024), which call them "linear input-varying systems".

Figure 1: An overview of the effective state-size metric and its various downstream applications.

how different models construct internal representations of the input data. Moreover, prior attempts in obtaining quantitative metrics, such as through spectral analysis of the operator $T$ (Min & Li, 2024; Bhojanapalli et al., 2020), are either limited to a specific model class or do not appropriately take into account important conflating factors like the causal masking of $T$ which significantly distorts the metric (Wu et al., 2024).

In this work, we focus our analysis on the working memory[2] of model architectures and examine two aspects of model memory in particular: ***memory capacity*** (i.e. cache/state size) and ***memory utilization***. Notably, memory capacity alone can be misleading, as models with similar capacities may learn to utilize their available memory to varying degrees. Therefore, we introduce the notion of memory utilization – a measure that provides deeper insight into the differences between architectures with comparable computational efficiency.

Our main technical contributions can be summarized as follows:

- We draw from classical signal processing and control theory, and propose ***effective state-size*** (ESS) – a metric computed by taking the rank of $T_{i:,:i-1}$ – as a proxy for *memory utilization* in LIVs (Section 3).
- We validate ESS beyond its theoretical interpretation by demonstrating its correlation with performance across a wide range of models and memory-intensive synthetic tasks, including associative recall and selective copying (Section 4).
- We construct initializers motivated by ESS which result in performance improvements relative to default initializers (Section 5.1).
- We explore the use of the ESS metric as a means of enhancing the performance-efficiency trade-off by demonstrating its ability to inform model distillation (Section 5.2).
- We extend the utility of ESS to language, demonstrating how it captures a previously uncharacterized property of LLMs: state modulation (Section 5.3). This phenomenon provides concrete intuition as to why introducing input dependencies into state transitions is crucial for in-context recall.

## 2 RELATED WORK

In this section, we briefly describe previous works that are closely related to the core findings of the paper. For an extended discussion of the related works, refer to Section B.

**Classifying sequence models.** A sequence model can be defined as a mapping from input sequences to output sequences that leverages a state to maintain information across time. Within the scope of deep learning, the early attempts at constructing sequence models fall under the class of non-linear RNNs (Rumelhart & McClelland, 1987; Hochreiter & Schmidhuber, 1997) which exhibit non-linear state space dynamics. However, in recent years, models with linear state transitions have grown in popularity due to inherent trainability (Pascanu et al., 2013) and efficiency (Martin & Cundy, 2018) limitations in the non-linear regime. These models fall under two categories: linear systems and systems with input-varying linear operators (LIVs). We note that linear systems can

---

[2]Here, we refer to "memory" in the sense commonly associated with the "state" of dynamical systems, as described by Willems (1989), as opposed to the notion of language models memorizing some fact encountered during training (Allen-Zhu & Li, 2024).

further be broken down into time-invariant (LTI) and time-varying (LTV) systems, depending on whether or not the parameters of the realized recurrence change as a function of time (i.e. sequence index). This taxonomy has also been used in prior work (Zancato et al., 2024) to motivate the construction of novel architectures. In modern deep learning, LTI systems (Lindquist & Picci, 1979; Ljung, 1999; Ho & Kalman, 1966) can be found in frameworks like S4 (Gu et al., 2022a). However, given the lack of expressivity afforded to LTIs, the current state-of-the-art models fall under the more expressive LIV class (Gu & Dao, 2024; Poli et al., 2023; Vaswani et al., 2023) which parameterizes the state-space dynamics as a function of the input. Since LIVs constitute the minimal superclass containing most sequence models of practical interest, the goal of this work is to develop a quantitative means of analyzing models under the LIV abstraction. To do so, we draw from minimal realization theory (DeWilde & van der Veen, 1998; Akaike, 1974), which has previously been applied within the scope of classical linear systems but has yet to be extended to modern LIV sequence models.

**Quantifying memory utilization in sequence models.** As discussed in Section 1, the current landscape of examining memory in sequence models primarily reduces to qualitative measures or limited quantitative ones. Here, we expand on the latter, noting that commonly used measures such as model size (Hoffmann et al., 2022) and cache/state size (which we define formally as theoretically realizable state-size in Section 3) are imperfect means of measuring memory in sequence models. Namely, while these metrics serve as reasonable proxies for the capacity of a model to learn, they fail to capture how much of that capacity is realized. As such, they ignore important aspects of the model pipeline such as data, initialization, and optimization. To capture these aspects of network training, we apply the notions of numerical and effective rank (Roy & Vetterli, 2007) to the minimal realization problem, which gives rise to the ESS metric. We elaborate on this in Section 3.

We highlight the concept of semiseparable rank (Vandebril et al., 2005; Xia et al., 2010), which has been recently explored in Dao & Gu (2024). Similar to ESS, semiseparable rank examines the rank of specific submatrices of $T$. However, as semiseparable rank is primarily motivated by the design of efficient algorithms, we find it most closely aligns with the concept of theoretically realizable state size (TSS) in the context of our work. In this study, we provide complete derivations of ESS, clearly distinguish it from TSS (and by extension semiseparable rank), and demonstrate its effectiveness as a proxy for memory utilization in practical settings for modern LIV sequence models.

## 3 THEORY

In this section, we begin by showing that most modern sequence models can effectively materialize a linear operator $T$. We then formally define ESS as a metric derived from $T$, providing theoretical insights grounded in minimal realization theory. Finally, we detail the practical computation of ESS, proposing two variants: tolerance-ESS and entropy-ESS.

Using the flattened notation, we let $T \in \mathbb{R}^{d\ell \times d\ell}$, $u, y \in \mathbb{R}^{d\ell}$ denote the operator, inputs, and outputs respectively, $\ell$ denote the sequence length and $d$ denote the channel dimension. Here, we index sequence indices with subscripts, i.e. $T_{ij} \in \mathbb{R}^{d \times d}$, $u_i \in \mathbb{R}^d$ and channels (and other non-temporal dimensions) with superscripts, i.e. $T^{\alpha\beta} \in \mathbb{R}^{\ell \times \ell}$, $u^\alpha \in \mathbb{R}^\ell$. For additional details on notation, refer to Section C.1.

**A unified representation of sequence models.** While typically nonlinear, most sequence models of interest can effectively materialize a linear operator $T$, where the equation $y = Tu$ faithfully expresses the computation performed by the model (see Section D.2 for further elaboration):

$$T_{ij} = C_i B_j \qquad \textit{linear attention,} \qquad T_{ij} = C_i A_{i-1} \cdots A_{j+1} B_j \qquad \textit{recurrence,}$$
$$T_{ij} = K_{i-j} \qquad \textit{convolution,} \qquad T_{ij} = C_i K_{i-j} B_j \qquad \textit{gated convolution,}$$
$$T_{ij} = \sigma(C_i B_j) \qquad \textit{attention.}$$

We make a distinction between linear systems (such as convolutions) and LIVs (such as attention and gated convolutions). In the former, the operator $T$ is *input-invariant* whereas the latter are constructed via *causal featurizers* that map past inputs into features, i.e. $f_B : u_{:i} \mapsto B_i$, which are then used to construct the elements of $T$ as outlined above. We begin our formulation by restricting the scope to linear systems.

**Operator-recurrence duality and the minimal recurrent realization of linear systems.** Consider a general linear recurrence formulated as follows:

$$s_{i+1} = A_i s_i + B_i u_i, \quad y_i = C_i s_i + D_i u_i, \tag{1}$$

where $(A_i \in \mathbb{R}^{n_{i+1} \times n_i}, B_i \in \mathbb{R}^{n_{i+1} \times d}, C_i \in \mathbb{R}^{d \times n_i}, D_i \in \mathbb{R}^{d \times d})_{i \in [\ell]}$; $s_i$ and $n_i$ are the state and state-size at sequence index $i$ respectively. Classic results (DeWilde & van der Veen, 1998, ch. 3) establish that for any given input-invariant operator $T$ (i.e. linear system), there exist infinite recurrent realizations in the form of Equation (1), motivating the search for the minimal one.

**Theorem 3.1.** *Given any causal input-invariant operator $T$, there exist infinite variations of linear recurrences in the form of Equation (1) that realize an equivalent input-output operator.*

A simple extension of the proof of this theorem (in Section C.2.4) demonstrates the following:

**Theorem 3.2.** *The rank of the operator submatrix ($H_i \equiv T_{i:,:i-1}$) determines the minimal state size required to represent the causal operation ($y = Tu$) as a recurrence.*

Refer to Section C.3 for the proof.

Note that since we use the flattened notation, where $T \in \mathbb{R}^{d\ell \times d\ell}$, $\mathrm{rank}(H_i)$ determines the minimal state-size required for processing all $d$ channels in a layer via a recurrence at the $i$th index in the sequence. We formally refer to this metric as per-sequence index ***effective state-size*** (ESS) and discuss its interpretation for both linear systems and LIVs below.

**Interpreting effective state-size.** As shown in Theorem 3.2, the ESS of an input-invariant linear system is given by its minimal state-size which is directly interpretable as a measure of model memory utilization. For LIVs, however, ESS is also a function of the input ($\mathrm{rank}(f_T(u)_{i:,:i-1})$) which means that the minimal realization process outlined in the proof of Theorem 3.1 is no longer guaranteed to obtain recurrences that preserve causality (i.e. the minimally realized features $A_i^*$ depend on future inputs $u_k$, $k > i$)[3]. *Nevertheless, ESS lower bounds the state-size $n_i$ (refer to Section C.2.2), meaning that for any LIV, an equivalent recurrence must necessarily materialize a state-size at least as large as its ESS.* Therefore, we claim that ESS serves as a proxy for memory utilization in not only linear systems, but in LIVs as well. We empirically validate the usefulness of this interpretation of ESS for LIVs in Sections 4 and 5.

**Memory capacity in LIVs.** The memory capacity of LIVs is given by the state-size $n_i$. We formally refer to it as ***theoretically realizable state-size*** (TSS), as it serves as a tight upper bound for ESS. For models without realization-agnostic minimal recurrent formulations, such as softmax attention, we resort to the trivial realization as shown in Equation C.2.5, in which TSS (for a single channel) is equal to the sequence index $i$. For models like SSMs and linear attention variants, TSS is equal to the state-size defined by their recurrent formulation. We refer readers to Section D.2 for detailed operator-specific derivations of TSS. Recall that ESS depends not only on the model's functional form, but also on the input data, optimization, and more generally anything that impacts the realization of $T$; in contrast, TSS is limited in that it is determined solely by the model's structure.

### 3.1 COMPUTING EFFECTIVE STATE-SIZE

In practice, computing ESS requires a few additional considerations due to the numerical errors and approximations involved in computing the rank of matrices. We propose two approaches – both of which rely on singular values ($\mathbf{\Sigma}_i$) from taking the singular value decomposition (SVD) of $H_i$ (which equals $f_T(u)_{i:,:i-1}$ for LIVs) – that provide complementary perspectives on the same metric.

**Tolerance-ESS.** Here, a tolerance value $\tau$ is manually selected to threshold the singular values of $H_i$, determining the tolerance-ESS metric as follows:

$$\text{tolerance-ESS}(H_i, \tau) := |\{\sigma_i^m : \sigma_i^m > \tau\}|, \quad \text{where } \sigma_i^m \in \mathbf{\Sigma}_i, \ U_i \mathrm{diag}(\mathbf{\Sigma}_i) V_i = \mathrm{SVD}(H_i). \tag{2}$$

According to the Eckart–Young–Mirsky theorem, the tolerance-ESS metric can be interpreted as the minimum state size necessary for an input-invariant recurrence to approximate the original operator, such that the spectral norm of the approximation error remains below the specified tolerance level ($||T_{ij} - T_{ij}^*||_2 \leq \tau$).

---

[3]An example of a causality preserving realization of LIVs is the trivial realization shown in Equation (C.2.5).

**Entropy-ESS.** One drawback of tolerance-ESS is its reliance on the somewhat arbitrary selection of a tolerance value. One can instead compute the effective rank (Roy & Vetterli, 2007), which involves exponentiating the normalized spectral entropy (perplexity) of $H_i$:

$$\text{entropy-ESS}(H_i) \coloneqq \exp\Big( -\sum_m p_i^m \log(p_i^m) \Big), \quad \text{where } p_i^m = \frac{\sigma_i^m}{\|\sigma_i\|_1}. \tag{3}$$

Entropy-ESS is particularly useful for summarizing metrics across the entire tolerance space, whereas tolerance-ESS offers a more precise and readily interpretable depiction of rank concerning approximation error. Unless a tolerance is specified, we use entropy-ESS throughout all our experiments. An additional discussion comparing tolerance and entropy ESS, along with our code for computing them, can be found in Section D.1.

**Computational complexity of ESS.** Since SVD scales cubically with the size of a square matrix, the time complexity of computing ESS for a single layer with $d$ channels, processing an input of sequence length $\ell$, is $O((d\ell)^3)$. Fortunately, most modern sequence models (outside of S5 (Smith et al., 2023)) process the $d$ channels independently (i.e. they are SISO). This means that each layer's $d\ell \times d\ell$ operator $T$ can be decomposed into $d$ independent $\ell \times \ell$ operators, which reduces the ESS computation to an SVD of $d$ independent operators ($\hat{T}^\alpha \coloneqq T^{\alpha\alpha}; \ \alpha \in [d]$). Consequently, a per-channel ESS can be computed and summed to obtain the equivalent per-sequence index ESS. This reduces the computational cost to $O(d\ell^3)$. Note that in models like attention, where channels share the same recurrence within a single head, the time complexity is further reduced by a factor proportional to the head dimension. Additionally, in recurrent models with bounded TSS ($n \ll \ell$), a truncated SVD can be employed to decrease the time complexity to $O(\ell^2 nd)$.

**Aggregation of ESS across model and data dimensions.** Initially, we formulated ESS as a per-sequence index metric, which is general in that it applies to all models that realize an operator $T$. Recall in the last section, we extended ESS along the channel dimension for SISO models. For LIV SISO models in particular, we can further extend ESS along the batch size dimension, since ESS is a function of the input. This means that for a multi-layer LIV SISO model with $m$ layers, $d$ channels, batch size $b$, and sequence length $\ell$ (which describes the entire set of models we analyze in this work), $\text{ESS} \in \mathbb{R}^{m \times d \times b \times \ell}$. Since ESS is a multidimensional tensor, there are various ways to aggregate it across the model and data dimensions. We define two particular modes of aggregation termed *average* ESS and *total* ESS as follows:

$$\text{average ESS} = \frac{1}{mdb\ell} \sum_{\alpha \in [d]} \sum_{\beta \in [m]} \sum_{\gamma \in [b]} \sum_{i \in [\ell]} \text{ESS}_i^{\alpha\beta\gamma} \quad \text{total ESS} = \text{average ESS} * d$$

Average ESS marginalizes across all of the ESS tensor dimensions by taking a mean, creating a per-channel measure of ESS. Since average ESS is the metric used throughout most of our experiments, we will refer to it as ESS unless otherwise specified. For models like softmax attention, where average TSS (which is computed analogously to average ESS) depends only on the sequence index $i$ and thus remains constant as a function of model width (i.e. channel dimension), we instead capture a model-dependent statistic, by first summing the ESS across channels and then averaging over the remaining dimensions. This approach allows both ESS (and TSS) to vary as a function of model width, a metric that we refer to as total ESS. For more details, refer to Sections D.1 and D.2.

## 4 EMPIRICAL VALIDATION OF EFFECTIVE STATE-SIZE

To demonstrate the practical utility of ESS beyond its theoretical interpretation discussed in Section 3, we next turn to an empirical analysis. In this section, we examine ESS across a wide range of tasks and models in order to understand how it varies across different regimes, with particular focus placed on its relationship with model performance on memory-intensive tasks.

**Task space.** To explore ESS in an extensive, yet controlled, manner, we iterate on a set of synthetic tasks proposed by Poli et al. (2024) which have been shown to effectively approximate model performance on large-scale language tasks. Specifically, we train models on the multi-query associative recall (MQAR), selective copying, and compression tasks, each of which probes the ability

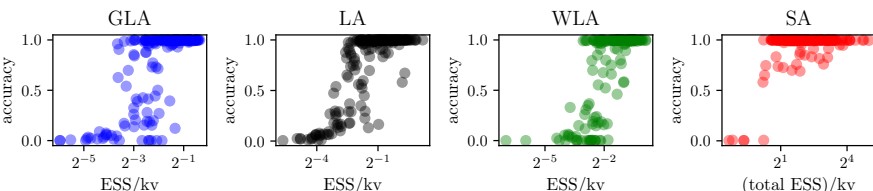

Figure 2: Scatter plots of accuracy vs ESS/kv across featurizers. Within each featurizer plot, all task-model configurations from the sweep corresponding to each featurizer are shown.

of models to effectively utilize their working memory. We note that here, we restrict the presentation of our results to MQAR and refer the reader to Section E.1 for the results on selective copying and compression, which showcase analogous trends.

**Model space.** We explore four models within the scope of this analysis: gated linear attention (GLA), weighted linear attention (WLA), linear attention (LA) and softmax attention (SA). We choose this set of frameworks since, together, they capture a large portion of the space of modern sequence models. The key distinctions between these models are as follows (more details can be found in Section D.2):

- GLA layer: This layer implements the gated linear attention formulation described in Yang et al. (2024a), where the recurrent feature $A$ (gating term) is input-varying, placing it in the same class as models like Liquid-S4 (Hasani et al., 2022) and Mamba (Gu & Dao, 2024; Dao & Gu, 2024).

- WLA layer: This layer is nearly identical to GLA, but with an input-invariant $A$ matrix. This lies in the same class as Hyena-S4D (Poli et al., 2023), RetNets (Sun et al., 2023), and gated-convolutions in general.

- LA layer: This layer is based on Katharopoulos et al. (2020); $A$ is not trainable and is instead fixed as the identity matrix.

- SA layer: This is the canonical attention layer which is similar to linear attention, but with the addition of a softmax non-linearity applied to the attention matrix (Vaswani et al., 2023), enabling unbounded TSS.

**Experimental setup.** In our analysis, we exhaustively sweep across the tasks and models (which are comprised of two sequence-mixing and two channel-mixing layers) detailed above.

Within each task, we also sweep across varying task difficulties. In the case of MQAR, we do so by modulating the number of key-value (kv) pairs the models are tasked to match, as well as the total sequence length of the prompt. Within each model, we sweep across varying TSS. For each task-model configuration, we compute the ESS and accuracy on a validation set every 10 epochs. We will refer to the entire space of tasks and models across which we sweep as the task-model space. Finally, we split our profiling of ESS into two sections: cross task-model analysis (Section 4.1) and within task-model analysis (Section 4.2). For more details on the setup, refer to Section D.3.

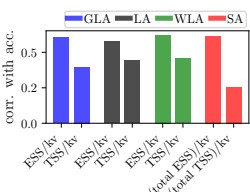

Figure 3: ESS/kv vs TSS/kv as a proxy for model performance as measured by correlation.

### 4.1 CROSS TASK-MODEL ANALYSIS

Our first goal is to understand how ESS empirically captures memory utilization by studying its correlation with post-training MQAR performance across the entire task-model space. To appropriately analyze ESS across tasks, we normalize it by the memory demands of MQAR, constructing an adjusted form of ESS given by ESS/kv.

**Finding 1:** Measured over the entire task-model space, ESS/kv exhibits a significantly higher correlation with accuracy than TSS/kv (Figures 2, 3, 9a, 9b).

Note that the strong correlation between ESS/kv and accuracy highlights the efficacy of ESS as a proxy for memory utilization. Furthermore, this finding underscores a significant gap in the explanatory power between ESS and TSS, emphasizing the importance of analyzing models beyond just their memory capacity.

## 4.2 WITHIN TASK-MODEL ANALYSIS

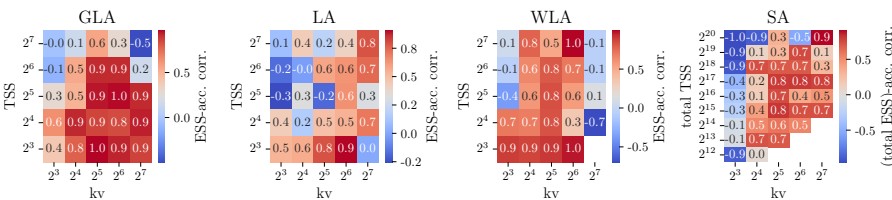

Figure 4: Correlation between ESS and accuracy over the course of model training bucketed by TSS and kv.

Next, to further establish ESS as a proxy for memory utilization, we study how ESS evolves as a function of MQAR performance in a regime where TSS is kept fixed and, therefore, does not correlate with accuracy. We do this by analyzing ESS-accuracy correlation on a per-model, per-task basis over the course of training, uncovering several insights that serve as the basis for our subsequent analysis.

> **Finding 2:** For less memory-intensive tasks trained using models with high TSS, we observe a lower correlation between ESS and performance compared to more memory-intensive tasks trained using a lower TSS (Figure 4).

This is in line with the interpretation of ESS as a measure of memory utilization. For easier tasks that are learned by a model with high memory capacity, the model is not incentivized to increase its memory utilization beyond where it resides at initialization. In contrast, for difficult tasks that operate in a memory-constrained regime, the model is forced to increase its memory utilization in order to learn, resulting in strong positive correlations between accuracy and ESS over training. [4] Digging a bit deeper, we find that this form of ESS analysis reveals two failure modes of model learning in recurrent frameworks (which recall have bounded TSS): **state saturation** and **state collapse**.

State saturation refers to the scenario in which a model has insufficient TSS to fully learn a task, resulting in its ESS converging near its TSS. This is reflected in its ESS/TSS (which we refer to as state utilization) residing near 1. We observe this in Figure 5 where we note that models with a TSS of 8 perform worse as the task difficulty scales due to a saturated state. State collapse, on the other hand, refers to the scenario in which a model has sufficient TSS to learn (or partially learn) a task, but its ESS fails to increase during training, resulting in a heavily underutilized state. With respect to state collapse, we observe the following:

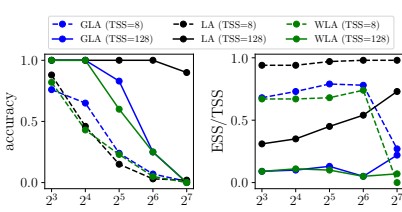

Figure 5: Accuracy and state utilization as a function of kv for low and high TSS models.

> **Finding 3:** For GLA and WLA, state collapse occurs in the high kv bucket of task-model space (i.e. kv = $2^7$) whereas for LA it does not (Figure 5). More generally, we find that LA has higher state utilization than GLA and WLA.

While state saturation can only be solved by increasing TSS, state collapse can in principle be solved by increasing ESS. Unlike TSS, which is a fixed hyperparameter of the model, one can modulate ESS by changing various aspects of the model pipeline. Furthermore, even outside of the state collapse regime, given the positive correlation between ESS and performance across the task-model space, increasing ESS is a generally viable approach to improving model performance without sacrificing efficiency. We explore this idea in the results to follow.

---

[4]In Figure 4, the empty spot in the WLA grid corresponds to a NaN from the entropy-ESS computation. The empty spots in the SA grid correspond to MQAR task constraints discussed in Section D.3.

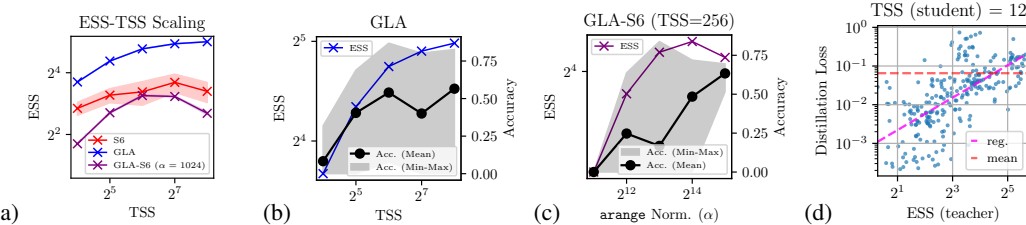

Figure 6: (a) ESS-TSS scaling in the S6, GLA and GLA-S6 featurizers. (b) ESS and accuracy on MQAR as a function of TSS in GLA. (c) ESS and accuracy on MQAR as a function of normalization factor for initialization in GLA-S6. (d) Distillation loss vs ESS of the teacher model.

## 5 APPLICATIONS OF EFFECTIVE STATE-SIZE

In Section 4, we showed that changes in ESS are correlated with changes in performance, both across models and during model training, indicating its importance beyond just interpretability. In this section, we aim to push this insight further by understanding how we can leverage ESS to both improve upon and improve our understanding of the existing performance-efficiency frontier in sequence models. We partition our results based on the stage of model training at which we apply ESS analysis: initialization-phase (Section 5.1), post-training (Section 5.2), and mid-training (Section E.3). Finally, we move beyond synthetics and extend ESS to language in Section 5.3.

### 5.1 INITIALIZATION-PHASE ANALYSIS

Initialization in weight space plays a crucial role in machine learning, significantly impacting model convergence and training stability (Glorot & Bengio, 2010). We extend this concept to the initialization of recurrent models in state space, leaning on the intuition from Figure 2 that suggests higher ESS can enhance performance. Namely, we illustrate how ESS at initialization can be used to inform featurizer selection – the selection of the function that maps the input to the operator $T = f(u)$ or equivalently the recurrent features $(A_i(u_{:i}), B_i(u_{:i}), C_i(u_{:i}), D_i(u_{:i}))_{i \in [\ell]}$ – and initialization schemes. In doing so, we uncover design flaws of a prominent model, S6 (Mamba) (Gu & Dao, 2024).

**ESS-informed featurizer selection.** To study the relationship between memory capacity and memory utilization in S6, we remove the short convolutional layer in the Mamba block and stack two of these modified blocks between SwiGLUs (Shazeer, 2020). Under the default MQAR task settings outlined in Poli et al. (2024) (see Tables 2, 3, and 4 for details), we observe that S6 is entirely unable to learn MQAR (accuracy $\approx 0$) across multiple scales of TSS (16 - 256) as shown in Figure 25. This aligns with results from Yang et al. (2024b), which independently demonstrate the poor performance of S6 without the additional short convolutional layer on a different in-context recall task. To investigate the cause, we look into how S6 is preconditioned to utilize its memory by computing its ESS when processing a Gaussian noise input, prior to training.

**Finding 4:** Figure 6a demonstrates that the ESS of S6 layers at initialization scales poorly with respect to TSS, notably failing to increase monotonically. In contrast, GLA layers (Yang et al., 2024a), configured with hyperparameters to match the TSS, model width, number of layers, and hidden-state normalization of the S6 model (see Section D.2 and Table 2), exhibit greater and monotonically increasing ESS-TSS scaling at initialization (Figure 6a). Despite the architectural similarities between the S6 and GLA layers, Figure 6b demonstrates that, unlike S6, GLA achieves accuracy improvements that correlate with increases in both TSS and ESS.

Based on these findings, we conjecture that the poor ESS-TSS scaling of S6 prevents the model from effectively utilizing all of its states, irrespective of increases in memory capacity.

**ESS-informed initialization scheme.** To further investigate the differences between the aforementioned S6 model and GLA model, we construct a composite model termed GLA-S6. This model adopts the feature-sharing structure of GLA (dividing dimensions into heads and sharing

computations within a head), but applies the S6 featurization to the $A$ matrix as follows:

$$\text{GLA (original):} \qquad A = \text{diag}(\text{sigmoid}(Wu)^{1/\beta}) \tag{4}$$

$$\text{GLA-S6:} \qquad A = \text{diag}(\exp(-([1/\alpha \quad 2/\alpha \quad \dots \quad n/\alpha]^T \odot \text{softplus}(Wu)))). \tag{5}$$

Like S6, GLA-S6 fails to learn MQAR across the same range of TSS (see Figure 25) and exhibits poor initialization-ESS scaling as shown in Figure 6a. Upon further inspection, we identify the cause of poor ESS scaling: with each new state introduced, the `arange` term ($[1 \quad 2 \quad \dots \quad n]$) exponentially pushes new entries of $A$ towards zero, negating the effects of additional states despite the increase in TSS. Therefore, to ameliorate the poor ESS scaling, we propose a simple solution: increase the normalization factor.

**Finding 5:** By scaling the normalization factor ($\alpha$), Figure 6c shows that GLA-S6 achieves improvements in MQAR accuracy post-training, reflecting the impact of increasing its initialization-ESS, despite the models having identical memory capacities.

These experiments demonstrate the efficacy of analyzing ESS at initialization, as they reveal how different models are preconditioned to utilize their working memory. This analysis helps identify potentially weak featurization and initialization schemes, enabling us to pinpoint shortcomings in the S6 featurizer and implement a straightforward fix.

## 5.2 POST-TRAINING ANALYSIS

Recall from Section 4.1 that we observed a strong correlation between ESS and post-training performance. Building on this insight, a natural question arises: can ESS be used for more than just performance analysis in the post-training setting? In this section, we answer this question by exploring an additional post-training application: model-order reduction.

**ESS-informed model-order reduction.** Model-order reduction refers to the process of improving model efficiency by reducing state-size while retaining performance. Previous works, such as Massaroli et al. (2023), have explored the distillation of linear time-invariant (LTI) operators ($T_{ij} = T_{i+k,j+k}$) into linear recurrences with small state-sizes using backpropagation. Other techniques for model-order reduction such as modal truncation and balanced truncation (Beliczynski et al., 1992; Gawronski & Juang, 1990) are also applicable to LTIs. In this study, however, we are concerned with improving the efficiency of general LIVs. Since ESS serves as a lower bound for the minimally realizable TSS (Section 3), we postulate that ESS can be used as a heuristic for conducting model-order reduction.

To test this, we distill multiple GLA models (with TSS = 256) across various task regimes to understand how the ESS of the original model (i.e. the teacher model) influences its ability to be distilled into a smaller student model. We apply the technique outlined in Bick et al. (2024), where the process can be divided into two steps. 1) matching the operators ($\min(||T_{(s)} - T_{(t)}||_F^2 / ||T_{(t)}||_F^2)$) and 2) matching the output activations ($\min(||y_{(s)} - y_{(t)}||_2^2 / ||y_{(t)}||_2^2)$). More details can be found in Section D.6. Figure 6d (and more comprehensively Figure 30) shows the relationship between the ESS of the teacher model and the final activation loss during distillation.

**Finding 6:** Higher teacher ESS correlates with greater activation loss. The downstream performance after single-layer distillation depends on both the teacher model's average ESS and student model's TSS, with higher teacher ESS and lower student TSS resulting in greater performance loss (Figure 29).

These findings position ESS as a useful heuristic for predicting model compressibility, enabling efficient estimation of the potential for state-size reduction without extensive experimentation.

## 5.3 STATE MODULATION OF LARGE LANGUAGE MODELS

In contrast to synthetic tasks like MQAR, selective copying, and compression, we find that strong recall performance on language depends not only on a model having sufficient ESS, but also on its ability to dynamically modulate its ESS in response to inputs. We demonstrate that this explains why linear attention, though effective on synthetic experiments (Section 4), is widely known to perform poorly on more complex language tasks (Katharopoulos et al., 2020; Arora et al., 2024).

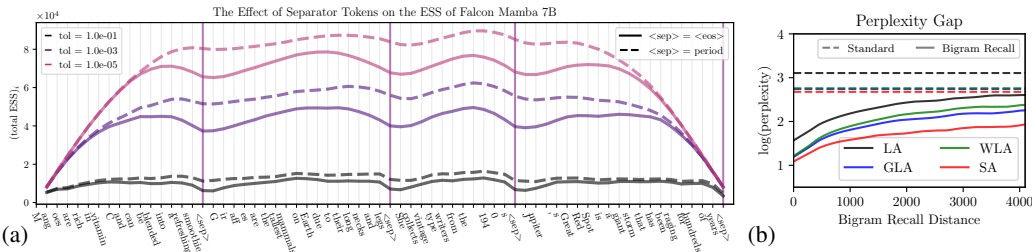

Figure 7: (a) The effect of separator tokens over Falcon Mamba 7B. See Section E.5 for plots of other open-weight models. (b) Comparison of standard perplexity and bigram recall perplexity (Arora et al., 2023).

We begin by evaluating the $(\text{total ESS})_i$ of open-weight pre-trained models. Borrowing notation from Section 3.1, $(\text{total ESS})_i$ is defined as follows:

$$(\text{total ESS})_i = \frac{1}{mb} \sum_{\alpha \in [d]} \sum_{\beta \in [m]} \sum_{\gamma \in [b]} \text{ESS}^{\alpha\beta\gamma}$$

Since we do not marginalize along the sequence dimension, we will examine this measure qualitatively by observing how it changes as a function of the sequence index. Our analysis shown in Figure 7a (and more broadly in Section E.5) reveals an intriguing phenomenon: ESS undergoes a noticeable dip whenever an end-of-speech (EOS) token is encountered (refer to Section D.8 for experimental details). This behavior aligns with our intuition regarding the role of EOS tokens and provides a quantitative measure of how effectively a model can 'reset' or 'forget' past contexts when transitioning between distinct segments of text. To investigate these effects in a more controlled environment, we trained four 1B parameter models (LA, WLA, GLA, and SA as described in Section 4) under identical conditions (see Table 9).

> **Finding 7:** We observe a clear hierarchy in the degree of state modulation, which can be summarized as follows: SA > GLA > WLA > LA (Figure 40).

SA exhibits the most pronounced state modulation, beginning at a tolerance level of $1\text{e}-2$, while also realizing the largest ESS. GLA follows, with modulation emerging at a tolerance of $1\text{e}-1$. WLA shows minimal modulation, only detectable at a tolerance of 1.0, while LA displays no discernible state modulation in response to separator tokens, demonstrating a clear lack of ability to modulate ESS. The importance of state modulation becomes apparent when examining model performance. Figure 7b illustrates that although standard perplexities (computed over a subset of the FineWeb dataset (Penedo et al., 2024)) are similar across SA, WLA, and GLA, significant differences emerge when considering the bigram recall perplexity metric introduced by Arora et al. (2023).

> **Finding 8:** The ability of a model to recall information, as measured by bigram recall perplexity across a pre-training dataset (rather than within a narrow task space), reveals a performance hierarchy that closely mirrors the observed state modulation capabilities.

This finding suggests that state modulation serves as a key mechanism enabling models to effectively manage complex context dependencies, directly impacting their performance on recall-heavy tasks.

## 6 CONCLUSION

In this work, we propose effective state-size (ESS), a measure of memory utilization in sequence models derived using dynamical systems theory. We motivate this metric as a valuable tool for analyzing memory utilization in LIVs by demonstrating its strong correlation with performance across a wide range of synthetic tasks. In doing so, we find that ESS offers a versatile framework for understanding both the performance and efficiency of causal sequence models. Leveraging these insights, we are able to construct novel, ESS-informed initializers, regularizers, and distillation strategies that improve beyond the existing performance-efficiency trade-offs found in recurrent models. Finally, we extend the ESS framework to language tasks, introducing the idea of state modulation – a concept that proves crucial for performance on bigram recall tasks. Overall, this work establishes ESS as a foundational tool for understanding and improving sequence model performance, opening new avenues for optimizing memory utilization and, more generally, model efficiency.

## REPRODUCIBILITY STATEMENT

To ensure reproducibility, we utilized open-source models and tasks, adhering to default task configurations unless otherwise specified. All crucial configurations are detailed in either the main text or the appendix. Additionally, our code for computing both the tolerance-ESS and entropy-ESS is provided in the appendix (Section D.1.3).

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

# Supplementary Material

CONTENTS

## A  MID-TRAINING ANALYSIS

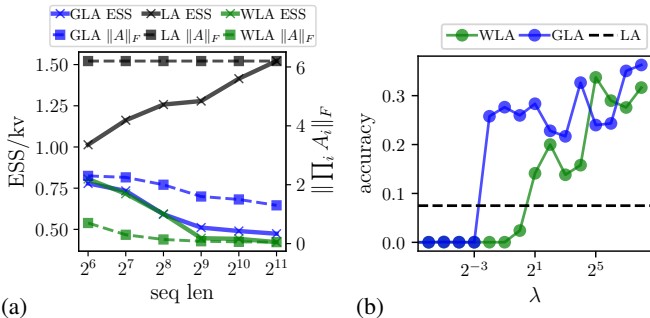

Figure 8: (a) ESS/kv and $\|\prod_i A_i\|_F$ as a function of sequence length. (b) Accuracy of models as a function of ESS-based regularizer strength.

To motivate the idea of increasing ESS mid-training, we revisit the concept of state collapse – a phenomenon that arises due to trainability issues in recurrent models (Figure 27), as discussed in Section 4.2. Recall that state collapse describes a failure mode of learning in GLA and WLA which, unlike LA, have learnable $A_i$ matrices (where $i$ denotes the index along the sequence dimension). To see why this contributes to state collapse, we note that the values of the operator submatrices $H_i$ are disproportionately influenced by $A_i$, due to the presence of terms in the form of $A_{i-1} \ldots A_1$ for each $i$. Hence, the closer $A_i$ lies to the 0-matrix, the faster these terms decay, reducing the numerical rank of $H_i$. We demonstrate this empirically in Figure 8a, which shows that for both GLA and WLA, ESS/kv and $\|\prod_i A_i\|_F$ decrease as a function of sequence length. In contrast, for LA, whose $A$ matrix is given by the identity, ESS/kv remains large as the sequence length grows.

Given this insight, one approach to addressing state collapse in GLA and WLA is pushing the $A$ matrices towards the identity by adding the following term to the loss function: $\lambda \|A - I\|_F$, where $\lambda$ denotes the strength of the regularizer and $I$ denotes the identity. In doing so, we are effectively decaying the model towards LA, increasing its ESS and giving us the following:

> **Finding 9:** GLA and WLA trained using the ESS-based regularization scheme described above outperform LA. When trained without it, they perform worse than LA (Figure 8b).

For more commentary on this result, please refer to Section E.3.

## B  EXTENDED RELATED WORK

**Causal sequence models.**  From classical linear recurrences to modern sequence models like Transformers, a vast array of causal model architectures have emerged (Vaswani et al., 2023; Tsai et al., 2019; Katharopoulos et al., 2020; Poli et al., 2023; Yang et al., 2024a; Gu & Dao, 2024; Dao & Gu, 2024; Sun et al., 2023). In recent years, the ability to process sequences in parallel has become increasingly critical, largely due to advancements in hardware accelerators such as GPUs. This need for parallelism likely explains the growing popularity of models like attention, Mamba, and S4.

We observe that all of these models, which support parallelization across the sequence dimension, can be formulated using a linear system representation ($y = Tu$) as detailed in the introductory and theoretical sections (Sections 1 and 3). For this work, we categorize these models into two types: linear systems and systems with input-varying linear operators. The key distinction between these two frameworks is that in the former, the operator $T$ in input-invariant models is composed of fixed system parameters, whereas in the latter, the parameters are dynamically generated from the input. Linear systems encompass both linear time-varying (LTV) and linear time-invariant (LTI) systems. Although LTV systems have been relatively unexplored in deep learning, several LTI models have been studied (Gu et al., 2022a;b; Smith et al., 2023; Orvieto et al., 2023; Parnichkun et al., 2024). Convolutional models use kernels $h$ to construct Hankel matrices $H$, whose rank corresponds to the minimal state-size of the model (DeWilde & van der Veen, 1998). Massaroli et al. (2023) explored methods to reduce the order of models by leveraging the Hankel matrix. Notably, the submatrix $H_i$ (defined in Theory 3.2) exhibits a Hankel structure in LTI models and provides per-sequence-index

information. In this work, however, we do not explore Hankel matrices further, as they are not easily generalizable to LTV systems.

In contrast, systems with input-varying linear operators, which we abbreviate as LIVs, are characterized by an operator $T$ that is dynamically constructed through a featurizer and is defined by $T = f(u)$. Examples of such models include softmax attention (Vaswani et al., 2023), linear attention (Katharopoulos et al., 2020), Liquid-S4 (Hasani et al., 2022), Mamba (Gu & Dao, 2024; Dao & Gu, 2024), and gated linear attention (Yang et al., 2024a). Although these models may appear nonlinear, they can still be represented like a linear system, enabling the application of linear analysis techniques. This forms the basis for the effective state-size metric.

**Interpretability.** Analysis tools for sequence models can be categorized into two types: extrinsic and intrinsic. Extrinsic tools focus solely on the input and output, treating the model's internal processes as black boxes. This approach is highly generalizable, as it can be applied to any model, including those with non-linear recurrences. A notable example by Shen (2019) uses statistical measures such as mutual information to compute metrics that capture model "expressivity". While these methods are versatile and applicable to various datasets, their generality makes them less effective at capturing the inner workings of causal sequence models, which is the primary focus of this work.

Intrinsic tools, conversely, directly visualize the model's internal mechanisms. A recently popular framework known as mechanistic interpretability provides one such example (Power et al., 2022). Mechanistic interpretability involves dissecting complex models to understand how specific components contribute to the model's overall behavior (Cammarata et al., 2020). Unlike our work, mechanistic interpretability does not target the operator view of the model but instead emphasizes the functional roles and interactions of individual model components.

For our purposes, we are primarily concerned with the visualization and analysis of classical and modern causal sequence models through the unifying lens of linear operators. Most analyses of these operators rely on visualization techniques (Olsson et al., 2022a; Vig, 2019; Abnar & Zuidema, 2020; Ali et al., 2024; Xiao et al., 2024; Sun et al., 2024) to gain insights into the model's internal processes. Visualizing the operator $T$ is advantageous, as it reveals important features like the formation of induction heads, strong activations, diagonal and block-diagonal patterns, and Toeplitz structures. However, raw visualizations are largely qualitative and oftentimes do not provide the quantitative metrics necessary for effectively evaluating a model's internal mechanisms – a gap we aim to address in this work.

Other, more quantitative, intrinsic methods perform some form of spectral analysis on the full operator (Dong et al., 2023; Min & Li, 2024; Tumma et al., 2023; Bhojanapalli et al., 2020). A limitation of these approaches is that they often disregard the causal masking of $T$, which significantly impacts the model's rank and singular values (Wu et al., 2024). As a result, the rank of the causal operator $T$ alone lacks a clear interpretation.

The proposed effective state-size metric is a method applicable to both linear systems and LIVs. As a quantitative proxy for memory utilization, it offers insights into the inner workings of causal sequence models, ensuring generality, usability, and interpretability.

**Synthetic and language benchmarks.** In this work, we build on synthetic tasks from the mechanistic architecture design (MAD) framework introduced in (Poli et al., 2024). MAD defines a set of small-scale tasks designed to evaluate key model capabilities, such as in-context recall (Akyürek et al., 2024; Bhattamishra et al., 2023; Elhage et al., 2021; Olsson et al., 2022b). Training models on these tasks is efficient, making them well-suited for exploring a large space of tasks and models, as demonstrated in several prior works (Dupont et al., 2019; Arora et al., 2024; Fu et al., 2023). In this work, we investigate the effective state-size across a subset of the MAD tasks: multi-query associative recall (MQAR), selective copying, and compression, varying the difficulty of each to gain a nuanced understanding of how effective state-size evolves across these task landscapes.

Among the synthetic tasks we examine, MQAR stands out in particular. Proposed by Arora et al. (2023), MQAR was designed to bridge the gap between synthetic and real language tasks explained by associative recall – the ability of a model to retrieve information based on relationships between different elements in its memory. This capability has long been sought after in the construction

of sequence model architectures (Ramsauer et al., 2021; Ba et al., 2016); as such, we evaluate the performance of our models on MQAR to measure the benefits of using effective state-size to iterate on canonical frameworks used in sequence modeling.

One notable aspect of MQAR observed in Arora et al. (2023) is that the size of the model cache needs to scale with the difficulty of the task to maintain performance. While this observation holds, our work demonstrates that model cache size is an imperfect measure in this context due to the discrepancy between memory capacity, as measured by theoretically realizable state size, and memory utilization, as measured by effective state-size. At a higher level, this demonstrates how our work provides a new perspective on analyzing memory-intensive synthetic tasks.

While the MAD framework and synthetic tasks have shown correlations with model performance on large-scale language tasks, language itself poses a unique challenge. Models are tasked with predicting the next token given previous tokens – a simple yet general objective. New tasks can be created simply by altering the prompts, thereby expanding the range of possible task domains.

Although numerous language evaluation tasks – such as those in Hendrycks et al. (2021); Wang et al. (2024); Zellers et al. (2019) – have been proposed, they often probe a narrow task space and tend to be brittle. For example, shuffling the order of multiple choices in MMLU can drastically change model rankings Alzahrani et al. (2024).

Unlike narrow benchmarks, perplexity scores can be computed across an entire pre-training dataset, covering a much broader task domain. However, small perplexity gaps between models make it a challenging metric for evaluation. Recently, Arora et al. found that much of the difference in perplexity between models can be attributed to bigram perplexity – a measure of a model's ability to utilize the context and predict a successor token (second token of a bigram) given a repeated context token (first token of a bigram) within a sequence. They demonstrate that most of the average perplexity difference between a gated convolution model and an attention model stems from differences in bigram perplexity, suggesting that recall is a key capability for language models.

The effective state-size analysis presented in this work reveals that strong recall performance as measured by bigram perplexity in language modeling tasks depends not only on memory capacity, but also on a model's ability to modulate its state-size within a given context.

## C  THEORETICAL BACKGROUND

### C.1  NOTATION

We adopt the following notation in this paper:

- Inputs, outputs, and operators follow flattened notation. i.e., $u, y \in \mathbb{R}^{\ell d}$ and $T \in \mathbb{R}^{\ell d \times \ell d}$. In particular, the original inputs and outputs with shape $\ell \times d$ are flattened in row-major ordering, resulting in $T$ having $\ell \times \ell$ sub-blocks, each of which is of size $d \times d$.

- Tensor subscripts index sequence indices (time-step) and superscripts index channel/hidden dimensions. I.e., for an input $u \in \mathbb{R}^{\ell d}$, $u_i \in \mathbb{R}^d$ denotes the input vector at sequence-index $i$, and $u^\alpha \in \mathbb{R}^\ell$ denotes the input vector for channel $\alpha$. Similarly, $T_{ij} \in \mathbb{R}^{d \times d}$ denotes the linear weighing of $u_j$ on to $y_i$.

- Indices within square brackets indicate matrix indices void of semantics (sequence index, channels, etc.). I.e., $A_{i[\alpha,\beta]}$ indexes row $\alpha$ and column $\beta$ of matrix $A_i$.

- Semicolons within subscripts denote a product over ranges ($A_{1;3} = A_1 A_2 A_3$).

- Tensor slices are denoted with colons and are inclusive over the ranges. I.e., $u_{0:2} = u_0 u_1 u_2$.

### C.2  DERIVATIONS AND PROOFS

#### C.2.1  THE OPERATOR REALIZATION OF LINEAR RECURRENCES

Unrolling the recurrence in Equation 1 unveils the following formulation:

$$s_0 = 0$$
$$s_1 = B_0 u_0$$
$$s_2 = B_1 u_1 + A_1(B_0 u_0)$$
$$s_3 = B_2 u_2 + A_2(B_1 u_1 + A_1(B_0 u_0))$$

$$s_i = \left( \sum_{j=0}^{i-1} \left[ \prod_{k=i-1}^{j+1} A_k \right] B_j u_j \right), \tag{C.2.1}$$

$$y_i = C_i \left( \sum_{j=1}^{i-1} \left[ \prod_{k=i-1}^{j+1} A_k \right] B_j u_j \right) + D_i u_i, \tag{C.2.2}$$

which corresponds to the operator:

$$T_{ij} = \begin{cases} 0 & i < j \\ D_i & i = j \\ C_i A_{i-1;j+1} B_j & i > j \end{cases} . \tag{C.2.3}$$

#### C.2.2  FACTORIZING THE OPERATOR REALIZATION SUBMATRIX $H_i$

Factorizing the strictly lower triangular submatrices of the operator ($T_{i:,:i-1}$) into causal and anti-causal factors, unveils that $n_i$ (i.e. the TSS) upper bounds the dimensionality of the inner product

between the factors, and thus, also the rank of the submatrix ($n_i \geq \text{rank}(H_i)$):

$$
T_{i:,:i-1} \equiv H_i = \begin{bmatrix} C_i & & \\ & \ddots & \\ & & C_{\ell-1} \end{bmatrix} \begin{bmatrix} A_{i-1;1} & \cdots & I \\ \vdots & \ddots & \vdots \\ A_{\ell-2;1} & \cdots & A_{\ell-2;i} \end{bmatrix} \begin{bmatrix} B_0 & & \\ & \ddots & \\ & & B_{i-1} \end{bmatrix}
$$

$$
= \begin{bmatrix} C_i & & \\ & \ddots & \\ & & C_{\ell-1} \end{bmatrix} \begin{bmatrix} I \\ A_i \\ A_{i+1;i} \\ \vdots \\ A_{\ell-2;i} \end{bmatrix} \begin{bmatrix} A_{i-1;1} & A_{i-1;2} & \cdots & A_{i-1} & I \end{bmatrix} \begin{bmatrix} B_0 & & \\ & \ddots & \\ & & B_{i-1} \end{bmatrix}
$$

$$
= \underbrace{\begin{bmatrix} C_i \\ C_{i+1}A_i \\ C_{i+2}A_{i+1;i} \\ \vdots \\ C_{\ell-1}A_{\ell-2;i} \end{bmatrix}}_{\substack{d(\ell-i) \times n_i \\ \textit{anti-causal}}} \underbrace{\begin{bmatrix} A_{i-1;1}B_0 & A_{i-1;2}B_1 & \cdots & A_{i-1}B_{i-2} & B_{i-1} \end{bmatrix}}_{\substack{n_i \times di \\ \textit{causal}}} \equiv \mathcal{O}_i \mathcal{C}_i.
$$

(C.2.4)

Besides unveiling the relationship between the rank of the realized operator and the original state-size $n_i$, the following insights can be drawn from the decomposition:

- The causal portion $\mathcal{C}_i$ is the input-state projection matrix at time-step $i$ (i.e., $s_i = \mathcal{C}_i u_{:i-1}$) corresponding to Equation (C.2.1).

- ESS ($\text{rank}(H_i)$) is simply the minimum rank between the causal and anti-causal projections.

- In conjunction with Theorem 3.2, we observe that the causally determinable minimal state-size (causal ESS) is equivalent to the rank of the causal projection. This insight allows us to construct a more efficient realization of the recurrence:
  - We can minimally factorize the causal projection as $\mathcal{C}_i = L_i R_i$, where $L_i \in \mathbb{R}^{n_i \times r}$ and $R_i \in \mathbb{R}^{r \times di}$, with $r = \text{rank}(\mathcal{C}_i)$.
  - The right factor $R_i$ becomes the new input-state projection matrix for $H_i$, effectively reducing the state dimension to the causal ESS.
  - $A_{i-1}^*$ and $B_{i-1}^*$ can be determined from $R_i$ using the process outlined in Theorem 3.1, and $C_i^* = C_i L_i$.

### C.2.3 THE TRIVIAL RECURRENCE REALIZATION

Any input-varying and input-invariant causal operator can be trivially realized with the following recurrence:

$$
s_{i+1} = \begin{bmatrix} I_{(di)} \\ 0_{(d)} \end{bmatrix} s_i + \begin{bmatrix} 0_{(di)} \\ I_{(d)} \end{bmatrix} u_i,
$$

$$
y_i = \begin{bmatrix} T_{i,0} & T_{i,1} & \cdots & T_{i,i-1} \end{bmatrix} s_i + T_{i,i} u_i.
$$

(C.2.5)

In simple terms, the state $s_i$ stores each input from $t \in [i-1]$, which is then mapped to the output with operator features at row $i$. Note that in the case where the operator is input varying, the trivial realization upholds the causality of the featurization process (i.e. the features $(A_i, B_i, C_i, D_i)_{i \in [\ell]}$ of the trivial realization are causally determined). Moreover, the causally determined ESS (see Section C.2.2) for the trivially realized recurrence is equivalent to its TSS, as $\mathcal{C}_i = I_{di}$.

### C.2.4 EXISTENCE OF INFINITE RECURRENT REALIZATIONS (PROOF OF THEOREM 3.1)

**Theorem 3.1** *Given any causal input-invariant operator $T$, there exist infinite variations of linear recurrences in the form of Equation (1) that realize an equivalent input-output operator.*

*Proof.* We first categorize the operator into two portions: the memoryless portion, where $i = j$, and the dynamical portion, where $i > j$. The memoryless portion can be trivially realized by setting $D_i = T_{ii}$. For the dynamical portion, we draw inspiration from (DeWilde & van der Veen, 1998, ch. 3) and approach the proof of existence by ansatz. The following steps outline the proof:

1. Section C.2.2 demonstrates that, given a linear recurrence in the form of Equation (1), the operator submatrix can be factorized into causal and anti-causal parts, where the causal part represents the input-state projection matrix. Therefore, we proceed by making the following ansatz: for any operator submatrix $T_{i:,:i-1} \equiv H_i$, $H_i$ can be arbitrarily factorized into $\mathcal{O}_i \in \mathbb{R}^{d(\ell-i) \times n_i}$ and $\mathcal{C}_i \in \mathbb{R}^{n_i \times di}$, and that $\mathcal{C}_i$ represents the input-state projection at time-step $i$ (i.e., $s_i = \mathcal{C}_i u_{:i-1}$).

2. Construct the dynamic features $(A_i, B_i, C_i)_{i \in [\ell]}$ such that the assumption above holds. Note that we additionally assume the initial and final states to be 0 without loss of generality, therefore the realization of $C_0, A_0, A_{\ell-1}$, and $B_{\ell-1}$ could be ignored.

   (a) Set $C_i = \mathcal{O}_{i[:d-1]}$ to obtain $(C_i)_{i \in [1,\ell]}$, as given the assumptions above, the first set of rows of $\mathcal{O}_i$ linearly projects $s_i$ onto $y_i - D_i u_i$, which is identical to $C_i$ in Equation (1).

   (b) Set $B_{i-1} = \mathcal{C}_{i[:,-d:]}$ to obtain $(B_i)_{i \in [\ell-1]}$, for which the identity can be obtained by deconstructing the input-state projection matrix $\mathcal{C}_i$ and equating its assumed state $s_i$ with Equation (1).

$$\begin{aligned} s_i &= A_{i-1}s_{i-1} + B_{i-1}u_{i-1} \\ &= \mathcal{C}_{i[:,:-d-1]}u_{:i-2} + \mathcal{C}_{i[:,-d:]}u_{i-1}. \end{aligned} \tag{C.2.6}$$

   (c) Using the same state-dynamics equation, we could equate the assumed state-projection matrices with each other, obtaining $(A_i)_{i \in [1,\ell-1]}$:

$$\begin{aligned} s_{i+1} &= A_i s_i + B_i u_i \\ \mathcal{C}_{i+1}u_{:i} &= A_i \mathcal{C}_i u_{:i-1} + \mathcal{C}_{i+1[:,-d:]}u_i \\ \mathcal{C}_{i+1[:,:-d-1]}u_{:i-1} &= A_i \mathcal{C}_i u_{:i-1} \\ A_i &= \mathcal{C}_{i+1[:,:-d-1]}\mathcal{C}_i^+. \end{aligned} \tag{C.2.7}$$

3. Verify that the realized recurrence maps back to the original operator $T_{ij}$, proving that arbitrary factorizations (of which there are infinite variations) of the operator submatrices can be used to construct equivalent operators.

$$\begin{aligned} T_{ij} = C_i A_{i-1} \cdots A_{j+1} B_j &= \mathcal{O}_{i[:d-1]}\mathcal{C}_{i[:,:-d-1]}\ldots \mathcal{C}_{j+2}^+\mathcal{C}_{j+2[:,:-d-1]}\mathcal{C}_{j+1}^+\mathcal{C}_{j+1[:,-d:]} \\ &= \mathcal{O}_{i[:d-1]}\mathcal{C}_{i[:,:-d-1]}I_{[:,:(j+1)d-1]}I_{[:,-d:]} \\ &= \mathcal{O}_{i[:d-1]}\mathcal{C}_{i[:,jd:(j+1)d-1]} = H_{i[:d-1,jd:(j+1)d-1]} = T_{ij}. \end{aligned} \tag{C.2.8}$$

$\square$

As an example, $H_i$ can be factorized minimally with SVD as follows:

$$\mathcal{O}_i \mathcal{C}_i = (U_{(r)}D_{(r)}^{1/2})(D_{(r)}^{1/2}V_{(r)}),$$

where $U_{(r)} \in \mathbb{R}^{m \times r}$, $D_{(r)} \in \mathbb{R}^{r \times r}$, $V_{(r)} \in \mathbb{R}^{r \times n}$ are the $r$-truncated SVD decompositions, and $r = $ rank of $H_i \in \mathbb{R}^{m \times n}$. Theorem 3.2 demonstrates that these factors realizes a recurrence with minimal state-size.

### C.3 MINIMAL RECURRENT REALIZATION (PROOF OF THEOREM 3.2)

**Theorem 3.2** *The rank of the operator submatrix ($H_i \equiv T_{i:,:i-1}$) determines the minimal state size required to represent the causal operation ($y = Tu$) as a recurrence.*

*Proof.* The proof of Theorem 3.1 shows that the operator submatrices $H_i$ can be decomposed arbitrarily into two state-projection matrices, $\mathcal{O}_i$ and $\mathcal{C}_i$, whose inner product dimension defines the state size of its recurrent realization at sequence index $i$. By the rank-nullity theorem, $\mathrm{rank}(H_i)$ represents the minimum inner product dimension of any such state-projection matrices and thus corresponds to the minimally realizable state size of the operator $T$ at sequence index $i$. □

## D METHODS

### D.1 ADDITIONAL DETAILS ON COMPUTING ESS

#### D.1.1 COMPUTING ESS FOR SISO MODELS

In this section, we provide additional details on the distinction between computing ESS in single-input single-output (SISO) models like S4 and multi-input multi-output (MIMO) models like S5.

Recall in Sections 3 and C.1, we introduced flattened notation, as it offers a general framework for formulating a wide range of operators and recurrences. Namely, a MIMO layer like S5 (Smith et al., 2023), which mixes both the channels and sequence simultaneously, can be formulated as $y = Tu$ (with the operator realization outlined in C.2.1) in the same way a SISO layer like S4 (Gu et al., 2022a) can, which only mixes the sequence. The difference between these two models lies in the structure of $T$: for models that only mix the sequence, such as S4, $T_{ij}$ is diagonal; for S5, it is dense.

Note that since all of the models in our experiments are SISO, we can compute the ESS independently for each channel using the standard operator formulation $T^{\alpha\alpha} \in \mathbb{R}^{\ell \times \ell}$, where $\alpha \in [d]$. This approach is significantly more efficient than computing ESS for the multi-channel (flattened) representation. Furthermore, in the case of attention layers, the computation can be further reduced to only the $h$ independent heads, as the operator (i.e. the attention matrix) is shared across channels within the same head.

#### D.1.2 TOLERANCE VS ENTROPY ESS

Regarding the distinction between the entropy and tolerance-based forms of ESS, we note that entropy-ESS is a valuable summary metric because its computation is independent of any specific tolerance value chosen. However, it can potentially be misleading when comparing ESS across sequence indices due to the unequal normalization applied to the singular values. Conversely, when comparing entropy-ESS across different operators, it can be useful as the normalization removes the effect of the norm of the operator. Moreover, in contrast to the tolerance-based metric which is discrete, entropy-ESS can assume continuous values ranging from 1 to $|\mathbf{\Sigma}_i|$.

In most of our experiments, we observe consistent trends between entropy-ESS and tolerance-ESS when the metrics are marginalized over the sequence length. Therefore, unless stated otherwise, our figures are presented using the entropy-ESS. In cases where we require ESS comparison across the sequence dimension, we instead plot ESS for multiple tolerance values. Finally, we note that for the analyses presented in Section 4, when computing ESS, we average across 8 samples (batch-size). For the rest of the ESS analyses, we average across 32 samples.

#### D.1.3 PYTORCH IMPLEMENTATION

Below, we provide a PyTorch implementation of various ESS metrics and helper functions that were leveraged in our analyses:

```
1  import torch
2
3  def T2H_i(T, i, d=1):
4      """
5      Extract H_i from T.
```

```python
    Args:
        - T: Flattened operator with shape [..., d*L, d*L].
        - i: Index of H (H_i) to retrieve.
        - d: Block size for multi-channel flattened operator
    representation (default is 1).

    Returns:
        - H_i: Submatrix of the operator at index i.
    """
    return T[...,d*i:,:d*i]

@torch.no_grad()
def T2Ss(T, d=1):
    """
    Converts an operator into a list of singular values (Ss).

    Args:
        - T: Flattened operator with shape [..., d*L, d*L]
        - d: Block size for multi-channel flattened operator
    representation (default is 1).

    Returns:
        - Ss: A list of singular values for each sequence index in T.
    """
    seqlen = T.size(-2)//d
    Ss = []
    for i in range(1, seqlen):
        H_i = T2H_i(T, i, d)
        _, S_i, _ = torch.svd(H_i)
        Ss.append(S_i)
    return Ss

@torch.no_grad()
def Ss2ToleranceESS(Ss, tol=1e-4):
    """
    Computes the tolerance-ESS from the list of singular values.

    Args:
        - Ss: List of singular values.
        - tol: Tolerance value.

    Returns:
        - tolerance-ESS
    """
    ranks = []
    for SV in Ss:
        rank = torch.sum(SV>=tol, dim = -1)
        ranks.append(rank)
    ranks = torch.stack(ranks, dim=-1)
    return ranks

@torch.no_grad()
def Ss2EntropyESS(Ss, clip=1e-12):
    """
    Computes the entropy-ESS from the list of singular values.

    Args:
        - Ss: List of singular values.
        - clip: clips probabilities below this value avoiding numerical
    instabilities when the probabilities are too numerically close to 0.

    Returns:
        - entropy-ESS
    """
```

```
68    ranks = []
69    for SV in Ss:
70        p = SV/SV.sum(dim=-1)[..., None]
71        p = torch.clip(p, clip)
72        H = -torch.sum(p * torch.log(p), dim=-1)
73        rank = torch.exp(H)
74        ranks.append(rank)
75    ranks = torch.stack(ranks, dim=-1)
76    return ranks
```

Example usage (Python-pseudocode):

```
1 >>> out = model(u, output_attentions=True)
2 >>> # T shape: [bs, layers, heads, len, len]
3 >>> T = out.attention_matrix
4 >>> Ss = T2Ss(T)  # List of singular values
5 >>> # ESS shape [bs, layers, heads, len-1]
6 >>> ESS = Ss2ToleranceESS(Ss, tol=1e-3)
7 >>> mean_ESS = torch.mean(ESS)
```

We note that calculating the effective rank may cause numerical instability when $p_i^m$ approaches 0 due to the logarithmic term. This is partially mitigated by clipping the normalized singular values, as shown above.

### D.2 FORMULATION OF THE FEATURIZERS

In this section, we first establish the equivalence between linear attention and state-space models, then proceed with formulating the LIVs discussed in this paper.

**Linear attention and state-space model equivalence.** We begin by demonstrating that linear attention models are state-space models, serving as the foundation for the subsequent formulation of featurizers for other models, such as gated linear attention and weighted linear attention.

A single linear attention head with dimension $d/h$, typically formulated as

$$y = qk^T v, \tag{D.2.1}$$

in which $q, k, v \in \mathbb{R}^{\ell \times d/h}$ are input features. They can be reformulated as recurrences with matrix-valued states $s_i \in \mathbb{R}^{d/h \times d/h}$ as follows (Katharopoulos et al., 2020):

$$
\begin{aligned}
s_i &= s_{i-1} + k_i v_i^T \\
y_i &= q_i^T s_i,
\end{aligned}
\tag{D.2.2}
$$

Without loss of generality, applying column-major flattening to the matrix-valued state and treating $v_i$ as the input $u_i$, the recurrence can be formulated like Equation (1), by setting $A_i = I_{(d/h)^2}$ and:

$$
B_{i-1} = \begin{bmatrix}
k_i^1 & 0 & \cdots & 0 \\
\vdots & \vdots & \ddots & \vdots \\
k_i^{d/h} & 0 & \cdots & 0 \\
0 & k_i^1 & \cdots & 0 \\
\vdots & \vdots & \ddots & \vdots \\
0 & k_i^{d/h} & \cdots & 0 \\
\vdots & \vdots & \vdots & \vdots \\
0 & 0 & \cdots & k_i^1 \\
\vdots & \vdots & \ddots & \vdots \\
0 & 0 & \cdots & k_i^{d/h}
\end{bmatrix}, \quad
C_i = \begin{bmatrix}
q_i^1 & \cdots & q_i^{d/h} & 0 & \cdots & 0 & \cdots & 0 & \cdots & 0 \\
0 & \cdots & 0 & q_i^1 & \cdots & q_i^{d/h} & \cdots & 0 & \cdots & 0 \\
\vdots & \ddots & \vdots & \vdots & \ddots & \vdots & \cdots & \vdots & \ddots & \vdots \\
0 & \cdots & 0 & 0 & \cdots & 0 & \cdots & q_i^1 & \cdots & q_i^{d/h}
\end{bmatrix}.
$$

$$\tag{D.2.3}$$

Notice that each individual channel forms a single-input-single-output (SISO) recurrence (like many of the SSM architectures including S4, S6, Mamba2, and more [27; 26; 16; 48; 49]), as there is no mixing across channels. Additionally, each of these SISO recurrences has a state-size of $d/h$.

**Formulation of the featurizers.** For the sake of completeness, we additionally characterize the "values" feature in attention-like models, with $f_u(u)$ as follows:

$$s_{i+1} = A_i s_i + B_i f_u(u_i)$$
$$y_i = C_i^T s_i + D_i f_u(u_i). \tag{D.2.4}$$

The following lists the formulations of the recurrent featurizers based on the results above, along with their per-channel (i.e. average) TSS and total TSS:

- **Linear Attention** (LA):

$$A^k = I, \quad B_{i-1}^k = \text{RoPE}(W_B^k u_i), \quad C_i^k = \text{RoPE}(W_C^k u_i), \quad f_u(u_i) = W_u^k u_i, \tag{D.2.5}$$

  where $W_C^k, W_B^k, W_u^k \in \mathbb{R}^{d/h \times d}$; $d$ and $h$ represent the number of channels and heads, respectively. $A$ is a fixed identity matrix. Per-channel TSS is $n_i^k = d/h$, and the total TSS is $d^2/h$. Each channel $c \in [d]$ is grouped into heads, where the head index corresponding to the channel is given by $k = \lfloor ch/d \rfloor$, and within a head, all corresponding projection matrices ($W_C^k$, $W_B^k$, etc.) are weight tied (shared). Moreover, a rotational positional encoding (RoPE) is by default applied to the $B$ and $C$ projections (Su et al., 2023).

- **Gated Linear Attention** (GLA):

$$A_{i-1}^k = \text{diag}(\text{sigmoid}(W_{A_2}^k W_{A_1} u_i)^{1/\beta}),$$
$$B_{i-1}^k = W_B^k u_i, \quad C_i^k = W_C^k u_i, \quad f_u(u_i) = W_u^k u_i, \tag{D.2.6}$$

  where besides having projections identical to those in LA, $W_{A_1} \in \mathbb{R}^{16 \times d}$ and $W_{A_2}^k \in \mathbb{R}^{d/h \times 16}$. Like LA, per-channel TSS is $n_i^k = d/h$, and the total TSS is $d^2/h$. By default, $\beta$ is set to 16.

- **Weighted Linear Attention** (WLA):

$$A^k = \text{diag}(\text{sigmoid}(\hat{A}^k)^{1/\beta}),$$
$$B_{i-1}^k = W_B^k u_i, \quad C_i^k = W_C^k u_i, \quad f_u(u_i) = W_u^k u_i, \tag{D.2.7}$$

  where $W_C$, $W_B$, and $W_u$ are identical to those in LA, and $\hat{A}^k \in \mathbb{R}^{d/h}$ is explicitly parameterized and initialized to 0. Like LA, per-channel TSS is $n_i^k = d/h$, and the total TSS is $d^2/h$.

- **Softmax Attention** (SA):

$$\hat{B}_i^k = \text{RoPE}(W_B^k u_i), \quad \hat{C}_i^k = \text{RoPE}(W_C^k u_i),$$
$$T^k = \text{softmax}(\hat{C}^k (\hat{B}^k)^T), \quad f_u(u_i) = W_u^k u_i, \tag{D.2.8}$$

  where $W_C$, $W_B$, and $W_u$ are identical to those in LA, and $T$ can be converted into a recurrence using the trivial realization in Equation C.2.5. Therefore, the per-channel TSS is $i$, and total TSS is $id$. Like LA, a rotational positional encoding (RoPE) is by default applied to the $\hat{B}$ and $\hat{C}$ projections (Su et al., 2023).

- **S6** (Gu & Dao, 2024):

$$\Delta^c = \text{softplus}(W_\Delta^c u_i + b^c), \quad A_{i-1}^c = \text{diag}(\exp(-\hat{A}\Delta^c)),$$
$$B_{i-1}^c = \Delta^c W_B u_i, \quad C_i = W_C u_i, \tag{D.2.9}$$

  where $\hat{A} \in \mathbb{R}^n$ is initialized to $\begin{bmatrix} 1 & 2 & \cdots & n \end{bmatrix}^T$, $c$ is the channel index, $W_C, W_B \in \mathbb{R}^{n \times d}$, and $W_\Delta^c \in \mathbb{R}^{1 \times d}$. Here, per-channel TSS is $n$, and total TSS is $nd$. Note that by setting $n = d/h$, S6 resembles GLA, with the following exceptions:

  - S6 has only one (not $h$) different projection matrices for $B$ and $C$.

  - S6 has channel-wise projections for the input-varying discretization applied to $B$ and $C$.

- S6 has an explicitly parameterized vector valued $\hat{A}$.

- S6 has some minor differences in the non-linearity applied to keep $0 < A < 1$. Similar to GLA, S6 also has diagonal $A$ matrices, whereas in Mamba2 Dao & Gu (2024), the A matrix is scalar-valued. However, the $B$ and $C$ projections in Mamba2 more closely resemble that of GLA.

- **GLA-S6**:

$$A_{i-1}^h = \text{diag}(\exp(-[1/\alpha \quad 2/\alpha \quad \ldots \quad n/\alpha]^T \odot \text{softplus}(W_{A_2}^h W_{A_1} u_i))),$$
$$B_{i-1}^h = W_B^h u_i, \quad C_i^h = W_C^h u_i, \quad f_u(u_i) = W_u^k u_i, \tag{D.2.10}$$

GLA-S6 is a combination of S6 and GLA such that the $B$ and $C$ projections are identical to that of GLA, while $A$ is featurized similarly to S6. Namely, it has identical $W_B$, $W_C$, and $W_u$ to those found in LA which means that the per-channel TSS is $d/h$ and total TSS is $d^2/h$. The $A$ matrix is featurized with the `arange` term ($[1 \quad 2 \quad \cdots n]^T$) like S6. We additionally added a normalization hyperparameter $\alpha$, which controls the rate at which elements of $A$ decay to 0.

## D.3 Empirical Validation

Here, we provide details on the task-model sweep presented in Section 4. Table 1 lists the hyper-parameters that were exhaustively swept across to generate the task-model space. Note that the hyperparameter controlling the task difficulty is task-dependent (for more details, see Poli et al. (2024)).

For the MQAR and selective copying tasks, a default vocab size of 8192 (Arora et al., 2023) was used for all models. For the compression tasks, the vocab size was varied to modulate task difficulty, as shown in Table 1. Any other task settings not specified here are defaulted to those presented in Arora et al. (2023). Two important constraints on the tasks from Arora et al. (2023) which we also utilize in our experiments are as follows: MQAR task requires that

$$4 * \text{num kv pairs} \leq \text{seq len}$$

and the selective copying task requires that

$$2 * \text{num tokens to copy} + 1 < \text{seq len}$$

Any of the task configurations from Table 1 that violate these conditions were not trained. This is why the SA plot in Figure 4 has empty spots in the grid.

Finally, we note that all architectures analyzed here consist of 4 layers: 2 sequence mixing layers (i.e. one of GLA, LA, WLA or SA) and 2 channel mixing layers (i.e. MLPs).

| Configuration | Value(s) |
|---|---|
| Tasks | MQAR, selective copying, compression |
| Num. key-value pairs | 8, 16, 32, 64, 128 |
| Num. tokens to copy | 8, 16, 32, 64, 128 |
| Vocab size (compression) | 8, 16, 32, 64, 128 |
| Vocab size (MQAR and selective copying) | 8192 |
| Sequence length | 64, 128, 256, 512, 1024, 2048 |
| Model (featurizer) | GLA, LA, WLA, SA |
| Model width | 64, 128, 256, 512 |
| Number of heads | 4, 8 |
| Optimizer | AdamW |
| Learning Rate | 0.002 |
| Weight Decay | 0.1 |
| Batch Size | 64 |
| Epochs | 70 |
| Steps Per Epoch | 2000 |
| Num. Training Samples | 128k |
| Num. Testing Samples | 6.4k |

Table 1: Set of hyperparameters for the task-model sweep.

Regarding the post-hoc analysis performed on the sweep, we note the following:

- Since the average TSS computed over the channels (which equals $\frac{\text{model width}}{\text{number of heads}}$ for GLA, LA, and WLA) explains more meaningful variation with respect to memory utilization than model width and number of heads individually, we consolidate those two dimensions into one by analyzing across the average TSS axis. For SA, since average TSS is a function of the task rather than model hyperparameters (see Equation C.2.5 and Section D.2), we instead compute the sum of TSS over all $d$ channels, given by the total TSS per layer $= d*i$. In any cases where the average/total qualifier is not specified, note that we are referring to the average ESS or TSS.

- Since we analyze the recurrent models across the average TSS dimension, we compute average ESS in the plots presented in Section 4.1 in order to compare ESS and TSS as proxies for performance. Similarly, since we analyze the SA models across the total TSS dimension, we compute total ESS for those plots. However, we note that plots for both the average/total ESS and TSS are presented in Section E.1.

- When we marginalize across dimensions, we average across all models in that bucket of task-model space. For example, in Figure 4, for each (TSS, kv) pair, we average over the correlations of all models that correspond to that pair. Note, however, that we never average across tasks (i.e. MQAR, selective copying, compression) or featurizers (i.e. GLA, LA, WLA, SA).

- When we compute cross-model correlations (Figure 2) for SA, we filter out models which have an accuracy $> 0.95$. This is done in order to observe meaningful variation as a function of (total ESS)/kv and (total TSS)/kv since many of the SA models obtain an accuracy of 1.

- When we compute within-model correlations (Figure 4) for MQAR, we drop epoch 0 from the computation since we observe a phase at the start of training in which ESS tends to decrease, but accuracy does not change. We elaborate on this phenomenon in Section E.1 and hope to characterize it further in future work.

- Regarding the task-adjusted forms of ESS and TSS which, in the case of MQAR, are computed by normalizing the raw ESS value by the number of kv-pairs in the task, we note that this normalization factor is critical for observing the cross task-model correlations presented in Figure 2. In particular, in Figure 9, we find that correlations across the task-model space break down when examining the unnormalized ESS. This points to the higher-level notion that ESS is expected to scale with the memory demands of the task.

- We interpret the state utilization of a model, which is given by ESS/TSS, as a proxy for what portion of the memory capacity of the network is realized in practice. By definition, state utilization takes on values ranging continuously from 0 to 1. Recall that a state utilization near 1 is indicative of state saturation.

- While for most of the ESS analysis conducted on the sweep we use the entropy-ESS, we note that for the state utilization plot presented in Figure 5, we use the tolerance-ESS with a tolerance level set at 1e-3. We do this because we find that entropy-ESS fails to capture the state collapse phenomenon. This is because state collapse is primarily dictated by the magnitude of the singular values, as opposed to the relative decay rate of the entire spectrum. In particular, if all of the singular values are close to 0, the layer is likely failing to learn an expressive state, resulting in poor performance. Due to the normalization applied to the spectrum, the entropy-ESS metric may potentially present this state as having a high effective rank; however, in practice, we know that this is a misrepresentation of the true dynamics. Tolerance-ESS, in contrast, appropriately captures the dynamics of the state with respect to the norm of the operator. Because of this, whenever we analyze ESS as it pertains to state collapse (e.g. Figure 8a), we present the tolerance-ESS instead.

## D.4 ESS-INFORMED FEATURIZER SELECTION AND INITIALIZATION SCHEME

| Configuration | Value |
|---|---|
| Model width | 128 |
| Num. heads | 8 |
| `arange` Norm. $(\alpha)$[a] | 1000 |
| Logit Norm. $(\beta)$ | 16 |
| $K$-expansion[b] | 1 |

Table 2: Default GLA hyperparameters.

| Configuration | Value |
|---|---|
| Model width | 128 |
| State expansion (`d_state`) | 16 |

Table 3: Default S6 hyperparameters.

---

[a]For GLA-S6.

[b]$K$-expansion is used to vary TSS in the featurizer experiments.

| Configuration | Value |
|---|---|
| Sequence length | 2048 |
| Num. KV Pairs | 128 |
| KV Dist. Const. | 0.1 |
| Optimizer | AdamW [a] |
| Learning Rate | 0.002 |
| Weight Decay | 0.1 |
| Batch Size | 64 |
| Epochs | 70 |
| Steps Per Epoch | 2000 |
| Num. Training Samples | 128k |
| Num. Testing Samples | 6.4k |
| Vocabulary Size | 8192 |

Table 4: Default MQAR task settings employed throughout the featurizer and initialization experiments in Section 5.1.

---

[a] Loshchilov & Hutter (2019)

## D.5 ESS-INFORMED REGULARIZATION

We use the following MQAR configuration for the regularization experiments presented in Section A.

| Configuration | Value |
|---|---|
| Sequence length | 4096 |
| Num. KV Pairs | 128 |
| KV Dist. Const. | 0.1 |
| Optimizer | AdamW [a] |
| Learning Rate | 0.002 |
| Weight Decay | 0.1 |
| Batch Size | 64 |
| Epochs | 70 |
| Steps Per Epoch | 2000 |
| Num. Training Samples | 128k |
| Num. Testing Samples | 6.4k |
| Vocabulary Size | 8192 |
| Model width | 128 |
| Num. heads | 8 |

Table 5: MQAR task settings and model hyperparameters employed throughout the midtraining experiments in Section A.

[a]Loshchilov & Hutter (2019)

Regarding the regularization scheme itself, since we examine models with two sequence mixing layers, we explore the following strategies: regularizing both layers, only regularizing the first layer and only regularizing the second layer. Empirically, we find that only regularizing the second layer performs the best and is thus the result presented in Figure 8b. We elaborate on why this is the most successful strategy in Section E.3.

## D.6 ESS-INFORMED MODEL-ORDER REDUCTION

The teacher models used in the distillation experiments are 2-layer GLA models (Yang et al., 2024a) with dimension = 128 and TSS = 256 (num_heads = 8 and expand_k = 16). We checkpointed the models every 10 epochs while training on MQAR across different task difficulties. The task ranges are given as follows:

- Sequence length: [512, 1024, 2048]

- Number of Key-Value Pairs: [64, 128]

Other settings follow the defaults shown in Table 4. For each task difficulty pair, we repeated the training run with three different seeds. For each teacher model checkpoint, both layers were distilled independently with student models of different state-sizes (16, 32, 64, and 128). Distillation settings are shown in Table 6.

The ESS metric in Figures 6d, 30, and 31 was computed by taking the minimum across the batch and channels, evaluated at the mid-point of the sequence ($\ell/2$). Using the mid-point of the sequence as a summary statistic was done to reduce compute. The midpoint in particular was chosen as it is the point in the sequence at which $H_i$ has the greatest dimensions, retaining the largest amount of information from the original operator. Note that in practice, where a large task space isn't being tested to validate our approach, a more thorough computation of ESS across sequence length is feasible. Other marginalization approaches such as taking the maximum or average across the batch and channels also show similar trends, but we found taking the minimum to best demonstrate the trend.

| Configuration | Value |
|---|---|
| Optimizer | AdamW |
| Batch Size | 1 |
| Learning Rate | 0.001 |
| Weight Decay | 0.0 |
| Training Steps (Operator) | 800 |
| Dropout (Operator) | 0.2 |
| Training Steps (Activation) | 3200 |
| Dropout (Activation) | 0.2 |

Table 6: Distillation settings used for the results presented in Section 5.2.

### D.7 ESS ANALYSIS FOR HYBRID NETWORKS

In our ESS analysis applied to hybrid networks, we restrict our scope to GLA-SA hybrids. In particular, we explore the following two settings:

- 8-layer hybrid networks in which 4 layers are sequence mixers (i.e. one of GLA or SA) and 4 layers are channel mixers (i.e. MLPs). We exhaust all possible hybrid networks (of which there are 16) and perform post-training, per-layer ESS analysis on the networks. We train these hybrid models on MQAR with task-model settings given below in Table 7.

- 16-layer hybrid networks in which 8 layers are sequence mixers (i.e. one of GLA or SA) and 8 layers are channel mixers (i.e. MLPs). Here, we explore all combinations of hybrid networks that follow the Jamba hybridization policy (Lieber et al., 2024) and perform post-training, per-layer ESS analysis on the networks. We train these hybrid models on MQAR with task-model settings given below in Table 8.

| Configuration | Value |
|---|---|
| Sequence length | 2048 |
| Num. KV Pairs | 512 |
| KV Dist. Const. | 0.1 |
| Optimizer | AdamW [a] |
| Learning Rate | 0.002 |
| Weight Decay | 0.1 |
| Batch Size | 64 |
| Epochs | 70 |
| Steps Per Epoch | 2000 |
| Num. Training Samples | 128k |
| Num. Testing Samples | 6.4k |
| Vocabulary Size | 8192 |
| Model width | 64 |
| Num. heads | 4 |

Table 7: Default MQAR task settings employed throughout the hybridization experiments conducted in the first setting described above.

---
[a]Loshchilov & Hutter (2019)

| Configuration | Value |
|---|---|
| Sequence length | 4096 |
| Num. KV Pairs | 1024 |
| KV Dist. Const. | 0.1 |
| Optimizer | AdamW [a] |
| Learning Rate | 0.002 |
| Weight Decay | 0.1 |
| Batch Size | 64 |
| Epochs | 70 |
| Steps Per Epoch | 2000 |
| Num. Training Samples | 128k |
| Num. Testing Samples | 6.4k |
| Vocabulary Size | 8192 |
| Model width | 16 |
| Num. heads | 2 |

Table 8: Default MQAR task settings employed throughout the hybridization experiments conducted in the second setting described above.

---

[a] Loshchilov & Hutter (2019)

Results for these experiments can be found in Section E.4.2.

### D.8  STATE MODULATION OF LARGE LANGUAGE MODELS

**State modulation of open-weight models.**  The following randomly generated sentences were used to study the effects of separator tokens on state modulation in open-weights pre-trained language models.

> <bos>*Mangoes are rich in vitamin C and can be blended into a refreshing smoothie*<sep> *Giraffes are the tallest mammals on Earth due to their long necks and legs*<sep> *She collects vintage typewriters from the 1940s*<sep> *Jupiter's Great Red Spot is a giant storm that has been raging for hundreds of years*<sep>

**State modulation on custom-trained 1B models.**  For our custom-trained 1B language models, we used longer sentences, as state modulation patterns were less discernible with shorter sequences. A collection of randomly generated sentences is shown below:

> <bos>*The deep blue ocean, teeming with an extraordinary array of marine life, from the smallest plankton to the largest whales, stretches out infinitely towards the horizon, a vast and mysterious expanse that has captivated the imaginations of explorers, scientists, and poets for centuries, hiding within its depths secrets yet to be discovered and stories yet to be told*<sep> *In a bustling city where skyscrapers tower over narrow streets filled with the constant hum of cars and the chatter of pedestrians, a small café, nestled between two imposing buildings, offers a quiet refuge for those seeking a moment of peace, with the comforting aroma of freshly brewed coffee and the soft sound of jazz music playing in the background, creating a cozy ambiance that feels like a world away from the urban chaos outside*<sep> *The ancient oak tree, with its gnarled branches stretching wide and its thick, sturdy trunk standing firm against the passage of time, has witnessed generations of families grow, seasons change, and countless stories unfold beneath its expansive canopy, becoming a silent guardian of the park, offering shade to those who seek solace and a sense of continuity in a rapidly changing world*<sep>

We note that the specific sentences and their order are not crucial to this analysis. Similar patterns have emerged with various sentence arrangements, provided the sentences are sufficiently long.

Training settings are outlined in Table 9.

| Configuration | Value |
|---|---|
| Batch Size | 16 |
| Max Sequence Length | 32k |
| Training Steps | 160k |
| Optimizer | AdamW |
| Learning Rate | 0.001 |
| Weight Decay | 0.1 |
| Num. Layers | 24 |
| Dimension | 2048 |

Table 9: 1B LLM settings.

The perplexity scores shown in Figure 7b were computed on 16k randomly sampled sequences over the FineWeb (Penedo et al., 2024) dataset. The raw perplexity samples were smoothed via a kernel density estimation method.

# E    EXTENDED EXPERIMENTAL RESULTS

## E.1    EMPIRICAL VALIDATION

In this section, we provide additional results and commentary from the sweep detailed in Section D.3 that were not presented in the main portion of the paper. One thing to note is that most of the ESS results presented in Section 4 were computed using the entropy-ESS. However, we also computed ESS using the tolerance-based approach to affirm that both forms of ESS showcase similar trends. In particular, we examined tolerances of 1e-1, 1e-3 and 1e-5. Since we observe similar trends across tolerances, we provide plots for a tolerance of 1e-3 below and omit the others for the sake of brevity.

### E.1.1    STATE COLLAPSE CONTINUED

Here, we continue our discussion on the state collapse phenomenon presented in Section 4.2. In particular, while we assert that state collapse is observable across all TSS in the high kv bucket for GLA/WLA, Figure 5 shows that accuracy differences between LA and GLA/WLA are only evident in the high TSS/high kv bucket of the task-model space. This is because state saturation is acting as a confounder, worsening performance in LA (see Figure 5 when TSS is 8). Therefore, although state collapse in GLA/WLA does not result in worse performance than LA in this specific task-model setting, it remains an issue even for models with smaller states when trained on sufficiently difficult tasks. This is the motivation behind the task-model setting explored in Section A.

### E.1.2    ENTROPY-ESS MQAR RESULTS CONTINUED

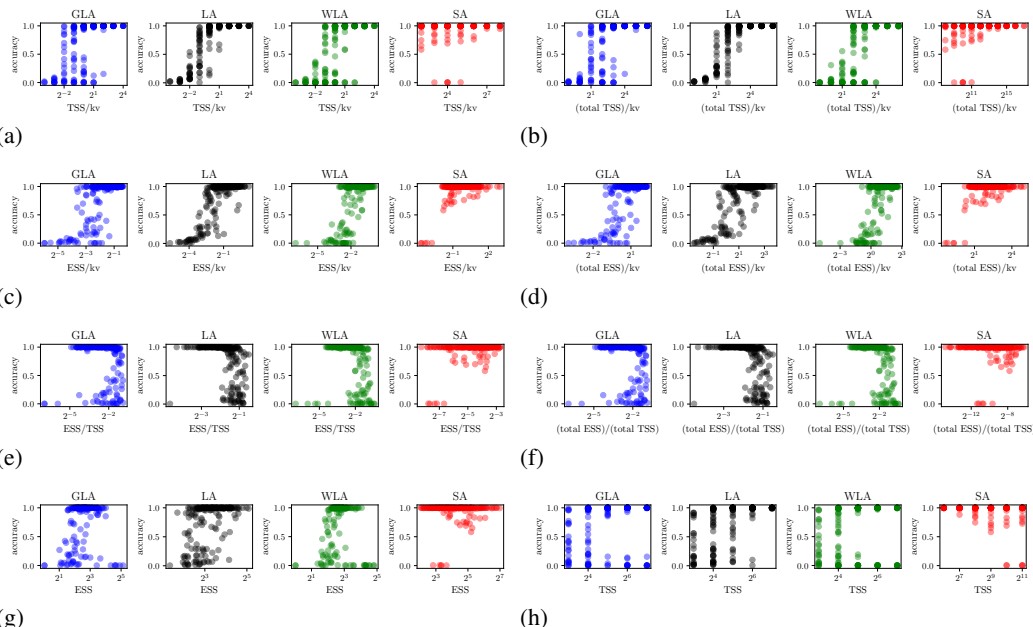

Figure 9: (a) TSS/kv vs accuracy across featurizers. This demonstrates that TSS/kv (i.e. memory capacity) is a worse proxy for model performance than ESS/kv as discussed in Section 4. (b) (total TSS)/kv vs accuracy across featurizers. This demonstrates that (total TSS)/kv is a worse proxy for model performance than (total ESS)/kv. (c) ESS/kv vs accuracy across featurizers. (d) (total ESS)/kv vs accuracy across featurizers. (e) ESS/TSS (i.e. state utilization) vs accuracy across featurizers. We note that models that saturate their state tend to perform worse on the task, which is evidence of the state saturation phenomenon discussed in Section 4.2. The models that do not saturate their state but still perform poorly are the models that undergo state collapse. (f) (total ESS)/(total TSS) vs accuracy across featurizers. (g) ESS vs accuracy across featurizers. Note that without normalizing by kv (i.e. the task memory), the correlation with accuracy breaks down substantially. (h) TSS vs accuracy across featurizers.

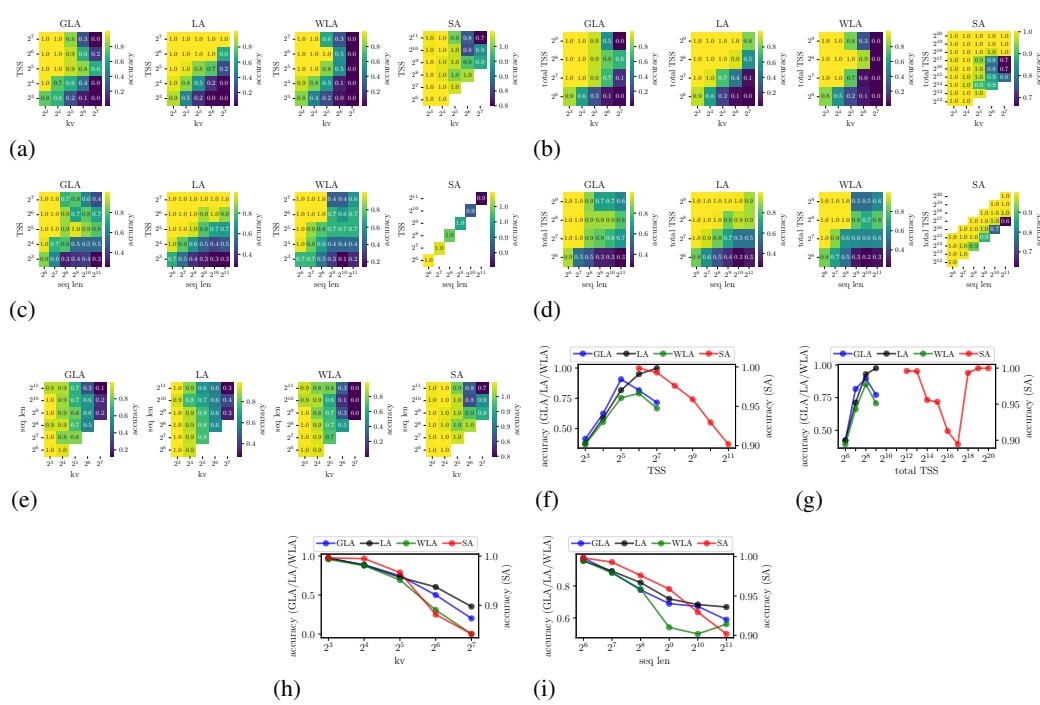

Figure 10: MQAR accuracies marginalized across different dimensions.

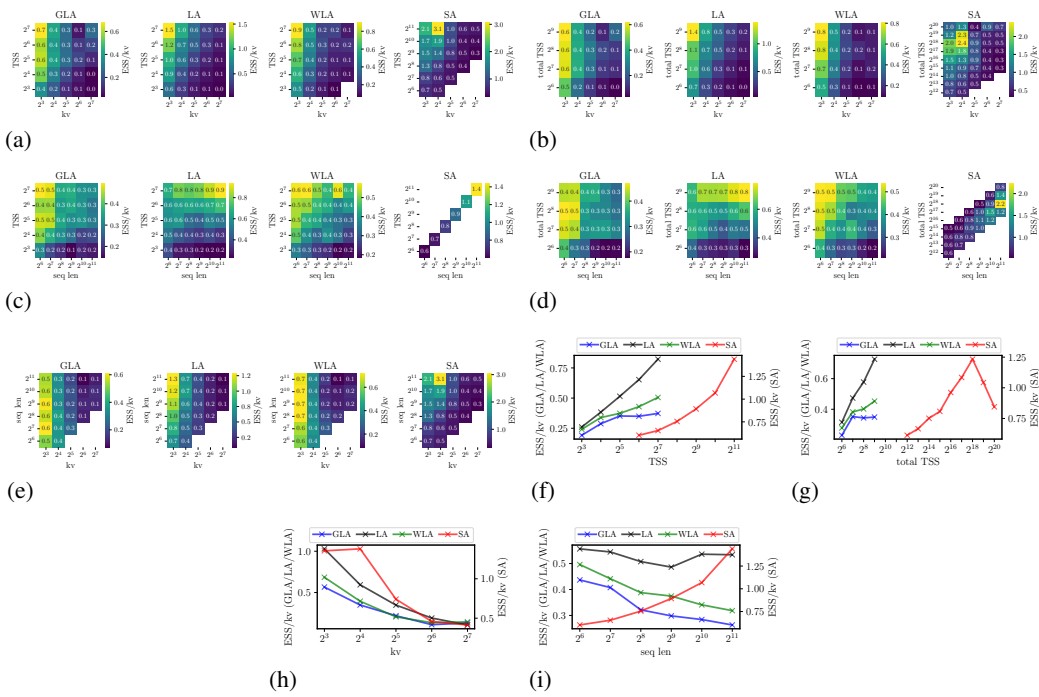

Figure 11: MQAR ESS/kv marginalized across different dimensions.

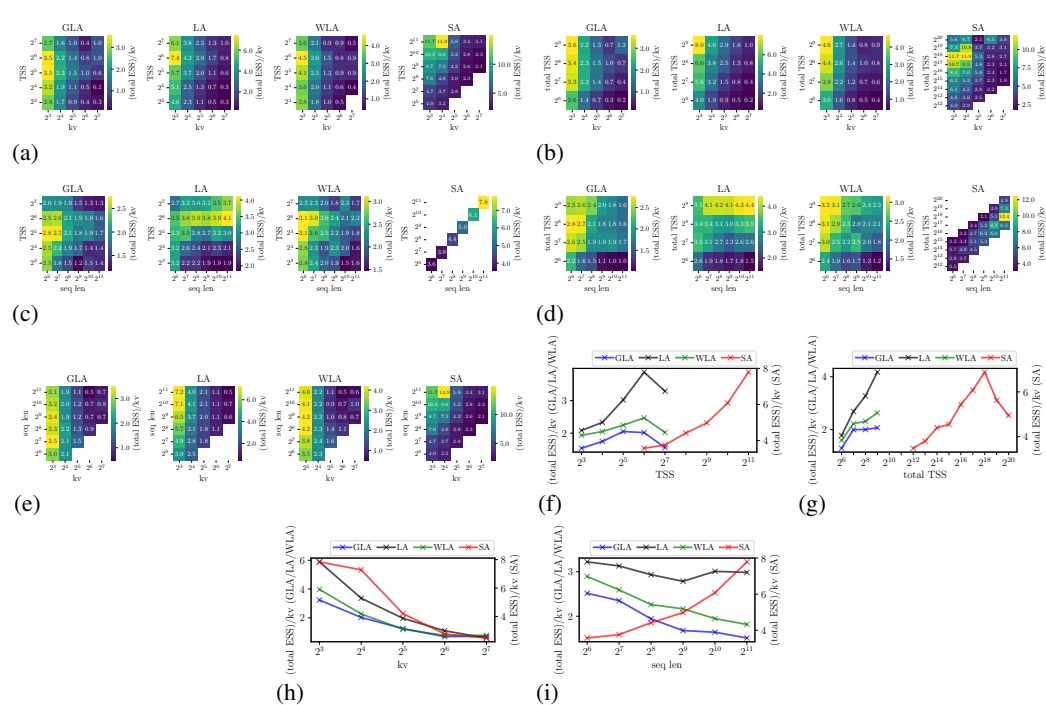

Figure 12: MQAR (total ESS)/kv marginalized across different dimensions.

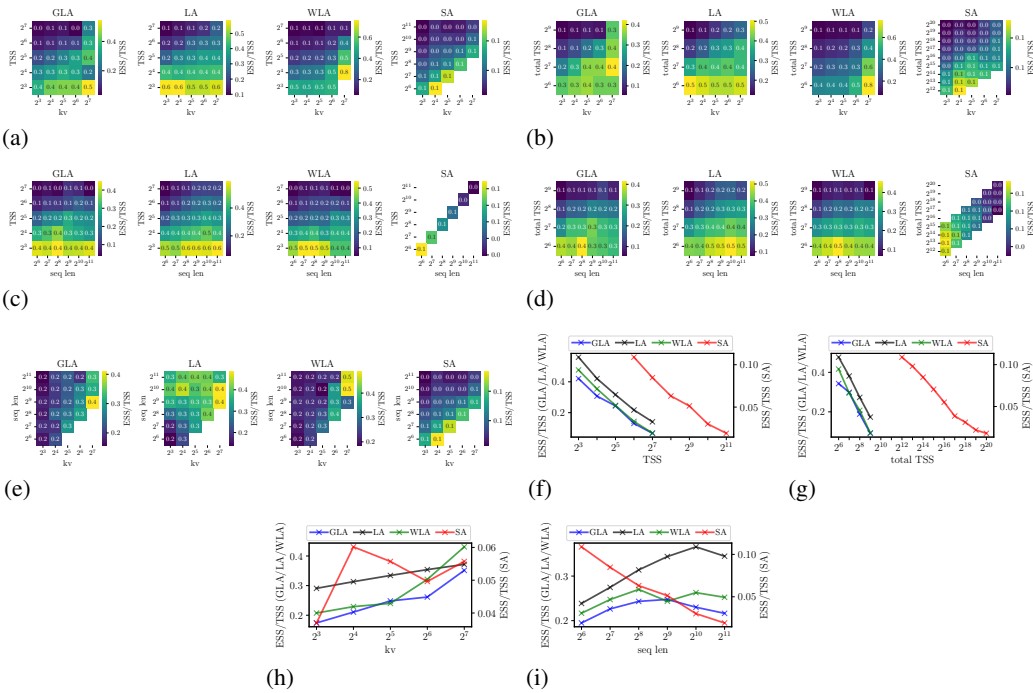

Figure 13: MQAR ESS/TSS marginalized across different dimensions.

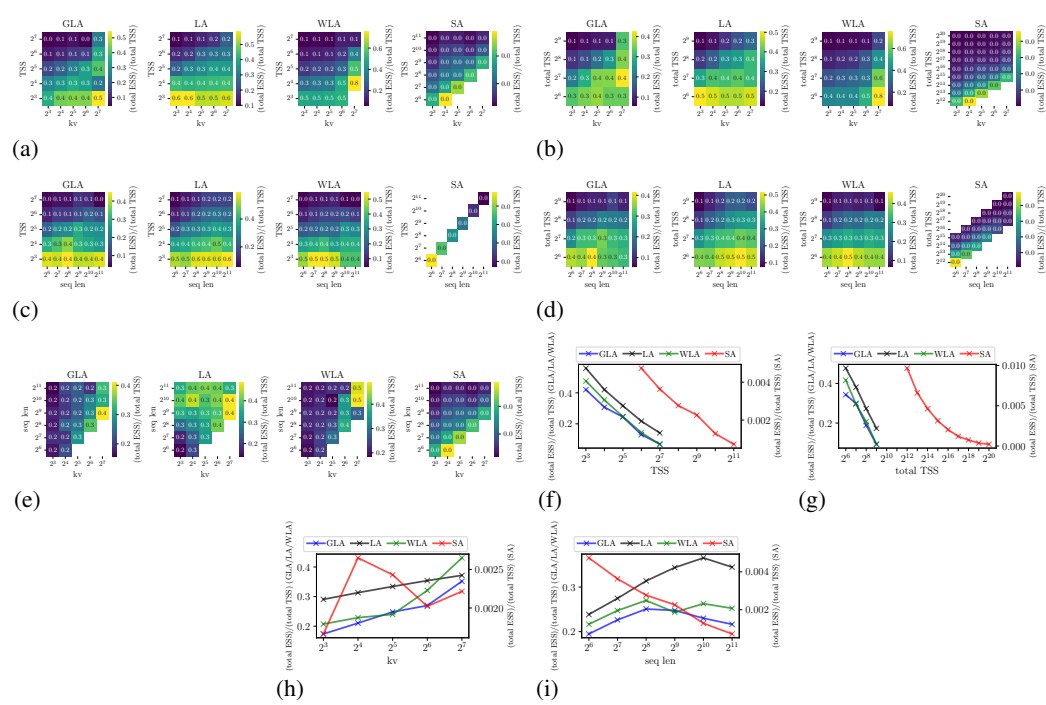

Figure 14: MQAR (total ESS)/(total TSS) marginalized across different dimensions.

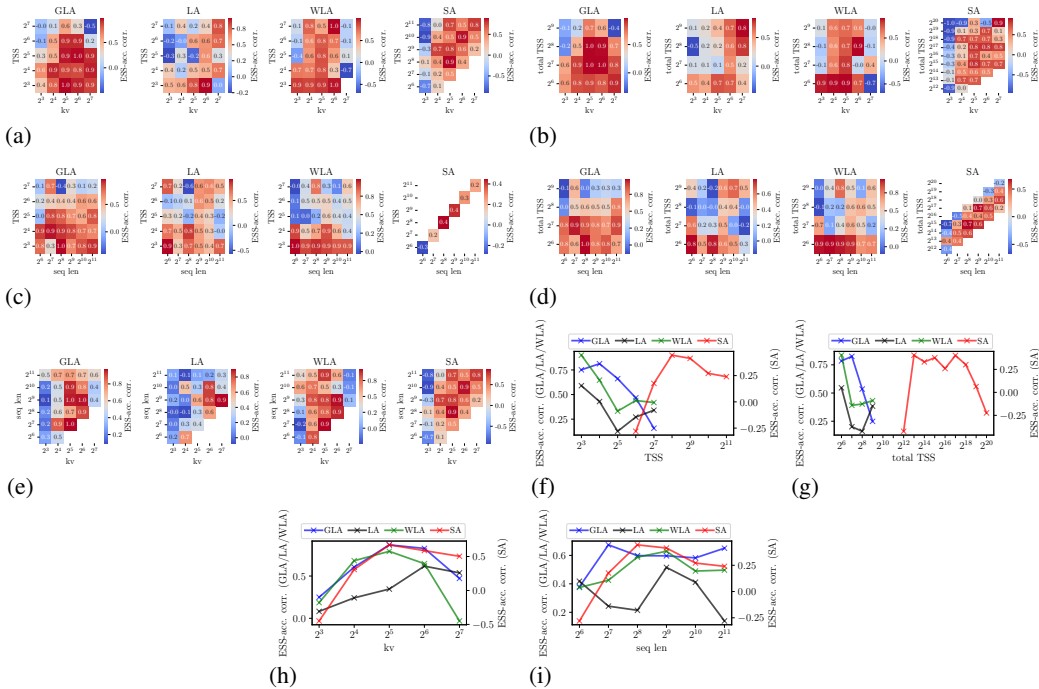

Figure 15: MQAR ESS-accuracy correlations computed over training marginalized across different dimensions.

### E.1.3 TOLERANCE-ESS MQAR RESULTS

Below are plots from the MQAR sweep using tolerance-ESS (tol=1e-3) instead of entropy-ESS. We note that all of the prevailing trends remain the same.

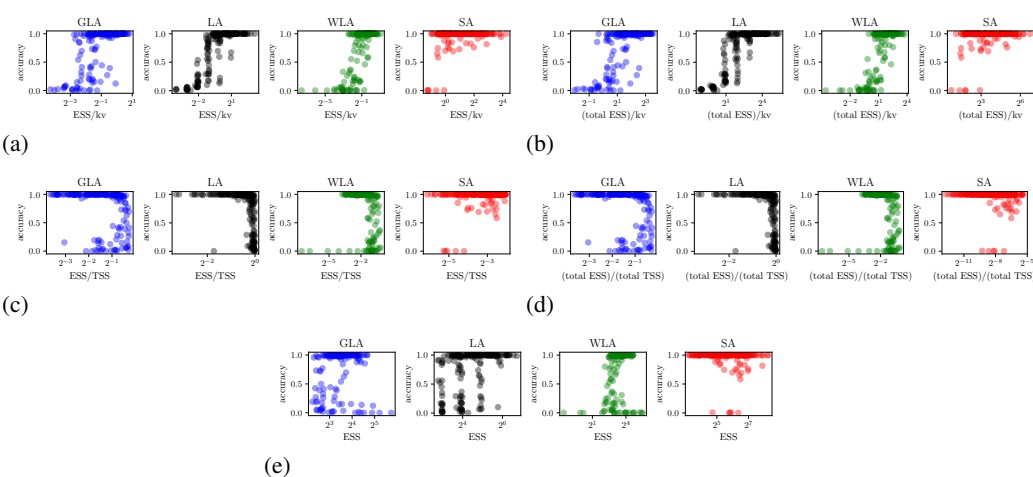

(a)

(b)

(c)

(d)

(e)

Figure 16: Accuracy vs various forms of tolerance-ESS across task-model space. Plots are entirely analogous to those shown in Figure 9.

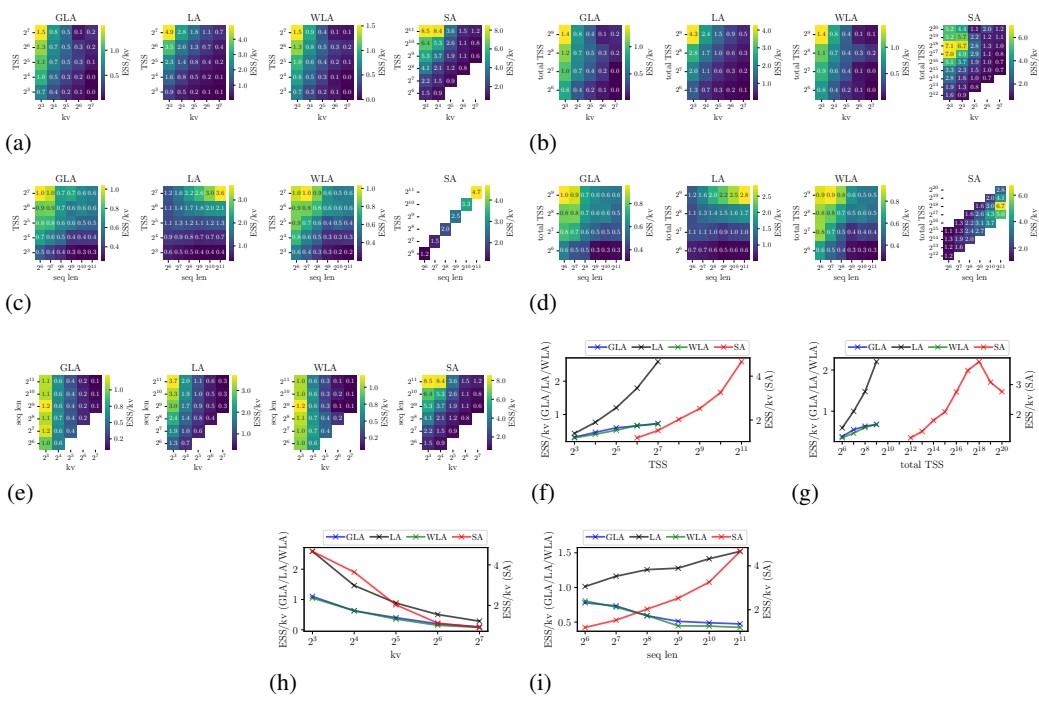

(a)

(b)

(c)

(d)

(e)

(f)

(g)

(h)

(i)

Figure 17: MQAR ESS/kv marginalized across different dimensions.

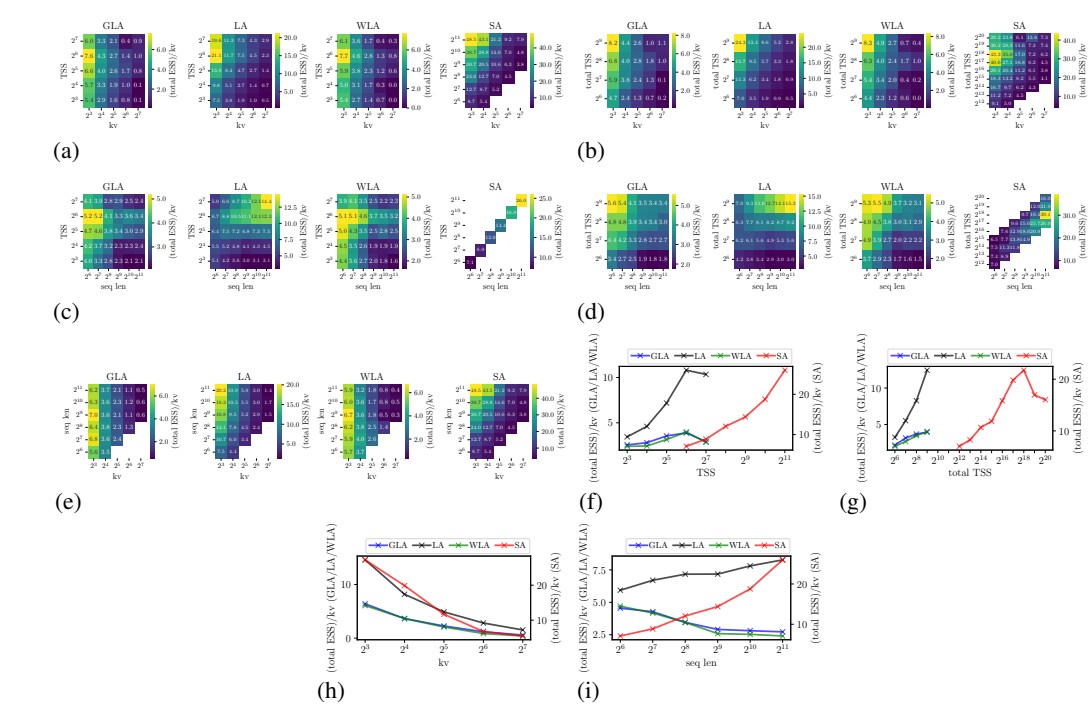

Figure 18: MQAR (total ESS)/kv marginalized across different dimensions.

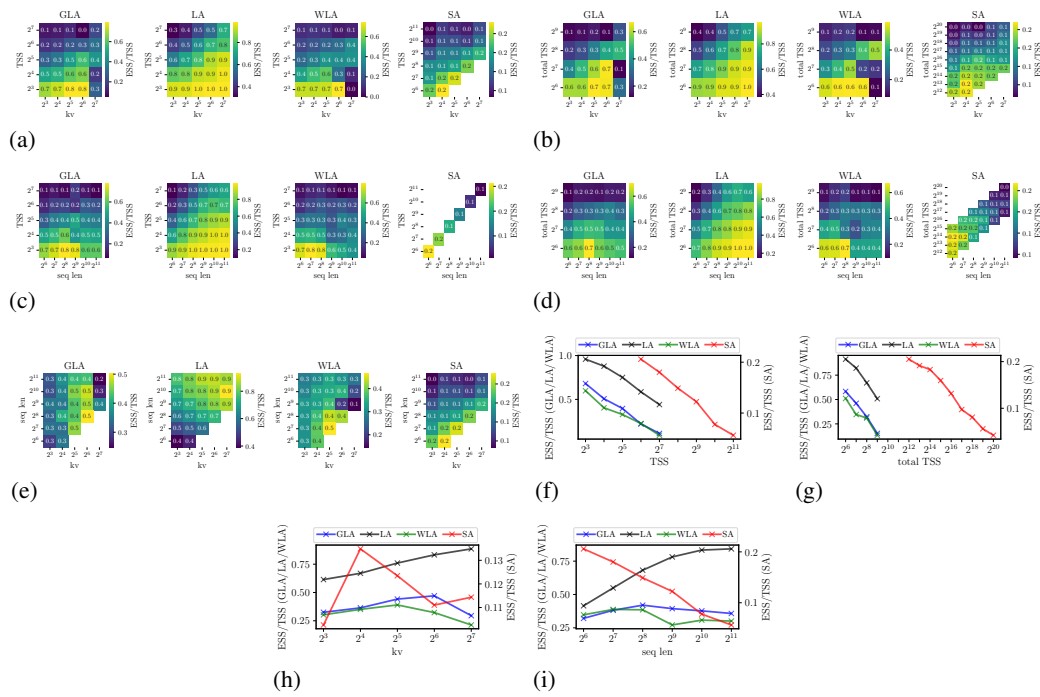

Figure 19: MQAR ESS/TSS marginalized across different dimensions.

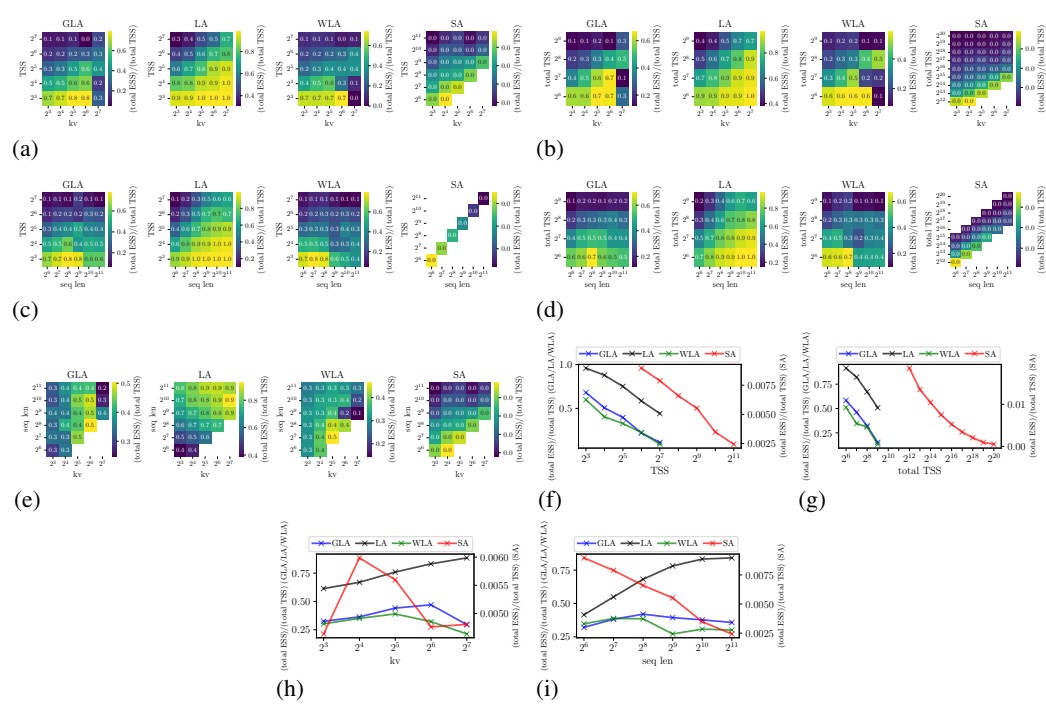

Figure 20: MQAR (total ESS)/(total TSS) marginalized across different dimensions.

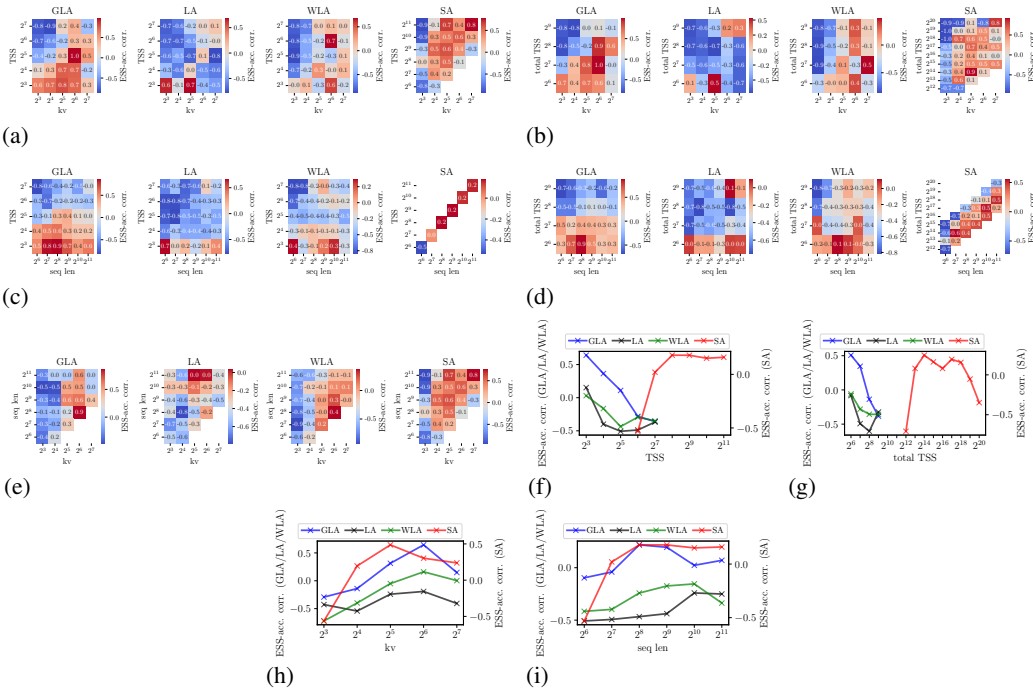

Figure 21: MQAR ESS-accuracy correlations computed over training marginalized across different dimensions.

### E.1.4 SELECTIVE COPYING AND COMPRESSION RESULTS

Below, we present results for the selective copying and compression tasks, analogous to the ones presented in Section 4 on MQAR.

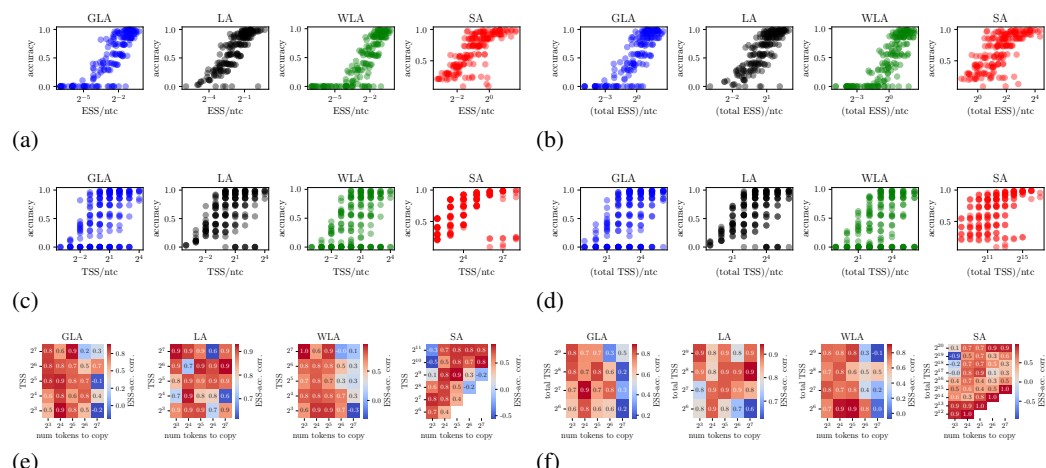

(a)                                                     (b)

(c)                                                     (d)

(e)                                                     (f)

Figure 22: Selective copying results. Note that ESS here refers to entropy-ESS and we abbreviate num. tokens to copy as ntc in plots above. (a) ESS/ntc vs accuracy across featurizers. (b) (total ESS)/ntc vs accuracy across featurizers. (c) TSS/ntc vs accuracy across featurizers. (d) (total TSS)/ntc vs accuracy across featurizers. (e) ESS-accuracy correlation computed over the course of training in (TSS, kv) buckets. (f) ESS-accuracy correlation computed over the course of training in (total TSS, kv) buckets.

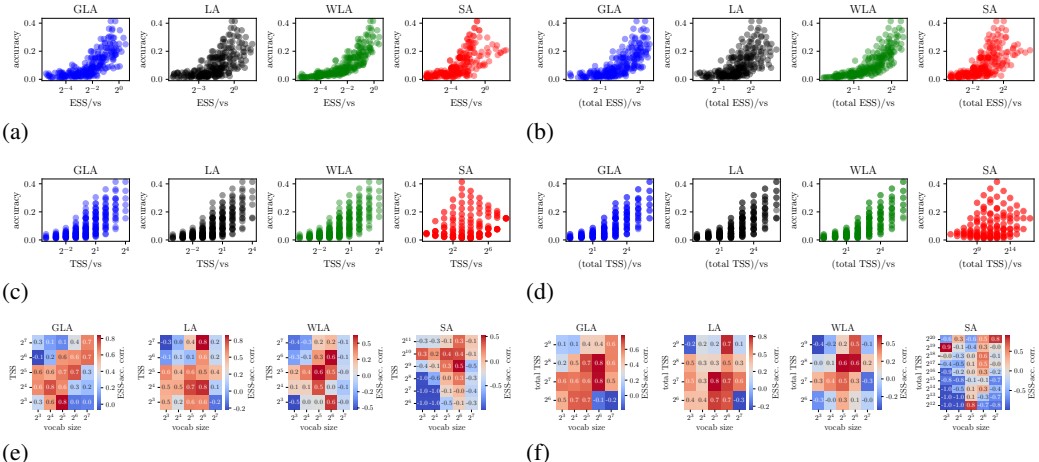

(a)                                                     (b)

(c)                                                     (d)

(e)                                                     (f)

Figure 23: Compression results. Note that ESS here refers to entropy-ESS and we abbreviate vocab size as vs in plots above. (a) ESS/vs vs accuracy across featurizers. (b) (total ESS)/vs vs accuracy across featurizers. (c) TSS/vs vs accuracy across featurizers. (d) (total TSS)/vs vs accuracy across featurizers. (e) ESS-accuracy correlation computed over the course of training in (TSS, kv) buckets. (f) ESS-accuracy correlation computed over the course of training in (total TSS, kv) buckets.

We note that with respect to the cross task-model trends, we find that in both selective copying and compression, task-adjusted ESS is a better proxy for model performance than task-adjusted TSS (Figures 22a, 22c, 23a, 23c). This is substantial as it demonstrates the utility of the ESS metric beyond just MQAR.

Regarding within task-model trends, we observe similar patterns for selective copying as those seen in MQAR (Figure 22e), with one notable distinction. Namely, ESS and accuracy are positively correlated across a larger portion of the task-model space in selective copying than in MQAR. For compression, however, the within task-model trends look a bit different from what we observe in selective copying and MQAR (Figure 23e). One potential reason for this is that the compression task is significantly more difficult than the MQAR and selective copying tasks (as noted by the

lower accuracies in Figure 23a), leading to more instabilities over the course of training. But in any case, this does highlight the fact that the strength of ESS as a proxy for model performance changes as a function of the task. The precise nature of this relationship is something we hope to explore in future work.

### E.1.5 ESS TRAINING DYNAMICS IN MQAR

As mentioned in Section D.3, we observe a phase at the start of training in MQAR in which ESS tends to decrease. This is shown in Figure 24 in which we select an arbitrary task-model configuration from the sweep and plot its ESS and accuracy over the course of training on a per featurizer basis.

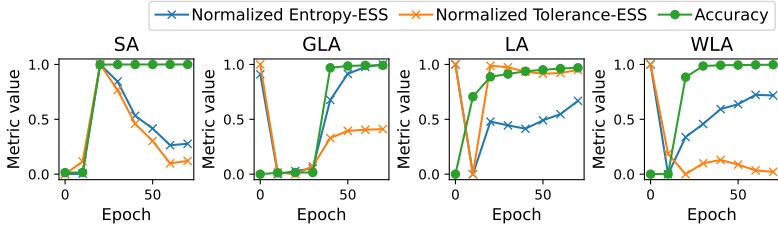

Figure 24: Training dynamics of ESS in select models (dmodel=256, heads=8) trained on MQAR (seqlen=2048, kv=64). We min-max normalize the ESS curves over the course of training to emphasize the shape of the curve as opposed to its magnitude. Note that the tolerance-ESS shown here is computed using a tolerance of 1e-3.

We find that at the start of training (i.e. in between epochs 0 and 10), even if the accuracy is not evolving, the ESS is. In particular, in the recurrent frameworks (GLA, LA and WLA), we note a sharp decrease in the ESS before it begins to rise later in training (and along with it the model accuracy). In contrast, in SA we observe the opposite: a sharp increase at the start of training followed by a steady decrease (even after it has solved the task). This points to a level of nuance in the training dynamics of MQAR ESS that we have yet to characterize and is something we hope to explore in future work.

## E.2 INITIALIZATION-PHASE ANALYSIS

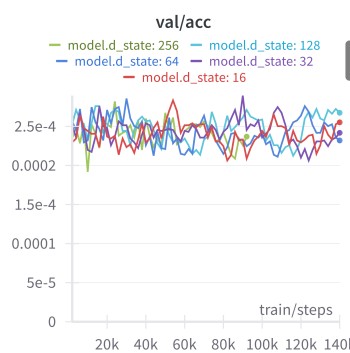

(a) Validation accuracy of S6

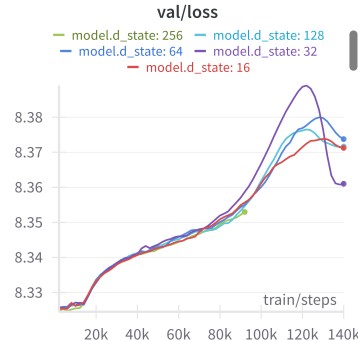

(b) Validation loss of S6

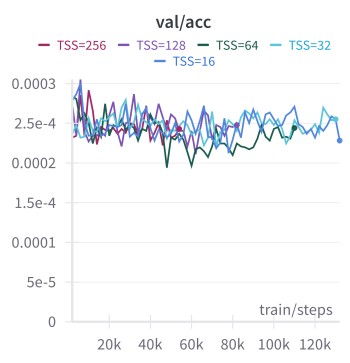

(c) Validation accuracy of GLA-S6

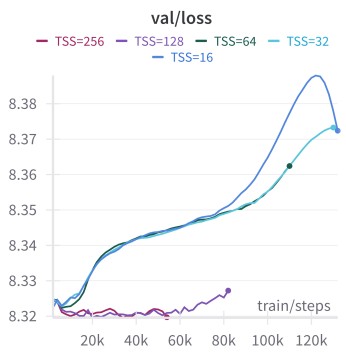

(d) Validation loss of GLA-S6

Figure 25: Loss curves of S6 and GLA-S6 showing that the models are unable to improve beyond random guessing on MQAR, across various state-sizes.

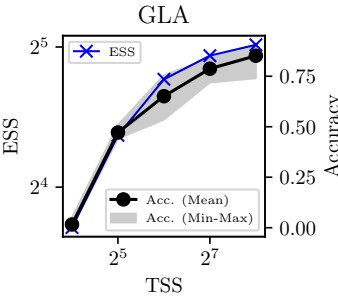

Figure 26: ESS and MQAR accuracy as a function of TSS on a custom task regime (sequence length = 1024, num. kv pairs = 256). This figure illustrates a strong correlation between MQAR accuracy, ESS and TSS.

## E.3 MID-TRAINING ANALYSIS

First, we provide some additional commentary on the ESS-based regularization results discussed in Section A. Recall we showed that decaying the $A$ matrices in GLA and WLA towards the identity matrix enables these models to outperform LA in the state collapse regime. Our intuition for this result is that by ameliorating state collapse, GLA and WLA can better leverage their increased expressivity, which stems from their learnable $A$ matrices – a feature absent from LA.

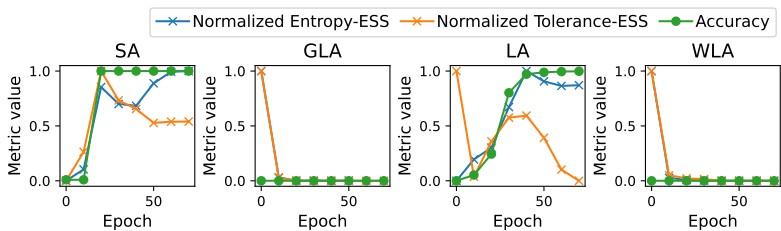

Figure 27: An example of the training dynamics of ESS in select models (dmodel=512, heads=4) trained on MQAR (seqlen=2048, kv=128) that undergo state collapse (i.e. GLA and WLA). We min-max normalize the ESS curves over the course of training to emphasize the shape of the curve as opposed to its magnitude. Note that the tolerance-ESS shown here is computed using a tolerance of 1e-3.

Next, as mentioned in Section D.5, we provide some intuition behind the efficacy of regularizing only the second layer of the network as opposed to the first or both layers.

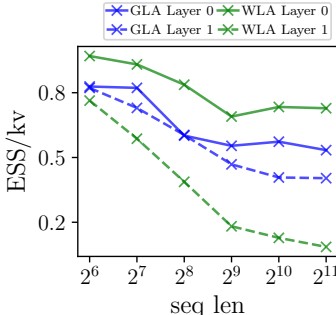

Figure 28: Per-layer ESS/kv as a function of MQAR sequence length for the GLA and WLA featurizers. ESS shown here is computed using a tolerance of 1e-3. Layers are 0-indexed.

Using 0-indexing for the layers, Figure 28 shows that layer 1 realizes a lower ESS/kv than layer 0, particularly in the case of WLA. This suggests that layer 1 contributes disproportionately to the observed state collapse (Figure 27); consequently, it makes sense that layer 1 would need to be regularized more heavily. Now, this begs the question as to why only regularizing the second layer leads to better performance than regularizing both layers (results of which were not shown). We have two possible hypotheses for this outcome. First, introducing regularization terms for both layers may complicate optimization by creating potentially conflicting objectives. Second, excessive decay of the $A$ matrices towards the identity matrix may cause the model to revert to the LA regime, which – as shown in Figure 8b – performs worse than GLA and WLA (when sufficiently regularized). Nonetheless, we hope to further explore this intuition and investigate other ESS-based forms of regularization in future work.

### E.4 POST-TRAINING ANALYSIS

#### E.4.1 MODEL-ORDER REDUCTION

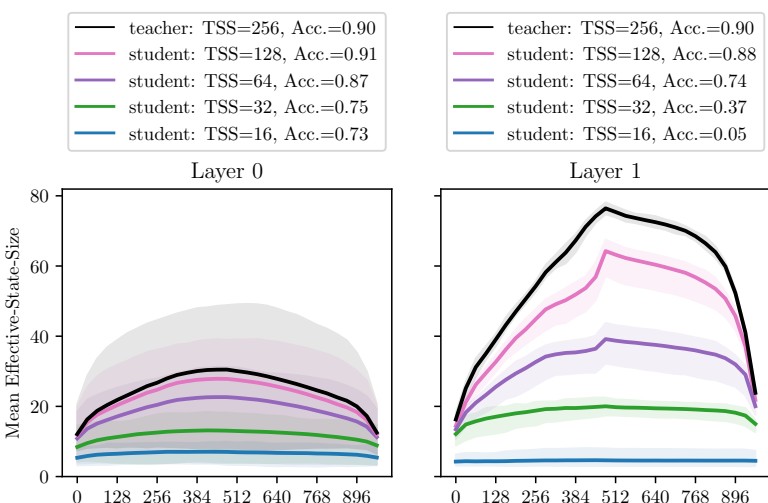

Figure 29: This figure compares MQAR accuracy and ESS across reduction scales for layers 0 and 1. The lower ESS in layer 0 of the teacher model leads to better downstream performance after distillation compared to distilling layer 1.

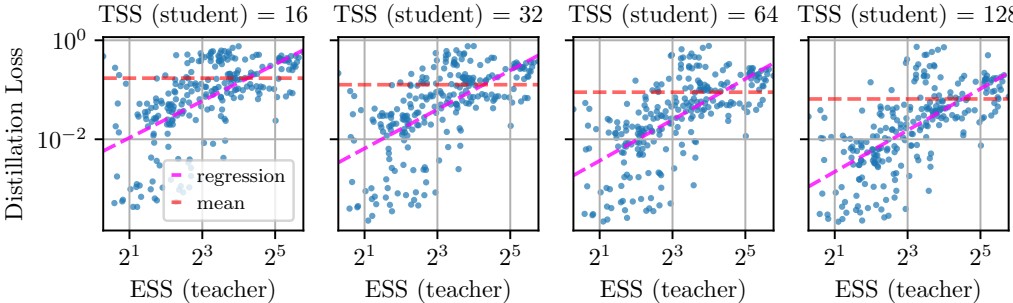

Figure 30: Correlation between ESS and distillation loss across multiple student TSSs (reduction ratios). The original teacher models have a TSS of 256.

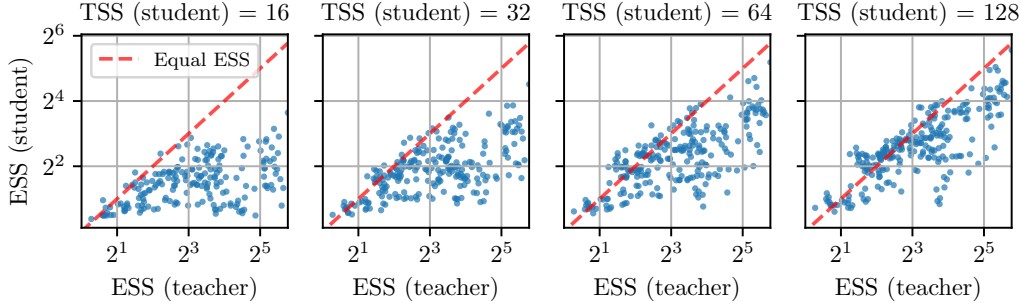

Figure 31: Teacher ESS vs distilled student ESS. As expected, we observe a clear trend: an increase in the student TSS results in the student's ESS more closely matching the teacher's ESS. Plots like these can help provide additional context during the distillation process.

## E.4.2 HYBRIDIZATION

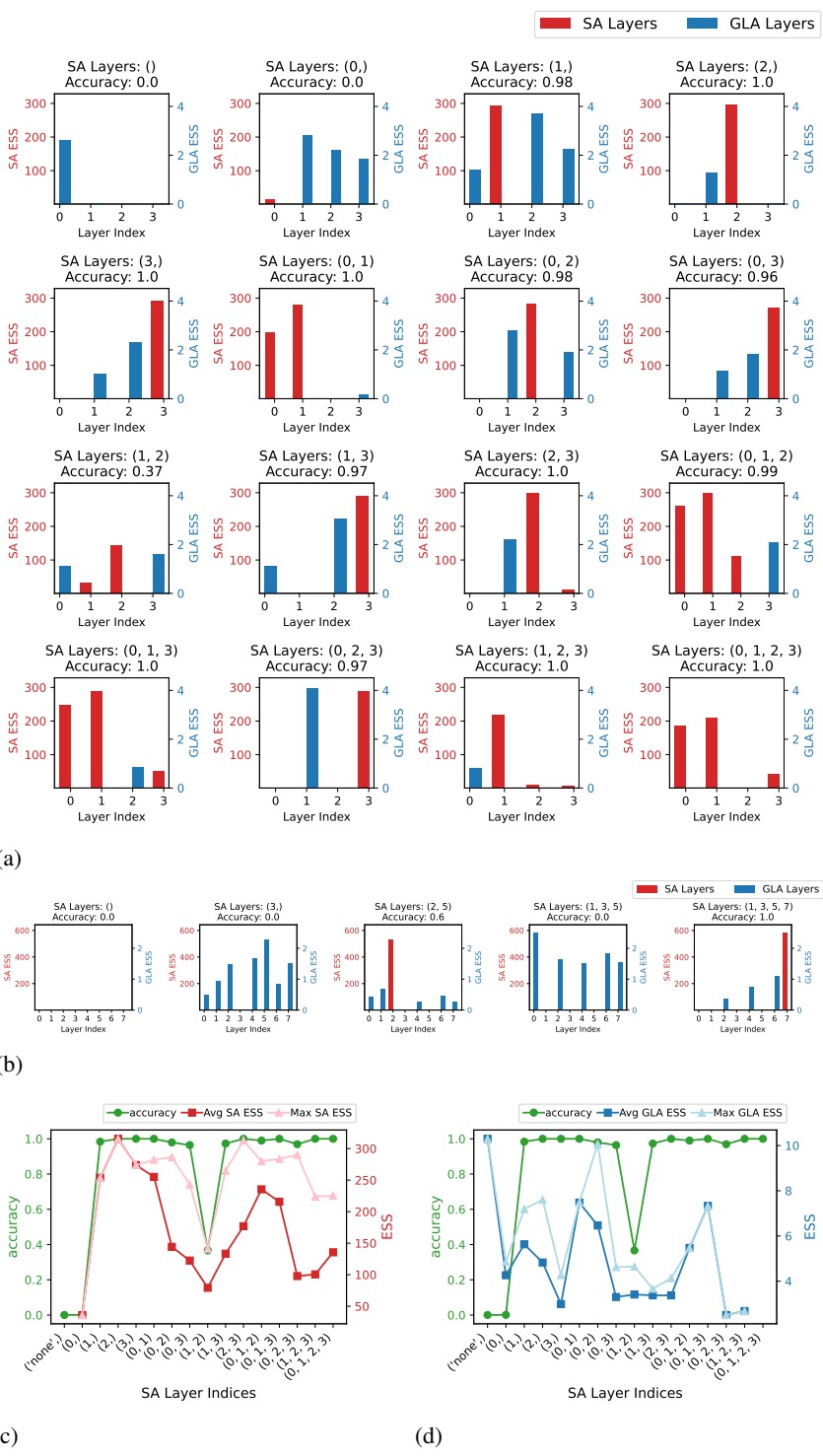

(a)

(b)

(c)                                                          (d)

Figure 32: All results presented here are computed using tolerance-based ESS with a tolerance set at 1e-1. Network layers are 0-indexed. (a) Per-layer ESS of all possible 4-layer GLA-SA hybrid networks. Experimental settings can be found in Section D.7. (b) Per-layer ESS of all possible 8-layer GLA-SA Jamba-inspired hybrid networks. Experimental settings can be found in Section D.7. (c) Model accuracy and max/average ESS of SA layers in the 4-layer GLA-SA hybrid networks. (d) Model accuracy and max/average ESS of GLA layers in the 4-layer GLA-SA hybrid networks.

Recall in Section 5.2, we demonstrated an application of post-training ESS analysis through the lens of model distillation. Here, we provide another example of post-training analysis that leverages ESS, this time in order to gain intuition into learned network dynamics – hybridization. Hybridization is the process of arranging different operators in a multi-layer sequence model (Lieber et al., 2024; Glorioso et al., 2024; De et al., 2024). Specifically, we measure the per-layer ESS across various hybrid networks and find that the precise ordering of layers significantly influences ESS dynamics, offering intuition as to why certain hybrids outperform others.

In this section, we present results from a post-training ESS analysis applied to GLA-SA hybrid networks to demonstrate the ability of ESS to capture differences among hybrid networks with varying topologies. In the first experimental setting, we train all possible 4-layer GLA-SA hybrid networks and compute the per-layer ESS on each model. We use the tolerance-based ESS since we want to analyze failure modes of learning in hybrid networks. In Figure 32a, we first note that in the pure GLA model, many of the layers fail to learn expressive states (as evidenced by the tolerance-ESS being 0), offering intuition as to why the model performs so poorly. Moving on to the hybrid networks with a single attention layer, we note that all of them perform quite well, except for the network that has attention in the first layer. Interestingly, we find that when attention is placed as the first layer, it suffers from state collapse. At a higher level, this substantiates why many state-of-the-art hybrid networks (such as Jamba) do not place attention as the first layer of the network. However, such hybrids are typically constructed purely on the basis of performance: here, ESS is able to provide a distinct perspective. Next, examining the hybrids with 2 SA layers, we find that the only poor performing topology is with attention placed in the second and third layers. Again, we find that the ESS of the attention layers is lower than what we observe in the hybrids that solve the task, indicating its usefulness as a proxy for performance beyond the 2-layer non-hybrid networks we explored in Section 4.

To clarify this, we examine the maximum/average ESS (computed across layers) of the SA and GLA layers separately to understand how each relates to model performance. Notably, we find that maximum ESS across attention layers best correlates with accuracy (Figure 32c). Interestingly, the average SA layer ESS is a worse proxy for performance, potentially indicating that having a single layer with high memory utilization in hybrid networks is more important than having many layers with lower memory utilization. This offers support as to why hybrid networks like Jamba have a 1:7 ratio between attention and non-attention layers. Regarding the GLA layers, we find that despite both the maximum and average SS varying across models, they do not correspond to changes in accuracy. One possible explanation for this is that since the attention layers are responsible for driving the total ESS of the network up due to their unbounded state size, the role of non-attention layers in hybrid networks may not be captured entirely by the magnitude of their ESS. Nonetheless, this is something we hope to explore in future work.

In the second experimental setting, we move beyond 4-layer GLA-SA hybrids to 8-layer GLA-SA hybrids. Here, instead of iterating over all possible topologies, we restrict the space of networks to those constructed via the hybridization policy proposed by Jamba. The Jamba hybridization policy takes in the number of layers as input and provides a particular hybrid topology as output (refer to Lieber et al. (2024) for more details). Since most topologies explored in the 4-layer setting solved the task, we both reduce the model dimension of the network and make the task more difficult to see if we can observe performance differences across the architectures (model settings can be found in Table 8). Unsurprisingly, we find that the pure GLA network is unable to solve the task and also realizes a tolerance-based ESS of 0 in all layers (Figure 32b). However, more interesting is the fact that while the 2 SA-layer Jamba hybrid partially learns the task, the 3 SA-layer does not. Examining the ESS shows that the attention layers in the 3 SA-layer hybrid suffer from state collapse, which we know is highly correlated with poor performance on MQAR. This points to a deficiency of fixed-topology hybridization policies like Jamba which do not take into account factors like network trainability which can significantly influence model performance. Furthermore, this suggests that the ESS metric can be used to better inform the construction of hybrid networks. We hope to further elucidate these per-layer ESS trends and leverage these insights to construct novel ESS-informed hybridization policies in future work.

### E.5 STATE MODULATION OF LARGE LANGUAGE MODELS

State modulation patterns on various open-weight models are illustrated in Figures 33, 34, 35, 36, 37, and 38.

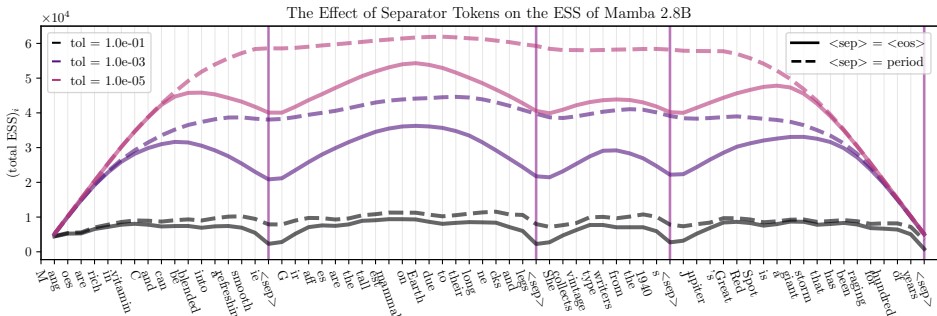

Figure 33

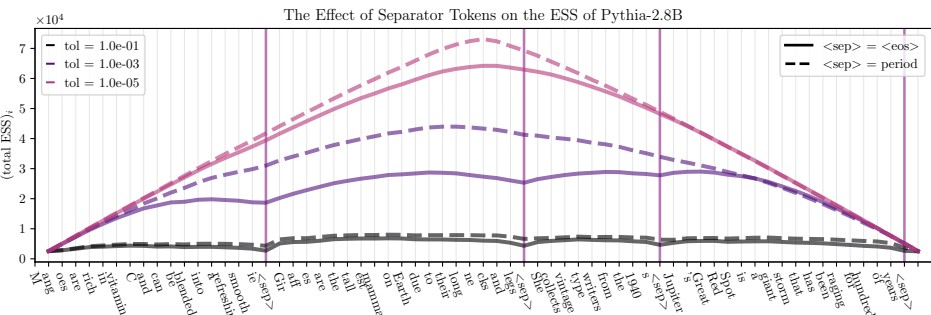

Figure 34

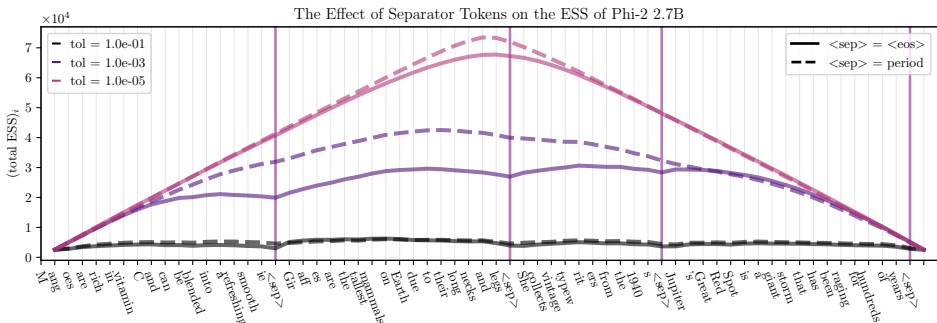

Figure 35

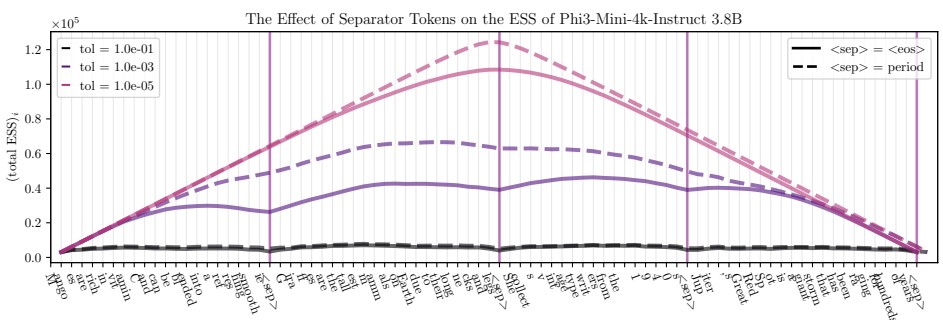

Figure 36

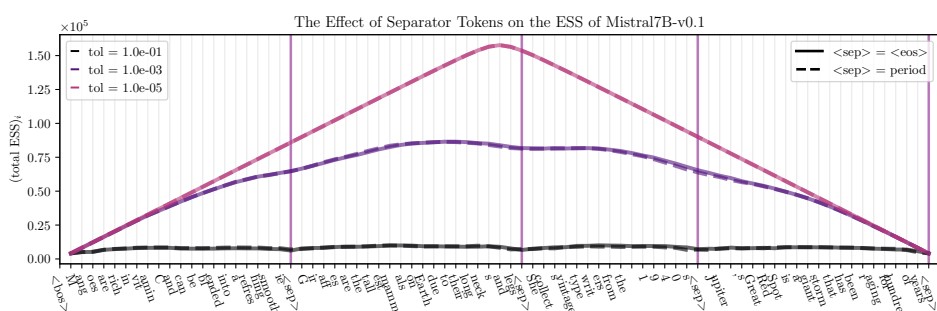

Figure 37

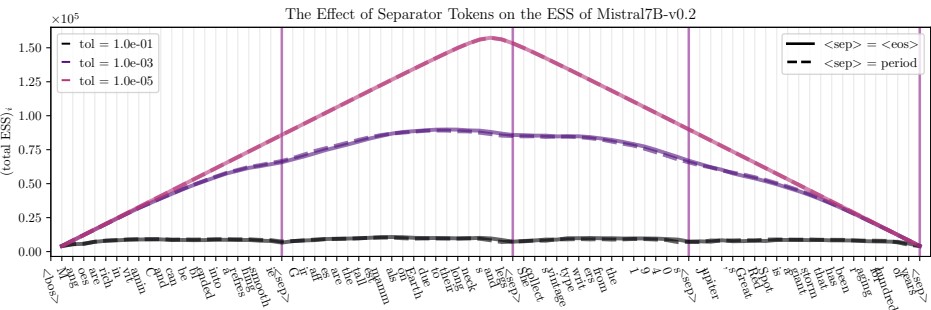

Figure 38

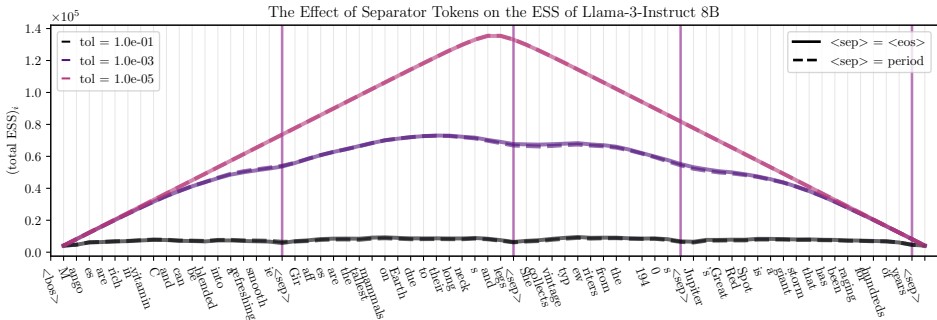

Figure 39

The figures above reveal significant cross-architectural differences in context processing. The attention-based model at a similar 7B scale (Figure 37, 38, and 39) shows minimal change in its

ESS pattern when an EOS token is replaced with a period (".") . In contrast, the limited cache-size state-space model (Falcon Mamba 7B, Figure 7a) exhibits a substantial reduction in state modulation under the same token substitution.

We attribute this difference to a phenomenon we term "preemptive state modulation" in limited state-size models, which stems from fundamental architectural differences. State-space models (SSMs) with limited cache must efficiently manage their finite memory capacity and learn to preemptively modulate state-size to optimize information retention, relying on explicit signals like EOS tokens to trigger context resets. In contrast, attention models with linearly increasing cache can store all past information without the need for selective forgetting, do not require preemptive state modulation, and show less sensitivity to explicit demarcation tokens. This distinction highlights the different strategies employed by various model architectures in managing context across diverse inputs, potentially influencing their performance on tasks requiring long-range recall or context separation.

However, a subset of attention models demonstrated varying state modulation patterns in response to different separator tokens, with this effect being more pronounced in smaller model sizes (see Figure 34, 35, and 36). This phenomenon, while not consistent across all attention architectures, merits deeper exploration.

Figure 40 illustrates the state modulation patterns at different tolerance levels for the four 1B language models (LA, WLA, GLA, SA), trained under identical conditions.

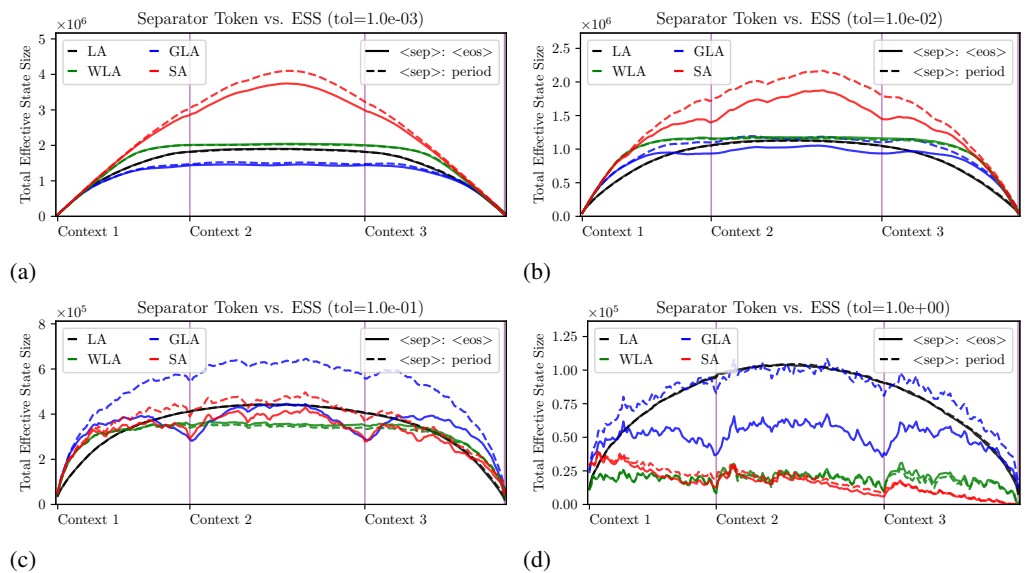

(a)          (b)

(c)          (d)

Figure 40: An illustration of the effect of different separator tokens over different layers across different tolerances. Softmax attention exhibits the most pronounced state modulation, beginning at a tolerance level of $1e-2$, followed by gated linear attention with significant modulation starting at a tolerance of $1e-1$. Weighted linear attention shows minimal modulation, only detectable at a tolerance of $1.0$, while linear attention displays no discernible separator token-induced state modulation. Here ESS is summed across channels and layers.

Notably, GLA exhibits a substantial variation in state modulation depending on the separator token, consistent with our earlier observations in Falcon Mamba regarding preemptive state modulation. In contrast, SA shows a smaller, yet non-trivial, effect. WLA and LA show no discernible differences across separator tokens, which may be attributed to their overall limited ability to modulate state size.

2862
2863
2864
2865
2866
2867
2868
2869
2870
2871
2872
2873
2874
2875
2876
2877
2878
2879
2880
2881
2882
2883
2884
2885
2886
2887
2888
2889
2890
2891
2892
2893
2894
2895
2896
2897
2898
2899
2900
2901
2902
2903
2904
2905
2906
2907
2908
2909
2910
2911
2912
2913
2914
2915

### E.6 MISCELLANEOUS

#### E.6.1 EFFECTIVE STATE-SIZE ON C++ CODE

Beyond sentence delimiters such as periods and end-of-speech tokens (discussed in Section 5.3), we observe similar "dips" in effective state-size where there are scope delimiter tokens such as "}".

The following plots demonstrate the ESS pattern of Llama3-8B processing the C++ code of a fast inverse square root algorithm and a Fibonacci sequence generator algorithm.

**Quake fast inverse square-root algorithm:**

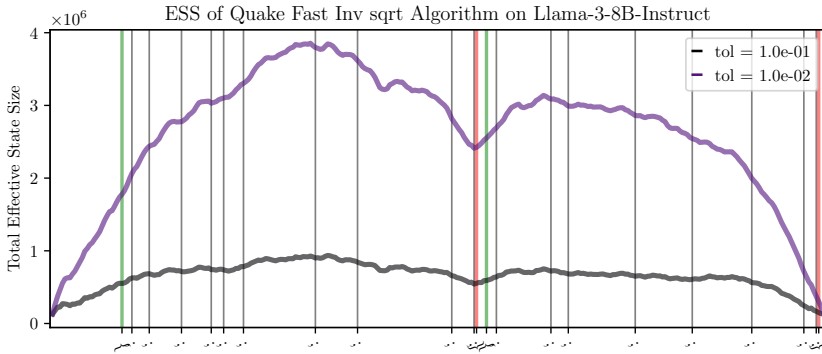

Figure 41: Effective state-size over a quake fast inverse square root algorithm's code. Here ESS is summed across channels and layers.

```cpp
1  #include <iostream>
2  #include <cmath>
3
4  // Quake Fast Inverse Square Root function
5  float quakeFastInvSqrt(float number) {
6      long i;
7      float x2, y;
8      const float threehalfs = 1.5F;
9
10     x2 = number * 0.5F;
11     y = number;
12     i = *(long*)&y;                 // Bit-level hacking: convert float to
       long
13     i = 0x5f3759df - (i >> 1);   // Initial magic number and bit shift
14     y = *(float*)&i;                // Convert back from long to float
15
16     // Newton's method step for refining the result
17     y = y * (threehalfs - (x2 * y * y));   // First iteration
18
19     return y;
20 }
21
22 int main() {
23     float number;
24
25     // Input: Get the number from the user
26     std::cout << "Enter a number: ";
27     std::cin >> number;
28
29     // Output: Display the result using the Quake fast inverse sqrt
30     float quake_result = quakeFastInvSqrt(number);
31     std::cout << "Quake Fast Inverse Sqrt: " << quake_result << std::endl
       ;
32
33     // Compare with standard sqrt function
```

```
34    float std_result = 1.0f / std::sqrt(number);
35    std::cout << "Standard Inverse Sqrt: " << std_result << std::endl;
36
37    return 0;
38 }
```

**Fibonacci sequence generating algorithm:**

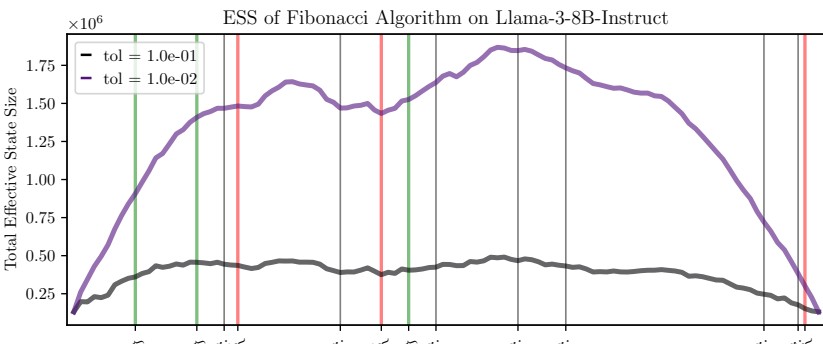

Figure 42: Effective state-size over a Fibonacci sequence generator algorithm's code. Here ESS is summed across channels and layers.

```
1 #include <iostream>
2 int fibonacci(int n) {
3     if (n <= 1) {
4         return n;
5     }
6     return fibonacci(n - 1) + fibonacci(n - 2);  // Recursive case
7 }
8 int main() {
9     int n;
10    std::cout << "Enter a positive integer: ";
11    std::cin >> n;
12    std::cout << "Fibonacci number at position " << n << " is: " <<
      fibonacci(n) << std::endl;
13    return 0;
14 }
```

### E.6.2 HOW THE NUMBER OF PROMPTING SHOTS AFFECTS THE EFFECTIVE STATE-SIZE OF LANGUAGE MODELS

Here, we explore how varying the number of shots when prompting large language models affects their effective state-size patterns. We use Phi-2 as the candidate attention model and Mamba-2.8B as the state-space model. The task we tested this on is MMLU (elementary mathematics).

At the start of the Q&A section for the attention model, there is a noticeable difference in state size between 0-shot and 1-shot prompts. Beyond 1-shot, the difference in ESS appears minimal. For the state-space model, varying the number of shots has minimal impact on the effective state-size.

Although these sparse experimental results require further investigation, we note the stark difference in the effective state-size patterns between these two architectures, which provides additional insights into understanding the fundamental differences in the way prompts are processed across models.

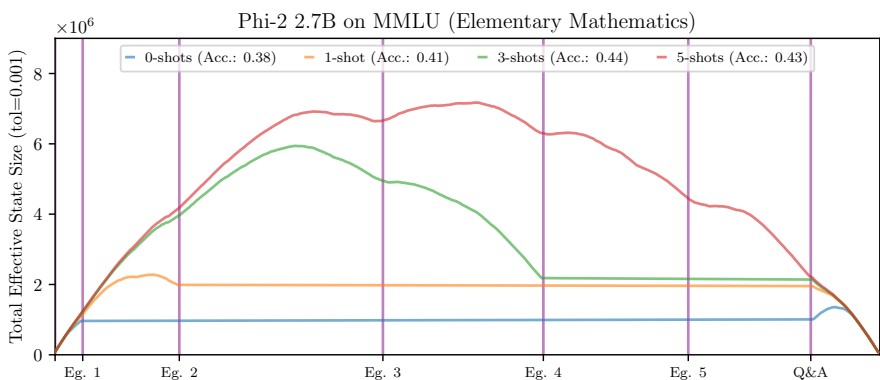

Figure 43: The variation in effective state-size with a varying number of shots (2.7B Attention). Here ESS is summed across channels and layers.

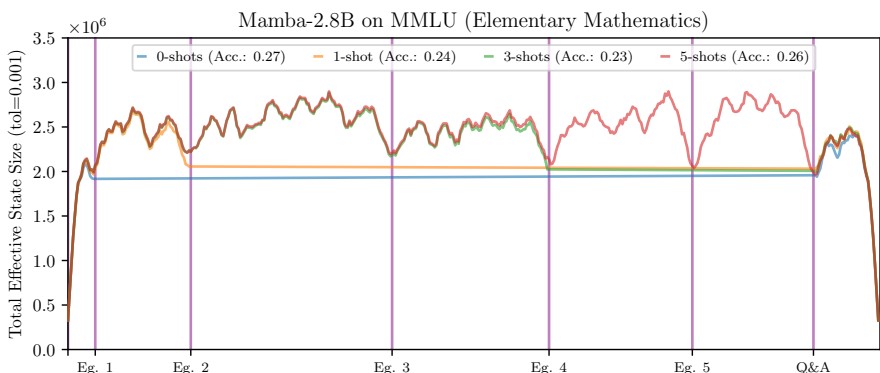

Figure 44: The variation in effective state-size with a varying number of shots (2.8B State-Space Model). Here ESS is summed across channels and layers.

