# OpenReview forum: "Quantifying Memory Utilization with Effective State-Size"
_ICLR.cc/2025/Conference — Submitted to ICLR 2025_

### Official Review · Reviewer_77Pw · 2024-11-04

**Soundness:** 2
**Presentation:** 2
**Contribution:** 2
**Rating:** 5
**Confidence:** 3

**Summary:**

The authors introduce a metric called "effective state size (ESS)" to capture the memory utilization of sequence modeling architectures. They compare the correlation between ESS and accuracy across various tasks and architectures, finding that ESS shows a stronger correlation with performance than the "theoretically realizable state size" in the SSM equations. They use ESS to guide initialization strategies and regularization methods. They also observe that ESS dynamically changes in response to inputs, particularly at EOS tokens, noting that this modulation effect is more stronger in more performant architectures.

**Strengths:**

- Identifying quantitative measures to compare different language modeling architectures is an important and timely problem.
- The framework proposed by the authors is broadly applicable to many sequence modeling architectures.
- The idea of the ESS is natural and connected to classical ideas in control theory.
- The authors’ observations on the phenomenon of "state modulation" are interesting from a linguistic perspective.

**Weaknesses:**

* While the concept of ESS is relatively straightforward, the paper is very difficult to read since it constantly refers to the appendix (which is 30+ pages long) for definitions and empirical results.
* Notation and plots are often not explained in the main text (e.g., the meaning of "total ESS").
* The core concept of ESS is not clearly defined; although the authors present it as an architectural property, it also depends on the input (as noted in Sec 4.2), so it should vary with both data and task.
* Key terms like "state saturation" and "state collapse" are introduced only informally.
* Although perhaps novel in this context, the basic idea of ESS is not new; for instance, it is closely related to the "semiseparable rank" in the Mamba2 paper.
* Beyond Finding 1, the other empirical findings seem to be very narrow and not especially useful or insightful.

**Questions:**

* Can the authors provide a clear definition of ESS, explaining whether it is a function of the model or the data, and describe how it is computed in practice? The two approaches introduced in Section 2.4 do not seem to be ever discussed in the paper again.
* Similarly, how is the TSS different from the conventional notion of state size?
* Minor: Why use the term "input-varying _linear_ operator"? Input dependence typically renders the operator non-linear, and I believe that any differentiable map $y=F(u)$ satisfying $F(0)=0$ can be expressed in the form $y = M_F(u) u$

---

> ### Author Response · Authors · 2024-11-20
> **Response to reviewer 77Pw**
>
> Thank you for the thorough review of our work. The weaknesses you have pointed out are entirely valid, and we have revised the manuscript to address them.
>
> Firstly, to improve the general readability, we clarified a few key concepts in the main text and reduced the density of the paper as follows. Please refer to the meta-review addressed to all reviewers for a high-level description of the structural modifications we made. To address the first two weaknesses you brought up, we made the following revisions in particular:
> * Streamlined the theory section:
>   - Removed unnecessary (but related) equations and discussion, thereby reducing references to the appendix
>   - More clearly defined ESS as a metric, clarifying that unlike TSS it is a function of both the model and input data
>   - Contextualized ESS against the concept of semiseparable rank presented in Mamba2
>   - Discussed TSS in greater detail
>   - Added a section clarifying the distinction between average and total ESS and TSS
>
> * Streamlined the introductory section to more clearly emphasize the contributions of our work
>
> Beyond these general improvements, we would like to address specific weaknesses and questions raised by the reviewer:
>
> > 1. The core concept of ESS is not clearly defined; although the authors present it as an architectural property, it also depends on the input (as noted in Sec 4.2), so it should vary with both data and task. Can the authors provide a clear definition of ESS, explaining whether it is a function of the model or the data, and describe how it is computed in practice? The two approaches introduced in Section 2.4 do not seem to be ever discussed in the paper again.
>
> Thank you for your insightful feedback. We agree that ESS reflects properties of both the architecture and the data, which allows us to investigate cross-architectural differences by applying the same input across different models (note that this is lacking from TSS). This approach also enables us to analyze ESS at a per-token level, as demonstrated in our state-modulation experiments. We've added a paragraph to clarify this in Section 3.1.
>
> Regarding how ESS is computed, we've added a paragraph to the paper titled "Average vs total ESS" which details how ESS is computed in practice. Additionally, we've added a paragraph titled "Computational complexity of ESS" which provides insight into how the time complexity of computing the ESS metrics scales as a function of the data and model. Both of these can be found in Section 3.1.
>
> > 2. Key terms like "state saturation" and "state collapse" are introduced only informally.
>
> While it is true that “state saturation” and “state collapse” are introduced informally, we aim to use those concepts similarly to a concept like “attention sink” [1], where the concepts help point to a phenomenon that recurs and has useful fixes (like through regularization as shown in the mid-training experiments).
>
> > 3. Although perhaps novel in this context, the basic idea of ESS is not new; for instance, it is closely related to the "semiseparable rank" in the Mamba2 paper.
>
> Indeed, ESS is connected to 'semiseparable rank,' which we observe can be conceptualized as the max(ESS) across sequence length. However, as presented in previous work [2, 3], semiseparable rank has been applied in a way that is more aligned with the notion of TSS to facilitate efficient algorithm design. This means that prior notions of semiseparable rank do not capture anything beyond model structure (i.e. input data, optimization, etc.). In this study, we provide complete derivations of ESS, clearly distinguish it from TSS, and demonstrate its effectiveness as a proxy for memory utilization in practical settings for modern LIV sequence models. This, in our view, is novel and is the core contribution of our work. Furthermore, applying the ideas of effective rank to produce the entropy-ESS metric has, to the best of our knowledge, never been done before.
>
>
> [1] Guangxuan Xiao., et al, "Efficient Streaming Language Models with Attention Sinks," 2024.
>
> [2] Tri Dao. Albert Gu, "Transformers are SSMs: Generalized Models and Efficient Algorithms Through Structured State Space Duality," 2024.
>
> [3] R. Vandebril, M. Van Barel, N. Mastronardi. "A note on the representation and definition of semiseparable matrices," in Numerical Linear Algebra with Applications, vol. 12, no. 8, pp. 839-858, 2005.

---

> ### Author Response · Authors · 2024-11-20
> **Response to reviewer 77Pw continued**
>
> > 4. Beyond Finding 1, the other empirical findings seem to be very narrow and not especially useful or insightful.
>
> With respect to your point about our findings being narrow, we note that in our work we aimed for breadth and not depth in the analyses we conducted, since the proposition of the ESS metric is itself novel. In particular, we aimed to showcase the potential for application across a wide domain and hope this inspires future, more in depth, work in some of the directions we have examined.
>
> With respect to your point about our findings not being especially useful or insightful, we tend to disagree. While we agree that our findings on initialization and regularization have been explored before and ESS is simply providing a different lens as to the efficacy of these approaches, our findings on distillation and hybridization, to the extent of our knowledge, are entirely novel in that ESS provides perspectives on these two fields that was previously not known. Furthermore, in our extension to language, we note that our observation of state-modulation covers a large model space which sheds new light as to why certain models perform better on recall which is a very popular topic in the current age of machine learning.
>
> > 5. Similarly, how is the TSS different from the conventional notion of state size?
>
> TSS corresponds to the conventional concept of state size; however, we use the term 'theoretically realizable state-size' to emphasize its role as an upper bound for the effective state-size metric. This distinction highlights its interpretation as the model's memory capacity. Moreover, we also wanted to emphasize the fact that TSS is the state-size in the form written in Equation (1), and is related but not the same as implementation specific quantities like “cache-size”, which is often used analogously with “state-size”.
>
> > 6. Minor: Why use the term "input-varying linear operator"? Input dependence typically renders the operator non-linear
>
> We originally used the term 'operator' to refer to both the operator 'T' and the model as a whole. To clarify, we now use 'system' for the latter. Thus, 'input-varying linear operators' are now described as 'systems with input-varying linear operators.'  and we now abbreviate this as LIVs to conform to classical conventions. This revision emphasizes that the models are 'systems' containing linear operators (i.e. matrix T) that vary with the input.
>
> > 7. Minor: I believe that any differentiable map $y = F(u)$ satisfying $F(0) = 0$ can be expressed in the form $y = M_F (u) u$
>
> While the framework is indeed broad, it is especially valuable for modern sequence models, most of which include a 'featurization' phase. In this phase, features like $A, B, C, D$ matrices for SSMs and $Q, K$ features for attention-like models are computed. After featurization, as noted in the paper, the model essentially becomes a simple linear system, allowing for the application of various efficient algorithms—particularly when the linear operator has known structures like semiseparable, Toeplitz, or low-rank.
>
> We thank the reviewer again for their thoughtful review of our work! We hope that our revisions have improved clarity on the definition and formulation of the ESS metric and that our answers to your questions have resolved any outstanding concerns you may have.

---

> > ### Comment · Reviewer_77Pw · 2024-11-24
> >
> > Thank you for your response. I appreciate the efforts to streamline the paper. However, I remain concerned by the absence of a single, clear formulation/definition for the ESS metric, which I believe significantly limits its usefulness (the authors switch between different ways of computing it, and different ways of aggregating across layers, channels, and timesteps). The limited novelty of the ideas (which other reviewers also noted), also remains a concern. For these reasons, I maintain my score.

---

> ### Author Response · Authors · 2024-11-25
> **Response to reviewer 77Pw**
>
> We thank the reviewer for their response and respond to the points raised below:
>
> Regarding the lack of a clear definition for the ESS metric, we tend to disagree. In the paper (line 180), there is a single, clear definition presented: ESS = rank($H_i$). The reason there are two ways of computing this rank is consistent with how rank is computed numerically in practice. Namely, the two canonical approaches are numerical rank (which corresponds to tolerance ESS) and effective rank (which corresponds to entropy ESS). There are different ways of aggregating it, simply due to the shape of the data and model. This is no different from other interpretability metrics like attention maps, which also have to choose which model/data dimensions to marginalize across to make the analysis more tractable. In fact, given the quantitative nature of the ESS metric, it is more concise in its presentation of information than a qualitative visualization of an attention map.
>
> Regarding the limited novelty of the idea, while we agree that there are ideas presented in our work that are similar to those found in previous works like Mamba-2, we maintain that the ESS metric in the context of LIV systems is novel. In particular, there are no prior works that apply minimal realization theory to input-varying linear operators (LIVs) which forms the basis for the ESS metric. LIVs are of great practical interest in deep learning, since many sequence models can be formulated under this framework. Furthermore, our results go on to demonstrate that the ESS metric is more correlated with performance on recall tasks than TSS is, which demonstrates the usefulness of ESS beyond its interpretability. Regarding the note on other reviewers citing lack of novelty, we note that only reviewer 8oZA made such comments and the main issue brought up here was with respect to our initial presentation of theory which suggested that the proofs we used from existing results on linear systems were novel, when in fact they are not. We have reworded the theory section to clarify this misconception.
>
> We thank the reviewer again for their time and hope they will reconsider their score in lieu of our above responses.

---

> ### Comment · Reviewer_77Pw · 2024-11-25
>
> Thank you for your response. I respectfully disagree that simply writing $ESS = {\rm rank}(H_i)$ is a clear definition. For example, this expression hides what ESS depends on as a function, and writing something like $ESS(x)_i$ would be better to highlight dependence on the input and timestep. Incidentally, the term "state size" may be somewhat misleading, since typically the size of the state -- defined as a summary statistic of the entire past -- should either be constant or increase over time.
>
> Appendix D.1 states: "unless stated otherwise, our figures are presented using the entropy-ESS. In cases where we require ESS comparison across the sequence dimension, we instead plot ESS for multiple tolerance values." While flexibility in variations and aggregation strategies can be appropriate for interpretability tools aimed at informal / qualitative explorations (like attention maps), a paper proposing a new metric should in my view be more precise.
>
> I would encourage the authors to consider either formalizing their treatment of ESS, or reframing their work as an exploratory analysis of the inner workings of sequence modeling architectures, without emphasizing the introduction of a quantitative metric.

---

> ### Author Response · Authors · 2024-11-28
> **Response to reviewer 77Pw (1/2)**
>
> We thank the reviewer for their insightful response, and agree with most of the points raised. As such, we have made significant revisions to the manuscript, which are detailed below.
>
> > 1. I respectfully disagree that simply writing ESS=rank(Hi) is a clear definition. For example, this expression hides what ESS depends on as a function, and writing something like ESS(x)i would be better to highlight dependence on the input and timestep.
>
> Thank you for clarifying what you meant by this. We agree that only writing ESS = rank($H_i$) is not a clear definition without appropriate elaboration. In lieu of this, in our revision, we clarify that for LIV systems (which are the models we consider in this paper) ESS is input dependent (for example in line 184 of the paper). Additionally, we revised the paper to clarify that rank($H_i$) computes the state-size for all $d$ channels, but for SISO models in particular (where we can perform a per-channel computation of ESS), we can view ESS as a per-channel metric. Initially, we made this logical leap without spelling it out.
>
> However, since the general formulation of ESS applies to both input-varying and input-invariant models, we still use ESS = rank($H_i$). When specifically discussing the multi-layer LIV SISO models used in this paper, we explicitly formulate ESS as a function of the input, sequence index, channel, and layer number as requested by the reviewer.
>
> > 2.  Incidentally, the term "state size" may be somewhat misleading, since typically the size of the state -- defined as a summary statistic of the entire past -- should either be constant or increase over time.
>
> Yes, we agree that the typical notion of state-size is as you described. However, we want to create a proxy for memory utilization, and therefore we think it's sensible that the state-size can vary non-monotonically depending on the conditions of the input. For example, the primary mechanism of "selective" SSMs that modulate the content of the recurrent state is akin to the "forget gates" in LSTMs, which ESS (a proxy for memory utilization) is able to capture (for selective SSMs like S6 or GLA).

---

> ### Author Response · Authors · 2024-11-28
> **Response to reviewer 77Pw (2/2)**
>
> > 3. While flexibility in variations and aggregation strategies can be appropriate for interpretability tools aimed at informal / qualitative explorations (like attention maps), a paper proposing a new metric should in my view be more precise.
>
> Yes, this is a fair point: the selection of the particular aggregation strategy and/or choosing how to deal with numerical precision via entropy-ESS vs tolerance-ESS is somewhat arbitrary. However, we also note that this is akin to other quantitative metrics like FLOPs: for example, there are multiple ways to determine the number of FLOPs of a single operation (some consider FMA as 2 operations, while others consider it as 1, depending on the analysis). Nonetheless, to be more precise and quantitative in our formulation of the ESS metric, we made the following changes to the paper:
>
> - We reframed the related works from the perspective of “quantifying memory utilization” instead of “quantitatively distinguishing sequence models” to de-emphasize the generally quantitative nature of the ESS metric and instead emphasize its ability to quantify memory utilization in particular
> - We formulate ESS more precisely, by iterating from a formulation general to all linear systems to a formulation specific to multi-layer LIV SISO models. In particular, we start Section 3 by developing the notion of per-sequence index ESS which shows that ESS is a function of the sequence index. We note that this is a general, quantitative formulation that encompasses both linear systems and LIVs that realize an operator $T$. In particular, the per-sequence index ESS (which equals rank($H_i$)) precisely computes a lower bound of the TSS needed for realizing the same operation at that sequence index.
> - Next, for SISO linear systems, we extend the per-sequence index ESS to the channel dimension by discussing the distinction between the ESS computation for SISO and MIMO models in the “Computational complexity of ESS” paragraph. We also elaborate on this in Section D.1.
> - We wrote a new paragraph called “Aggregation of ESS across model and data dimensions” which discusses ESS specifically in the context of multi-layer LIV SISO models since these are the models considered in our paper. This extends the per-channel, per-sequence index ESS discussed in the “Computational complexity of ESS” paragraph to the input (i.e batch) and layer dimensions as well.
> - In this paragraph, we clarify that ESS in multi-layer LIV SISO models can be aggregated in various ways, but we justify the particular aggregations we employ in the paper. In particular, we provide a mathematical formulation of both the average ESS and total ESS to provide quantitative definitions for the metrics. We also clarify that references to ESS in the analysis section are referring to the average ESS.
> - In Section 5.3 where we discuss state-modulation, we define another form of aggregation which we term $(\textrm{total ESS})_i$ which is total ESS that is not marginalized along the sequence dimension. We explicitly state that we are qualitatively analyzing ESS in this case, since this form of aggregation produces a tensor as opposed to a single-number summary of the system.
>
> > 4. I would encourage the authors to consider either formalizing their treatment of ESS, or reframing their work as an exploratory analysis of the inner workings of sequence modeling architectures, without emphasizing the introduction of a quantitative metric.
>
> Via the changes described above, we aimed to be more precise and quantitative in our formulation of the ESS metric by specifying the modes of aggregation employed in this paper and providing the formulas used to compute ESS under the chosen aggregation schemes.
>
> However, we have also revised various parts of the paper to de-emphasize the quantitative nature of the ESS metric a bit since we agree that in the previous iteration of the paper, we did not appropriately acknowledge the certain degree of arbitrariness in choosing things like entropy vs tolerance ESS or the aggregation strategy. We hope that the current set of revisions have clarified the precise, quantitative nature of the average and total ESS metrics, but also made clear that these are not the only ways we can view ESS.
>
> We thank the reviewer again for their response and hope our revisions have resolved the concerns that have been raised.

---

> ### Author Response · Authors · 2024-12-02
> **Follow-up on previous response**
>
> As we approach the end of the discussion period, we would like to follow up on our previous responses to ensure that they have adequately addressed the concerns raised in your previous review. If you have any additional questions, we are happy to provide further clarification.

---

### Official Review · Reviewer_8oZA · 2024-11-08

**Soundness:** 3
**Presentation:** 2
**Contribution:** 2
**Rating:** 6
**Confidence:** 5

**Summary:**

The paper proposes a quantitative measure of memory utilization for modern causal Large Language Models (LLMs). The proposed metric, called Effective State-Size (ESS) is obtained following standard results of realization theory of dynamical systems and can be used to analyze models employing different time mixing layers like Attention, convolutions and more recent (Selective) State Space Models. The paper shows that ESS is informative of the quality of initialization and regularization strategies for LLMs’ training.

**Strengths:**

The paper addresses an important and timely question on how modern LLMs’ sequence mixing layers modulate memory. The proposed metric ESS that is derived for input-invariant dynamical systems can be applied to many different modern architectures and is sound, as it is directly derived from standard results in realization theory of dynamical systems.

**Weaknesses:**

Novelty, the paper argues (Line 108) that one of the main contributions is to prove that the rank of submatrices of the input-output causal operator T determines the minimally realizable state-size of the recurrent operator. There is nothing to prove, this is textbook material reported verbatim from realization theory of linear time invariant dynamical systems. Please make sure to point the reader to proper textbook material for both Theorem 2.1 and Theorem 2.2, see [1, 2, 3].

While the main results on linear input-invariant recurrences are well known, in practice modern LLMs utilize input varying recurrences which are not as straightforward to analyze. This latter case is quickly and vaguely described in a single paragraph (starting in Line 175). This is the key novelty and contribution of the paper (aside from the empirical validation), however the current manuscript does not provide any theoretical guarantees and only describes, non-tight, bounds on the ESS of an input-varying recurrence using its input-invariant counterpart.

A thorough related work section is missing, please consider adding one in the main text. Some references to complement the ones already cited that might be helpful to the readers are provided below (Questions section).

**Questions:**

1. The paragraph starting in Line 175 is unfocused and hard to read, yet it should be the key novel contribution of the paper. How should one define ESS for input-varying recurrences in theory? The ESS needs to depend on the specific input being considered and can, indeed, oscillate depending on the type of the input signals. Have the authors considered using a set of test reference signals or an optimization procedure over the space of sequences of a given length that maximize the required state dimension of their equivalent realization? Are these feasible choices in practice? A deeper discussion about other possible choices that are more theoretically motivated yet not feasible in practice would be beneficial for the reader.
2. Why ESS definitions in Eq. (3) and (4) do not depend on the time index i? Given a sequence and a linear input-invariant system of finite order, one needs to progressively increase the sequence length of the Hankel submatrices until the rank stabilizes to the value of the state dimension. However, Equations (3) and (4) do not scan over different sequence lengths.
3. Why in the empirical validation (section 3) Mamba-1 model has not been validated? I am specifically referring to Mamba-1 and not Mamba-2 since the latter is already closely matched by GLA that is already discussed by the authors.
4. Can the authors explicitly define TSS for each architecture used in the empirical section?
5. Figure 3 is quite strange, GLA and WLA are strictly more expressive than LA yet accuracy of the latter is higher across all evaluations. Is it possible that this is due to some training instability/poor optimization/parametrization? Can the authors contextualize their results in light of the empirical study conducted in [5], specifically in Section 3.4? In particular, highlighting the relationship of fast decaying eigenvalues and normalization at initialization (this is relevant also for your section 4.1).
6. Results in Section 4.2 are not surprising nor specific to the proposed metric and are in general well known, see a similar analysis in [4, 5]. A simple analysis of the eigenvalues of the recurrent updates will lead to the same insights. Is ESS providing further insights?
7. Can the authors further comment on ESS for hybrid models? Why the order of Attention and Recurrent layers matters and how does ESS can guide the user to those insights? Recent Hybrid models that are worth mentioning/commenting are [4, 6, 7].
8. Why in Figure 6 the ESS decreases (almost symmetrically) near the end of the plot? And why do we expect dips in the ESS to account for a state reset? Take for example a LTI system whose state is set to zero at some random point in an interval of length N (where N is the LTI’s state dimension). The ESS value should not change since the state dimension is a structural property of the dynamical system being studied and not of the specific values taken by the state.




Minor:
- Contributions’ section is rather generic; it is worth rewriting it.
- Line 142 typo on the dimension of C_i. Why does the dimension of the A matrix change? Is it to cover the Attention layers whose state needs to increase as the processed sequence length increases?
- Line 235 typo: “as is pertains”.


[1] B. L. HO, R.E. Kalman, “Effective construction of linear state-variable models from input/output functions.”, Regelungstechnik, vol. 14, pp. 545-548, 1966.

[2] Akaike. H. (1974b). Stochastic theory of minimal realization. IEEE Trans. Automatic Control. AC-19:667-674.

[3] L. Ljung, “System Identification: Theory for the User”, Prentice Hall, 1999.

[4] L. Zancato et al., “B'MOJO: Hybrid State Space Realizations of Foundation Models with Eidetic and Fading Memory”. arXiv preprint arXiv:2407.06324 (2024).

[5] A. Orvieto et al., “Resurrecting Recurrent Neural Networks for Long Sequences”, PMLR 202:26670-26698, 2023.

[6] De, Soham, et al. "Griffin: Mixing gated linear recurrences with local attention for efficient language models." arXiv preprint arXiv:2402.19427 (2024).

[7] P. Glorioso, et al. "Zamba: A Compact 7B SSM Hybrid Model." arXiv preprint arXiv:2405.16712 (2024).

---

> ### Author Response · Authors · 2024-11-20
> **Response to reviewer 8oZA**
>
> We thank Reviewer 8oZA for their comprehensive and insightful review! We have incorporated many of your suggestions into the revised draft and address each of your points in turn below:
>
> > 1. Novelty, the paper argues (Line 108) that one of the main contributions is to prove that the rank of submatrices of the input-output causal operator T determines the minimally realizable state-size of the recurrent operator. There is nothing to prove, this is textbook material reported verbatim from realization theory of linear time invariant dynamical systems. Please make sure to point the reader to proper textbook material for both Theorem 2.1 and Theorem 2.2, see [1, 2, 3].
>
> We agree that as originally presented, it appeared that we were arguing that the results we present on linear systems were novel. To address this, we make the following revisions:
> * We explicitly state where the proof of Theorem 3.1 comes from in our presentation of the theorem. Readers can now see where exactly we adapted Dewilde’s proof, as we also point the chapter of the book.
>
> * We make clear that our main contribution isn’t in the proof, but rather in extending the ideas from minimal realization theory in linear systems to LIVs. Namely, we clarify that the novelty of our work lies in showing that ideas from minimal realization theory that have been well established in the context of linear systems can in practice be usefully extended to modern LIV sequence models.
>
> > 2. While the main results on linear input-invariant recurrences are well known, in practice modern LLMs utilize input varying recurrences which are not as straightforward to analyze. This latter case is quickly and vaguely described in a single paragraph (starting in Line 175). This is the key novelty and contribution of the paper (aside from the empirical validation), however the current manuscript does not provide any theoretical guarantees and only describes, non-tight, bounds on the ESS of an input-varying recurrence using its input-invariant counterpart.
>
> We agree that there are no theoretical guarantees other than the non-tight bound that $TSS \geq ESS$ which is why our paper is heavily focused on the empirical validation of ESS as a proxy for memory utilization in LIVs. Our results clearly show that even though we are not presenting a tighter theoretical bound, extending ESS from linear systems to LIVs is empirically justified since we observe strong correlations between ESS and recall performance. This makes it useful in practice given that the space of existing proxies for memory utilization is lacking and ESS serves as a significantly better proxy for memory utilization than TSS which is currently the canonical measure.
>
> > 3. A thorough related work section is missing, please consider adding one in the main text. Some references to complement the ones already cited that might be helpful to the readers are provided below (Questions section).
>
> Agreed, there should be one. We have now added a dedicated section in the main text and have incorporated many of the references you suggested in your comment. Furthermore, we modified our framing of the space of operators to clearly disambiguate between input-invariant linear systems (or just linear systems for short) and systems with input-varying linear operators (which we now abbreviate as LIV). We hope that this helps in disambiguating which results are being used from classic control theory and how we are extending those results to modern sequence models.
>
> > 4. The paragraph starting in Line 175 is unfocused and hard to read, yet it should be the key novel contribution of the paper. How should one define ESS for input-varying recurrences in theory? The ESS needs to depend on the specific input being considered and can, indeed, oscillate depending on the type of the input signals. Have the authors considered using a set of test reference signals or an optimization procedure over the space of sequences of a given length that maximize the required state dimension of their equivalent realization? Are these feasible choices in practice? A deeper discussion about other possible choices that are more theoretically motivated yet not feasible in practice would be beneficial for the reader.
>
> Agreed, we have significantly refactored the theory section in order to emphasize this paragraph and better contextualize it. Instead of taking the route of trying to define ESS for input-varying operators, we directly define ESS as the metric given by the rank($H_i$). Then, we describe theoretical properties of the metric with respect to both classical linear systems and systems with input-varying operators, enabling its interpretation as a proxy for memory utilization. We recognize that our theoretical bound for the input-varying case is not tight, which is why we focused our work heavily on the empirical validation of ESS as a proxy for memory utilization in LIVs. We address the rest of the question in the next point.

---

> ### Author Response · Authors · 2024-11-20
> **Response to reviewer 8oZA continued**
>
> > 5. Why ESS definitions in Eq. (3) and (4) do not depend on the time index i? Given a sequence and a linear input-invariant system of finite order, one needs to progressively increase the sequence length of the Hankel submatrices until the rank stabilizes to the value of the state dimension. However, Equations (3) and (4) do not scan over different sequence lengths.
>
> Yes, ESS does depend on the time index, which is why we used a subscript $i$ when indexing the entropy and tolerance ESS. However, you are correct in that we did not clarify that we marginalize across the entire sequence length when computing ESS. To clarify this, we added a paragraph in Section 3.1 which discusses this among other things that we consider when computing ESS.
>
> Steady-state rank is very interesting, but we did not yet find much use for it in the practical setting. This is because of the following:
> * Most tasks already have a predefined sequence length
> * For input-varying systems, the rank isn’t guaranteed to stabilize.
>
> Nonetheless, we believe it would be interesting to think of the types of input to use to study the steady state rank, but this is something we consider as future work.
>
> > 6. Why in the empirical validation (section 3) Mamba-1 model has not been validated? I am specifically referring to Mamba-1 and not Mamba-2 since the latter is already closely matched by GLA that is already discussed by the authors.
>
> We refrained from empirically validating Mamba-1 since, as noted in prior works like [A], the functional form of S4 (which later led to Mamba-1) is quite complex and contains many components that turn out to be unnecessary for performance. One component in particular is the discretization step, which is lacking from other recurrent models like GLA or LA. Since this, among other aspects of the architecture, introduces unnecessary confounders in the analysis, we found it more appropriate to restrict ourselves to the GLA, LA and WLA functional forms which are varied methodically with respect to how they differ in parameterizing the state transition dynamics through $A$. Furthermore, we do explore S6 (the SSM component of Mamba-1) in the initialization-phase analysis (Section 5.1). Nonetheless, we are happy to run a proof of concept on a subset of the task-model space demonstrating that the correlation between ESS and performance holds in Mamba-1 as well if the reviewer still maintains that this is necessary.
>
> > 7. Can the authors explicitly define TSS for each architecture used in the empirical section?
>
> Yes, we have added this to the appendix which can be found in Section D.2.
>
> > 8. Figure 3 is quite strange, GLA and WLA are strictly more expressive than LA yet accuracy of the latter is higher across all evaluations. Is it possible that this is due to some training instability/poor optimization/parametrization? Can the authors contextualize their results in light of the empirical study conducted in [5], specifically in Section 3.4? In particular, highlighting the relationship of fast decaying eigenvalues and normalization at initialization (this is relevant also for your section 4.1).
>
> As you correctly noted, GLA and WLA are strictly more expressive than LA, and it turns out that this is precisely the reason we observe the phenomenon of state collapse (discussed in Section 4) which is the training instability you are hinting at that lies at the crux of this result. In particular, the gap between performance on LA and GLA/WLA is particularly notable in the set of difficult tasks (where difficulty in the context of MQAR is given by a large number of kv pairs). In this portion of task-model space, we find that GLA and WLA are prone to learning $A$ matrices with values close to 0. This causes signals from the distant past to decay exponentially (resulting in low ESS) because the eigenvalues of the state-transition matrices are parameterized to be less than 1. In language modeling, we think the need for effective state modulation outweighs the requirement for stable long-range training signals, which explains the opposite trend we observe when evaluating bigram recall perplexity.
>
> With respect to the empirical study conducted in [5], we note that unlike their work, our study was conducted on input-varying models and therefore, their findings are not directly applicable to our case. Nonetheless, to contextualize our results with respect to their study, we note that:
> * For Linear Attention, the $A$ matrix is basically at the ring where r_min and r_max = 1 (following notation in [5]).
> * For WLA and GLA the average eigenvalue is 0.96 upon initialization. This is similar to setting r_min =0.9 and r_max = 0.99 in the experiments conducted in [5], which they found to perform well for tasks like LRA.
>
>
> [A] A. Orvieto et al., “Resurrecting Recurrent Neural Networks for Long Sequences”, PMLR 202:26670-26698, 2023.

---

> ### Author Response · Authors · 2024-11-20
> **Response to reviewer 8oZA continued**
>
> > 9. Results in Section 4.2 are not surprising nor specific to the proposed metric and are in general well known, see a similar analysis in [4, 5]. A simple analysis of the eigenvalues of the recurrent updates will lead to the same insights. Is ESS providing further insights?
>
> We agree that the results in Section 4.2 (which has now been moved to the appendix in Section A) in particular are not very surprising in hindsight as having fast decaying eigenvalues results in low ESS, and therefore, we could’ve instead looked at the eigenvalues to have arrived at similar insights. We note, however, that these prior works demonstrate drawbacks of having a parametrization of the eigenvalues with norm less than 1. In this work, we additionally identify that not only is having a norm of less than 1 hurting recall ability, increasing the state-size is also futile if they all decay to values very close to 0, as it effectively does not increase memory utilization, and provide a very simple fix for it.
>
> Moreover, unlike the decay rates which do not capture the complexity/richness in the long range dependencies, ESS provides a metric that can be compared to TSS in order to assess how much of the memory capacity is actually realized. For instance, we see that GLA’s ESS where TSS = 16, ESS is very close to 16, but as state-size increases, the rate at which ESS increases with respect to TSS saturates. Mamba and GLA-S6 have significantly lower ESS at initialization, and all increases in ESS with larger TSS, but at a significantly slower rate (note that decay rates stay relatively constant even with increased state-size). We also see that after increasing arange norm beyond $2^{14}$, the ESS decreases for GLA-S6. These observations prompt further investigation into the parametrization of the different models, and is something that would not be captured if only looking at the eigenvalues and the rate of decay.
>
> Finally, we note that decay rates as measured by the eigenvalues of $A$ fail to directly capture dependencies from the input-to-state and state-to-output transitions given by the $B$ and $C$ matrices, respectively. In contrast, a metric like ESS does capture these aspects of the model.
>
> > 10. Can the authors further comment on ESS for hybrid models? Why the order of Attention and Recurrent layers matters and how does ESS can guide the user to those insights? Recent Hybrid models that are worth mentioning/commenting are [4, 6, 7].
>
> Thank you for bringing the other works on hybrid models to our attention, we have added those citations to the paper.
>
> We note that a full discourse on ESS in hybrids can be found in Section E.4.2, but we provide a distilled version of the interesting findings here in order to answer your questions. In hybrid networks with both 4 and 8 layers, we observe that models with lower ESS in their softmax attention layers tend to perform worse on MQAR compared to models with higher ESS in these layers. As you suggested, this relationship is influenced by both the order and number of attention layers within the network. A particularly notable finding is that placing attention in the first layer often results in significantly lower ESS, which correlates with decreased performance (see Figure 32a). This aligns with prior observations, such as in [B], which use a hybridization policy that avoids assigning attention to the first layer. Here, we see that ESS can guide the user to this insight, as a post-training ESS analysis on the hybrid network provides a concrete explanation as to why this design choice is effective. Concrete justifications as to why certain design choices are made when constructed hybrid networks is something that is currently lacking in works like [C, D]. This is something we hope can be resolved with ESS.
>
> However, making more general claims regarding the impact of hybrid network topology on ESS and performance requires more experiments and ablations, and this is something we intend to explore further in future work. We think that this could lead to the design of hybridization policies that are ESS-informed.
>
> [B] Opher Lieber., et al, "Jamba: A Hybrid Transformer-Mamba Language Model," 2024.
>
> [C] Paolo Glorioso., et al, "Zamba: A Compact 7B SSM Hybrid Model," 2024.
>
> [D] Soham De., et al, "Griffin: Mixing Gated Linear Recurrences with Local Attention for Efficient Language Models," 2024.

---

> ### Author Response · Authors · 2024-11-20
> **Response to reviewer 8oZA continued**
>
> > 11. Why in Figure 6 the ESS decreases (almost symmetrically) near the end of the plot? And why do we expect dips in the ESS to account for a state reset? Take for example a LTI system whose state is set to zero at some random point in an interval of length N (where N is the LTI’s state dimension). The ESS value should not change since the state dimension is a structural property of the dynamical system being studied and not of the specific values taken by the state.
>
> ESS for LTI and LTV is a measure of state-size in the sense that it looks at the number of states required to store from the past to process the future. Therefore, the state-size at the ends are not constrained by the model, but rather by the sequence index. Early sequences have less information to store, and vice versa, when there are not many tokens left to process, it doesn't require as many states.
>
> LTI systems cannot have the state reset to zero, without external intervention, and ESS doesn’t capture such external interventions. To model a state of a LTI system being reset, we can set $A$ and $B$ to 0 at some specific time-step, note that this actually converts the model to an LTV system instead. When $A$ and $B$ are set to 0, the rank of the submatrices will drop because $T_{ij} = C_i A_{i-1} \cdots A_{j+1} B_j$
>
> The dips are not caused by us explicitly resetting, the dips are caused by sending EOS tokens, and the finding is interesting because it shows that some models experience larger dips in state-size compared to others due to this EOS token. Also note that the phenomenon also occurs without the EOS token (using a period instead), but to a lower degree.
>
> > 12. Minor: Rewrite contributions and typos
>
> We have addressed both of these. Please refer to the meta-review addressed to all reviewers for information on how the contributions section was restructured. And yes, this was a typo: thank you for bringing it to our attention.
>
> We thank the reviewer again for their thoughtful review! We hope our responses and revisions have adequately addressed the weaknesses and questions you've posed.

---

> > ### Comment · Reviewer_8oZA · 2024-11-25
> >
> > Thank you for your thorough and clear responses! Most of my concerns have been addressed, and I appreciate the authors' willingness to revise parts of the previous manuscript. I am raising my score as a result.
> >
> >
> > I have two last comments:
> >
> > > 6. Why in the empirical validation (section 3) Mamba-1 model has not been validated? I am specifically referring to Mamba-1 and not Mamba-2 since the latter is already closely matched by GLA that is already discussed by the authors.
> >
> > For the camera ready please include Mamba models, this would certainly help your work to increase its impact and guide many practicioners that tent to trust and use Mamba models more than other variants like GLA.
> >
> > > Finally, we note that decay rates as measured by the eigenvalues of A fail to directly capture dependencies from the input-to-state and state-to-output transitions given by the B and C matrices, respectively. In contrast, a metric like ESS does capture these aspects of the model.
> >
> > I think this is a true statement in general but I don’t expect B and C to play a huge role in practice, since it is not likely that simplifications occur on trained models (i.e. the trained layers are fully observable/controllable) do authors have a better intuition or evidence to support the fact that using ESS rather than a simpler eigenvalues analysis can make a difference? Perhaps a simple baseline to test this could be to measure the complexity of the state by counting the number of eigenvalues grater than a given threshold and confirm whether this performs closely to ESS.

---

> ### Author Response · Authors · 2024-11-28
> **Response to reviewer 8oZA**
>
> We thank reviewer 8oZA for their continually detailed responses as well as the positive evaluation of our work. We respond to the points raised below.
>
> > 1. For the camera ready please include Mamba models, this would certainly help your work to increase its impact and guide many practicioners that tent to trust and use Mamba models more than other variants like GLA.
>
> Thank you for the suggestion. We agree that these results will likely be useful for practitioners, and are happy to include ESS analysis on Mamba models analogous to what is currently in the paper for the camera ready version.
>
> > 2. I think this is a true statement in general but I don’t expect B and C to play a huge role in practice, since it is not likely that simplifications occur on trained models (i.e. the trained layers are fully observable/controllable)
>
> In fact, ESS can be thought of as the minimum rank between the observability (state-output projection) and controllability (input-state projection) matrices (see Equation C.2.4), and because we observe phenomena such as state-collapse during training (Section 4.2), models can in fact simplify during training.
>
> > 3. Do authors have a better intuition or evidence to support the fact that using ESS rather than a simpler eigenvalues analysis can make a difference?
>
> Some intuition on this can be seen through the lens of weight decay on the $B$ and $C$ projection matrices. In particular, note that applying weight decay to $W_b$ and $W_c$ can lower the norm of $B$ and $C$ projections during training, which induces a decrease in ESS. **This is an example of something that cannot be captured by a simple eigenvalue analysis.** This may be one of the reasons why SSMs (like S4, S4D, RTF) typically set weight decay for $B$ and $C$ matrices to 0, while enabling it for other components of the model. However, as mentioned in our earlier response, we do agree that for simpler models (e.g. SISO + diagonal recurrence), there are findings obtained via an analysis of ESS that can also be uncovered with a simpler eigenvalue analysis. As the reviewer pointed out in an earlier response, this insight is leveraged in our mid-training analysis (Section A).
>
> > 4. Perhaps a simple baseline to test this could be to measure the complexity of the state by counting the number of eigenvalues grater than a given threshold and confirm whether this performs closely to ESS.
>
> We tried this baseline for the state-modulation task on Mamba 2.8B and included the results in the supplementary material (Figure 33 is the corresponding ESS plot), a plot of which can also be found [here](https://github.com/iclr2025-ESS/rebuttal-figures/blob/main/state_modulation_A.pdf). We note the similarities and differences below:
> - Both ESS and the number of eigenvalues above a threshold show a noticeable dip around the EOS tokens.
> - ESS highlights the difference in the minimum state size needed to perform the same operation with different separator tokens.
> - In contrast, using only the number of eigenvalues above a threshold to estimate the lower bound for state size is less informative, as the peak state sizes are very similar across both separator tokens.
>
> One final note is that eigenvalue analysis is restricted to SSMs, which means that unlike ESS, it cannot be used to compare between GLA and SA.
>
> Once again, we thank the reviewer for raising important questions and helping us further refine our work.

---

### Official Review · Reviewer_3frp · 2024-11-09

**Soundness:** 3
**Presentation:** 3
**Contribution:** 3
**Rating:** 6
**Confidence:** 3

**Summary:**

This paper introduces the notion of effective state size (ESS), and demonstrate that it correlates with downstream performance on recall-based tasks better than theoretical state size (TSS).  The effective state size (ESS) is the smallest dimension required to represent the causal sequence operation as a linear recurrence.  They test their method for a variety of training runs, and task complexities.

**Strengths:**

* The effective state size metric is intuitive and of conceptual value.
* The experiments show better correlation with accuracy on recall tasks than theoretical state size
* The method may be useful to compare the effective memory capabilities of different architectures, in contrast with their theoretical expressivity.

**Weaknesses:**

* The paper seems to be lacking ablations on network depth, at least in the main body.  Showing how the results change with the number of layers could make the paper stronger.
* There is little discussion about the computational costs of the method.  The state sizes of modern SSMs like Mamba are very large, thus it would help to have some reassurance that the method is scalable to modern architectures.
* The authors briefly mention hybrid architectures in the main body.  However, since Hybrid architectures are becoming more prominent, some further discussion might be warranted.  Since Hybrid architectures have attention layers and skip connections, we might expect that they have full recall capabilities within their context window, which could permit the SSM layers memorize less in their state.

**Questions:**

Line 142: I believe $C \in \mathbb{R}^{d \times n_i}$ (dimensions should be swapped)

How expensive is it to compute the ESS?

“which are comprised of two sequence mixing and two channel mixing layers) detailed above”. Do you do any ablations to consider effect of greater depth?

What is (total ESS) vs ESS Figure 2?

Why such low correlation for WLA method with KV = 2^7 but high at 2^6?

Figure 4b is a bit confusing to read since ESS and Accuracy are on the same axis.  It would also to help explain the experimental process, as it seems there are multiple configurations and you take the mean accuracy across those.

---

> ### Author Response · Authors · 2024-11-20
> **Response to reviewer 3frp**
>
> We thank reviewer 3frp for the constructive feedback. We respond to your points below:
>
> > - There is little discussion about the computational costs of the method. The state sizes of modern SSMs like Mamba are very large, thus it would help to have some reassurance that the method is scalable to modern architectures.
> How expensive is it to compute the ESS?
>
> Thank you for bringing this up. We have added a discussion about the computational cost of the method in the main text (Section 3.1).
>
> > - The paper seems to be lacking ablations on network depth, at least in the main body. Showing how the results change with the number of layers could make the paper stronger.
> “which are comprised of two sequence mixing and two channel mixing layers) detailed above”. Do you do any ablations to consider effect of greater depth?
> The authors briefly mention hybrid architectures in the main body. However, since Hybrid architectures are becoming more prominent, some further discussion might be warranted. Since Hybrid architectures have attention layers and skip connections, we might expect that they have full recall capabilities within their context window, which could permit the SSM layers memorize less in their state.
>
> We present results on network depths greater than two sequence mixing and two channel mixing layers in the hybridization experiments, which can be found in Section E.4.2. However, we note that ablating on network depth is not a central focus in this work since ESS itself is a novel metric and thus warrants detailed exploration in simpler network regimes (i.e. 2 layers). Furthermore, since we explore such a vast task-model space, it is quite costly to extend this analysis across another dimension (in this case network depth). This is in line with prior works like [1] which also restrict the scope of their models to 2 layers.
>
> Nonetheless, some of the findings in the hybrid experiments provide some insight into the robustness of the ESS metric in deeper networks and is something that we are currently exploring as a part of our continued work on the research directions presented in this paper. Namely, in hybrid networks of both 4 and 8 layers, models that realize lower ESS in the softmax attention layers tend to perform worse on MQAR than models with attention layers that realize higher ESS. Furthermore, this realization is sensitive to both the order and number of attention layers in the network. One particularly interesting finding is that placing attention in the first layer of the network tends to result in significantly lower ESS (and also lower performance, see Figure 32a). This is in line with empirical observations in works like [2] which construct hybridization policies that avoid placing attention first. Note that ESS provides concrete intuition as to why this is a good design choice, which is something that is currently lacking in the space of hybrid model design. Making more general claims regarding the impact of hybrid network topology on ESS and performance requires more experiments and ablations, and this is something we intend to explore further in future work.
>
> > - Line 142: I believe … (dimensions should be swapped)
>
> Yes, thank you for pointing that out. It was a typo.
>
> > - What is (total ESS) vs ESS Figure 2?
>
> We have added an explanation in the main text, which can be found in Section 3.1. Total ESS refers to a sum across all channels of the model layer, and average ESS refers to an average across the channels.
>
> > - Why such low correlation for WLA method with KV = 2^7 but high at 2^6?
>
> This is the crux of the state collapse phenomenon we observe and elucidate in Section 4. In particular, a core finding of our work is that for difficult tasks (in the case of MQAR this is captured by a large number of key-value pairs), recurrent models with learnable A matrices like WLA tend to realize significantly lower ESS than a model like LA which has a fixed A. This is because during training, the model tends to learn values in A close to 0, leading to a fast decay across the sequence dimension which results in poor memory utilization.

---

> ### Author Response · Authors · 2024-11-20
> **Response to reviewer 3frp continued**
>
> > - Figure 4b is a bit confusing to read since ESS and Accuracy are on the same axis. It would also to help explain the experimental process, as it seems there are multiple configurations and you take the mean accuracy across those.
>
> Unless specified otherwise, we take the average across sequence length, channels, and layers (we clarify this in Section 3.1). This approach is detailed in the appendix, and we have also added a note in the main text to clarify why we marginalize over these dimensions.
>
> Thanks again for your thoughtful comments! We hope that our discourse regarding deeper, hybrid networks clarifies the questions you posed and that the rest of your concerns have been resolved by our revisions.
>
> [1] Michael Poli, et al, "Mechanistic Design and Scaling of Hybrid Architectures," 2024.
>
> [2] Opher Lieber, undefined., et al, "Jamba: A Hybrid Transformer-Mamba Language Model," 2024.

---

> > ### Comment · Reviewer_3frp · 2024-11-21
> > **Response to authors**
> >
> > I thank the authors for their response.  I have reviewed the other reviewers responses and the authors corresponding comments.  The authors have made steps to address my concerns, thus I have raised my confidence score from 2 to 3.  I appreciate the clarification on total ESS vs. ESS, as well as the plot on hybridization in the appendix, although the plot is a bit difficult to interpret.  I look forward to hearing the dialogue on the other reviews.

---

> > > ### Author Response · Authors · 2024-11-28
> > > **Response to reviewer 3frp**
> > >
> > > We thank the reviewer for their response and positive evaluation of our work! We will try to replot the hybridization results in a more readable manner for the camera ready version of the paper.

---

### Official Review · Reviewer_o977 · 2024-11-10

**Soundness:** 2
**Presentation:** 2
**Contribution:** 3
**Rating:** 5
**Confidence:** 4

**Summary:**

The paper provides detailed analyses of memory utilization for many input-invariant and input-varying linear operators, including attention, convolutions, and recurrences. It introduces the metric called effective state size by leveraging rank or entropy. This metric is used to evaluate and compare the memory utilization of different linear operators from three model phases: initialization, mid-training, and post-training, with a lot of findings. The authors also showed using this to get better performance through better initialization, normalization, and distillation.

**Strengths:**

* The paper proposes considering memory utilization to evaluate the performance of various linear operators. Specifically, it introduces two metrics to measure effective state size, which can serve as a proxy for memory utilization.

* Empirical experiments are conducted across different phases of model training, yielding many interesting findings.

* The analyses of memory utilization may help people better understand the behavior and performance of different models and inspire further improvements.

**Weaknesses:**

* For the recurrence, did the authors investigate the A matrix in the vectorized shape or the scalar value shape? The vectorized A is used in many classical RNN papers as well as in S4, S5, and S6. However, the scalar value A is used in the recent Mamba2 paper and RetNet, which directly builds the connection between attention and state-space models. I think this choice can affect the memory utilization and also efficiency.

* Could the paper provide more discussion and analyses of memory utilization across different sequence lengths? I see the paper extends the analysis to a sequence length of 2048, but what about even longer sequences? Many state-space models highlight their long-context abilities more than attention, and long-context is also a crucial ability required in LLMs.

* While there are many findings (9 in the main paper) related to memory utilization, I wonder which are the most important. Could the authors highlight the most meaningful findings, perhaps in the introduction, to better emphasize the paper's contribution?

* What is the direct conclusion regarding the memory utilization of different linear operators? I didn’t find a direct conclusion or explanation in the findings.

* The terms GLA and WLA are ambiguous. Why do the authors label the input-invariant A matrix as weighted linear attention? Because the softmax attention is also weighted. Also, in line 242, the authors categorize Mamba as a GLA layer. However, Mamba or S6 is not exactly linear attention, although Mamba2 can be.

**Questions:**

Please see the weakness part.

---

> ### Author Response · Authors · 2024-11-20
> **Response to reviewer o977**
>
> We thank reviewer o977 for raising important points regarding our paper. To improve our work, we restructured the paper in order to address higher-level concerns with readability and density. Regarding your point in particular, we clarified our core contributions in the introductory section. We would also like to discuss the questions and weaknesses you raised here:
>
> > - The terms GLA and WLA are ambiguous. Why do the authors label the input-invariant A matrix as weighted linear attention? Because the softmax attention is also weighted.
>
> Gated Linear Attention [1] is a sequence model with a recurrent update similar to Linear Attention. However, it incorporates a gating mechanism on the A matrix, analogous to other deep learning components like Gated Linear Units (GLUs). Weighted Linear Attention (WLA) is derived from Gated Linear Attention; instead of using a gating mechanism, the A matrix remains trainable but is no longer input-dependent. This approach effectively “weights” each state rather than “gating” it, hence the name Weighted Linear Attention. That said, we welcome any suggestions regarding revisions to the naming convention.
>
> > For the recurrence, did the authors investigate the A matrix in the vectorized shape or the scalar value shape? The vectorized A is used in many classical RNN papers as well as in S4, S5, and S6. However, the scalar value A is used in the recent Mamba2 paper and RetNet, which directly builds the connection between attention and state-space models. I think this choice can affect the memory utilization and also efficiency.
>
> We investigated the effective state size for vectorized A matrices, focusing on ESS in models like GLA, WLA, and LA, which have vectorized A matrices of size d/h per channel. Additionally, we provide a derivation in the appendix, illustrating the progression from LA to a state-space model form similar to Mamba 1/2.
>
> > - Also, in line 242, the authors categorize Mamba as a GLA layer. However, Mamba or S6 is not exactly linear attention, although Mamba2 can be.
>
> We agree that GLA is not identical to S6, though it is somewhat similar to Mamba2. Notably, GLA is like Mamba in its use of a vectorized A matrix rather than a scalar one, while its B and C projections more closely resemble those in Mamba2. We chose GLA because our primary goal was to examine ESS variations across a diverse range of model classes. Models like GLA, S6, Mamba2, and Liquid-S4 represent a class where A, B, and C are all input-dependent. In contrast, models like WLA, gated-convolutional models (e.g., RetNet and Hyena) have input-dependent B and C but not A (though A is trainable). Finally, linear attention models offer a case where B and C are input-dependent while A is fixed to an identity matrix. Given that the impact of these configurations on memory utilization is not fully understood, our aim was to compare models from these distinct classes with the fewest changes necessary; thus, we found it the most appropriate to compare using those models selected.
>
> > - Could the paper provide more discussion and analyses of memory utilization across different sequence lengths? I see the paper extends the analysis to a sequence length of 2048, but what about even longer sequences? Many state-space models highlight their long-context abilities more than attention, and long-context is also a crucial ability required in LLMs.
>
> We maintain that our extension to sequence lengths of 2048 is sufficient for capturing the long-context regime as prior related works that work with synthetics such as [2] extend their models to a context length of only 512. Moreover, we note that although state-space models are understood to have strong long-context abilities, [3] contradicts this claim, and our results in Figure 7b also demonstrates similar performance drops of SSMs from sequence length of 512.
>
> > - What is the direct conclusion regarding the memory utilization of different linear operators? I didn’t find a direct conclusion or explanation in the findings.
>
> The direct conclusion is that LA realizes better memory utilization on synthetics than GLA and WLA. We added this to Finding 3.
>
> Thank you again for the detailed evaluation of our work! We hope that our revisions and clarifications address the concerns you have raised.
>
> [1] Songlin Yang, undefined., et al. "Gated Linear Attention Transformers with Hardware-Efficient Training," in ArXiv, vol. abs/2312.06635, 2023.
>
> [2] Simran Arora, undefined., et al, "Zoology: Measuring and Improving Recall in Efficient Language Models," 2023.
>
> [3] Samy Jelassi, undefined., et al, "Repeat After Me: Transformers are Better than State Space Models at Copying," 2024.

---

> ### Comment · Reviewer_o977 · 2024-11-25
> **post-rebuttal**
>
> I thank the authors for their response. Part of my concerns are addressed. However, I disagree with the author’s claim that they "focus on vectorized A matrices, specifically in models like GLA, WLA, and LA, which have vectorized A matrices of size  d/h per channel." In linear attention or general attention mechanisms, scalar A is used per head rather than vectorized A. This can be observed in models like Mamba2 (and RetNet) that they employed scalar A and thus demonstrated that "transformers are SSMs." With vectorized A matrix, parallel scan can only be adopted for fast training rather than the same sequence length parallelism as in attention.
>
> To conclude, my concern is that while attention uses scalar A per head and the paper focuses on vectorized A, it is unclear how the paper categorizes all these methods under the names of (gated or weighted) linear attention. Also, I checked the models that the paper investigated and found that some models adopt scalar A while others use vectorized A. These two kinds of designs bring different efficiency and accuracy differences, but the paper doesn't distinguish them in its statements and experimental comparisons.
>
> Thus, I tend to maintain my score.

---

> ### Author Response · Authors · 2024-11-28
> **Response to reviewer o977 (1/2)**
>
> We thank the reviewer for their detailed response. We respond to their points below.
>
> > 1. However, I disagree with the author’s claim that they "focus on vectorized A matrices, specifically in models like GLA, WLA, and LA, which have vectorized A matrices of size d/h per channel." In linear attention or general attention mechanisms, scalar A is used per head rather than vectorized A. This can be observed in models like Mamba2 (and RetNet) that they employed scalar A and thus demonstrated that "transformers are SSMs."
>
> To clarify, linear attention (LA) has no trainable $A$ matrix, and it can be formulated both in a “vectorized” form (i.e. $A=I$, the identity matrix) and a scalar form (i.e. $A=1$). Please refer to Section D.2 for the formulation of LA with “vectorized” state-transition. Gated linear attention [1] (GLA) is an extension of LA, in which the $A$ matrix is now an input varying diagonal matrix with trainable projections (which is vectorized). Weighted linear attention (WLA) is similarly an extension of LA, with a diagonal $A$ matrix that is trainable but not input-varying. Softmax attention (SA) has a different notion of an $A$ matrix due to the softmax. The complete recurrence for SA can only be formulated with a truncated diagonal $A$ matrix that grows with sequence index (as in Equation C.2.5). To summarize, since the models we examined in this work are either state-space models with “vectorized” state-transitions (GLA and WLA), or models without trainable $A$ matrices (e.g. SA and LA, where the notion of having a vectorized or a scalar state-transition is arbitrary), we maintain the following: **the focus of the experiments presented in our work is on comparing models with “vectorized” $A$ matrices.**
>
> We did not specifically examine models like Mamba2 and RetNet (i.e. models with scalar state-transitions), and simply grouped them into the model class of input-varying $A$ and input-invariant $A$ matrices, respectively. **This is because our axis of comparison is not scalar vs vectorized $A$, but rather input-invariant (WLA) vs input-varying $A$ (GLA)**, which is in line with the classical (and standard) classification of linear systems: linear time-invariant systems (LTI) and linear time-varying systems (LTV).
> Moreover, we chose the input-varying axis of comparison, since intuitively it should have an impact on ESS, since ESS is directly a function of the input. Indeed, we find that this is true when comparing GLA and WLA in the state modulation results presented in Section 5.3. We note that the scalar vs vectorized axis of comparison is certainly interesting to examine, however it is entirely orthogonal to the input-invariant vs input-varying axis of comparison. Therefore, its absence from the paper does not impact the validity of the findings presented.
>
> > 2. It is unclear how the paper categorizes all these methods under the names of (gated or weighted) linear attention.
>
> > 3. I checked the models that the paper investigated and found that some models adopt scalar A while others use vectorized A. These two kinds of designs bring different efficiency and accuracy differences, but the paper doesn't distinguish them in its statements and experimental comparisons.
>
> Continuing from what was mentioned above, our axis of comparison can be characterized as follows:
> - Models with input-varying $A$ matrix such as GLA, Mamba, Mamba2, LiquidS4 (of which GLA and Mamba via S6 are examined in this paper)
> - Models with input-invariant $A$ matrix such as WLA, RetNet, Hyena-S4D, Hyena-RTF (of which WLA is examined in this paper)
> - Models with input-invariant $A$ matrix (where A matrix is fixed to an identity matrix) such as LA (which is examined in this paper)
> - Models with linearly growing state-size such as SA (which is examined in this paper)
>
> While we did not distinguish between scalar and vectorized state-transitions, the categorization itself is consistent and relatively straightforward. Again, the reason we put Mamba and GLA (models with vectorized $A$) and Mamba2 (a model with scalar $A$) in the same category is because we are primarily interested in the effects of input-varying state-transition matrices, and not necessarily on the effects of scalar vs vectorized $A$ matrices.

---

> ### Author Response · Authors · 2024-11-28
> **Response to reviewer o977 (2/2)**
>
> Nonetheless, we do agree with the reviewer’s general sentiment that the scalar vs vectorized axis of comparison is potentially illuminating. In lieu of this, we have just conducted an experiment on a scalar GLA model to show that it, like its vectorized analog presented in the paper, also demonstrates a correlation between ESS and accuracy over the course of training.
>
> In particular, scalar GLA is defined the same as the "vectorized" GLA presented in the paper, except the $A$ matrix is parameterized as
>
>  $A^k_{i-1} = \text{diag}(\text{sigmoid}(W_{A}u_i)^{1/\beta} \mathbf{1})$ where $W_A \in \mathbb{R}^{1 \times d}$ and $\mathbf{1}$ denotes a vector of ones of size $d/h$.
>
> This is distinct from how the "vectorized" GLA [1] explored in the paper is parameterized where
>
> $A^k_{i-1} = \text{diag}(\text{sigmoid}(W_{A_2}^kW_{A_1}u_i)^{1/\beta}) $ where $W_{A_1} \in \mathbb{R}^{16 \times d}$ and $W_{A_2} \in \mathbb{R}^{d / h \times 16}$.
>
> In this experiment, we train a scalar GLA model with a channel dimension of 64 and 8 heads (which means that TSS = 8) on MQAR (with 16 key-value pairs and a sequence length of 128). Analogous to the experiments presented in Section 4.2, we collect the ESS and accuracy over the course of training; a plot of the results can be found [here](https://github.com/iclr2025-ESS/rebuttal-figures/blob/main/acc_and_ess_vs_epoch.png). As shown in the plot, we observe a positive correlation between ESS and accuracy in scalar GLA. This suggests that the scalar vs vectorized nature of $A$ does not impact the utility of the ESS metric as a predictor of performance on MQAR.
>
> Since running the sweep over the entirety of the task-model space explored in the paper takes substantial time given our compute budget, we are happy to run a more extensive set of experiments for the camera ready version and add a section to the appendix discussing the impact of a scalar vs vectorized $A$ on ESS. But again, we note that this axis of comparison is orthogonal to the central narrative of the work and would simply serve as an additional finding that is of potential interest to practitioners.
>
> We thank the reviewer again for the points they have raised and hope our response above has clarified the use of scalar vs vectorized $A$ models in our work.
>
> [1] Songlin Yang., et al. "Gated Linear Attention Transformers with Hardware-Efficient Training," in ArXiv, vol. abs/2312.06635, 2023.

---

> ### Author Response · Authors · 2024-12-02
> **Follow-up on previous response**
>
> As we approach the end of the discussion period, we would like to follow up on our previous responses to ensure that they have adequately addressed the concerns raised in your previous review. If you have any additional questions, we are happy to provide further clarification.

---

### Official Review · Reviewer_6qwH · 2024-11-10

**Soundness:** 3
**Presentation:** 2
**Contribution:** 3
**Rating:** 6
**Confidence:** 3

**Summary:**

The paper frames many sequence modeling architectures as "input-varying linear operators" $u\mapsto T(u)u$, with $u$ being the sequence features. It then focuses on the linear operator $T(u)$ and introduces the "effective state size" (ESS) of this operator as the minimal possible hidden size of the state of an SSM representing $T$. The paper demonstrates that ESS is a relevant measure of the capacity to process memory in the model and compares it favorably to "total state rank" (see discussion below on its definition). The paper also demonstrates how insights about ESS of a model can drive architecture development, initialization and regularization techniques. Finally, the paper shows that the ability to change ESS along the sequence is helpful for real models in bigram recall.

**Strengths:**

- The ESS measure is clever, well justified theoretically through Thm 2.1, 2.2, and, to the best of my knowledge, novel.
   - Even just the rephrasing of other sequence models through the lens of input-dependent SSM parameters can be useful for the community.
- The paper performs _a lot_ of experiments to determine usefulness of ESS.  I especially liked the insight of ESS saturation for the GLA model at initialization.
- The approach is fairly multi-disciplinary within ML, with insights from neural ODEs, SSMs, and control theory.

**Weaknesses:**

# Density
This should have been a journal article. The paper wants to say so much, that the related work section is completely moved into the appendix! The paper attempts to cover extremely wide ground, from theory about ESS to experiments clarifying the notion to at least three fixes to the learning pipeline, and becomes very difficult to follow. In my view, in the current state the paper cannot be accepted even though the contributions are sufficient for ICLR. Hence, I am giving a 5/10, but I will happily adjust the score if the density concerns are addressed. This would require the following:
- Increasing the size of the figures (Fig. 2 and 3 would be impossible to read on paper)
- Moving some of the findings or the related discussions to the appendix. See below for some findings that could be suitable for that, but I will leave it up to authors to decide on the things to move.
- Reintroducing related work, even if just a couple of paragraphs, in the main text

# Other concerns
- We already have a measure of model capabilities that is widely used by the practitioners and that, among other things, correlates with ability to store and recall memories. That is the model size (number of parameters). In my view, the biggest issue other than density is that the paper never discusses the size as a baseline, eg in Figure 2 when comparing TSS to ESS.
- The TSS measure, as discussed in appendix, is not a proper function of the operator T. While model size comes naturally with the architecture and hyperparams, TSS depends on a specific way of _rewriting_ the model's architecture. The paper mostly shows TSS in a negative light as an easy baseline for ESS. The need for introducing TSS is thus not clear to me.
- Finding 3 is too specific. I suspect that the "state collapse", namely the model not utilizing all of its potential, might also be relevant to hyperparameter selection.

## Typos
- Fig 6a y axis: should this be ESS? The text talks about ESS, and I don't think TSS changes over time?
- Fig 5a axis "vs" instead of "kv"

**Questions:**

- Although ESS is introduced as a per-token measure, when presenting most of the results (apart from last section)  ESS is presented as a single value. Is it maximum, average, or some other statistic, and is there intuition for this?
- Finding 6 tells us that GLA and WLA perform worse than LA on assortative recall. This is surprising to me. Shouldn't architectures like Mamba outperform simple linear attention on such things? Maybe it is the specific task, but it was chosen by Poli et al (2024) to be specifically correlated with real-world performance.
  - Related: if it is indeed ESS, how did authors compute it for Mamba 7B? The numbers are of the order of `6e6`, I am not sure we can run SVD on matrices with this number of this size.

---

> ### Author Response · Authors · 2024-11-20
> **Response to reviewer 6qwH**
>
> Thank you very much for the constructive feedback. To improve the paper, we addressed the issue regarding the density (which other reviewers have also criticized directly and indirectly), and also reintroduced important concepts such as TSS into the main text.
>
> Here are the modifications made to the manuscript (as suggested) to address the points you made above:
>
> - We have increased sizes of Figure 2 and 3.
> - We moved the mid-training experiments and discourse on hybridization in post-training to the appendix.
> - Added a section dedicated to related works in the main text.
> - Streamlined the theory section and removed information that is related to but not necessary for formalizing and interpreting ESS
> - Streamlined the introductory section and more clearly emphasized the contributions of our work.
> - Added a section dedicated to discussing TSS in greater detail in the main text.
>
> Next, we would like to discuss other concerns and questions raised:
> > - We already have a measure of model capabilities that is widely used by the practitioners and that, among other things, correlates with ability to store and recall memories. That is the model size (number of parameters). In my view, the biggest issue other than density is that the paper never discusses the size as a baseline, eg in Figure 2 when comparing TSS to ESS.
> > - The paper mostly shows TSS in a negative light as an easy baseline for ESS. The need for introducing TSS is thus not clear to me.
>
> The primary objective of our paper is to explore how different sequence models utilize their memory by using ESS as a framework. While the correlation between model size and recall capability can provide some insights, it does not fully capture the way memory utilization varies across different models, especially given the influence of other factors like MLP size. In contrast, we find that TSS provides a more accurate benchmark, as it represents the theoretical maximum memory utilization (i.e. an upper bound on ESS). Moreover, we wanted to compare metrics that are more closely related to one another (ESS vs TSS), rather than (ESS vs model size). Therefore, we believe it serves as a more suitable baseline for evaluating our proxy. To clarify this further, we have added a dedicated section on explaining and contextualizing TSS.
>
> > - The TSS measure, as discussed in appendix, is not a proper function of the operator T. While model size comes naturally with the architecture and hyperparams, TSS depends on a specific way of rewriting the model's architecture.
>
> As you rightly noted, TSS is not directly a function of the operator. In this paper, however, we use the conventional recurrence formulation for each model. Specifically, for state-space models, TSS corresponds to the state size. For models using softmax attention, where there is no recognized “minimal realization-agnostic” recurrent formulation, we use the trivial realization, with per-channel (i.e. average) TSS being equal to the sequence index. We also added the formulation of each model analyzed to the appendix and hope that the added section on TSS in the main text clarifies this point further.
>
> > Finding 3 is too specific. I suspect that the "state collapse", namely the model not utilizing all of its potential, might also be relevant to hyperparameter selection.
>
> We agree that finding 3 as originally presented was too specific. **Taking advice from reviewer o977, we have added to that finding to clarify more generally that LA better utilizes its memory on synthetics than GLA or WLA.** However, we maintain that noting the specific observation of state collapse occurring for difficult tasks is still pertinent; heavy recall on long sequences is of great interest in language tasks and our finding holds across a large set of tasks and models (i.e. it is not restricted to a small, statistically-irrelevant set of hyperparameters in task-model space).

---

> ### Author Response · Authors · 2024-11-20
> **Response to reviewer 6qwH continued**
>
> > - Although ESS is introduced as a per-token measure, when presenting most of the results (apart from last section) ESS is presented as a single value. Is it maximum, average, or some other statistic, and is there intuition for this?
>
> For the most part, we use the average or sum across the sequence length and channels, which correspond to the average and total ESS respectively. However, we specify if we deviate from this and take the maximum.  To further clarify this, we have also added a section in the “Computing Effective State-Size” section (Section 3.1) in order to explain this distinction. Generally, we find that the observed trends are not particularly sensitive to taking the average vs max and thus tend to present results where we take the average. However, a case where we do note a distinction is in the hybridization experiments (Section E.4.2) where we find that the max ESS more strongly correlates with performance than the average ESS. We can interpret the average ESS as a measure of memory utilization that takes into account all points in the sequence equally, whereas the max ESS is a measure of the highest level of memory utilization observed across the entire sequence length. It is possible that there are other settings where the max and average tell different stories, but for the most part, the analyses conducted in this work were not sensitive to this choice.
>
> > - Finding 6 tells us that GLA and WLA perform worse than LA on assortative recall. This is surprising to me. Shouldn't architectures like Mamba outperform simple linear attention on such things? Maybe it is the specific task, but it was chosen by Poli et al (2024) to be specifically correlated with real-world performance.
>
> We, too, were initially surprised by these results, because as you noted, GLA and WLA are strictly more expressive than LA. However, after examining ESS in both initialization-phase and mid-training experiments, we believe this outcome stems from a trainability issue (which is observed most notably in the state collapse regime). Specifically, signals from the distant past tend to decay exponentially (resulting in low ESS) because the eigenvalues of the state-transition matrices are parameterized to be less than 1. In language modeling, we think the need for effective state modulation outweighs the requirement for stable long-range training signals, which explains the opposite trend we observe when evaluating bigram recall perplexity.
>
> > - Related: if it is indeed ESS, how did authors compute it for Mamba 7B? The numbers are of the order of 6e6, I am not sure we can run SVD on matrices with this number of this size.
>
> Models like Mamba, Softmax Attention, Linear Attention, and S4 use single-input-single-output recurrences, meaning there is no channel mixing within the recurrence. This feature is beneficial for calculating ESS, as it allows us to calculate ESS independently for each channel (and hence system) and then sum them (i.e. total ESS which is described in Section 3.1). In this particular analysis, we also summed ESS across layers, resulting in values on the order of 6e6. Importantly, the actual computation does not require taking SVDs of matrices of this size, as we can decompose the calculation into smaller matrices since we can compute SVDs on the operators for each channel individually instead of computing a single SVD on an operator that comprises all channels.
>
> Again, thank you for your thorough attention to detail when reviewing our work! We hope that our revisions resolve your primary concerns with the readability/density of the paper.

---

> > ### Comment · Reviewer_6qwH · 2024-11-25
> > **Updated rating, remaining issues with related work**
> >
> > The authors did a thorough job in addressing my and other reviewers' concerns and streamlining the exposition of the paper. The new version is indeed more self-contained, and the figures are more readable. As a result, I am updating my score to 6, and recommend accepting the paper. I am also updating my confidence down to account for other reviewers' familiarity with the related work.
> >
> > Regarding the definition of TSS, I agree that it makes intuitive sense for SSMs or RNNs, but the choice to assign it proportional to  the timestamp in transformer models seems not as justified, and provides an overly loose bound. I do suspect that in practice transformers have limited memory capacity, bounded by the dimension of the residual stream and the relevant attention matrices. Hence, ever-growing memory does not seem like a useful baseline. I do acknowledge that, to my knowledge, there is no standard way of measuring the memory in this case beyond hidden dimensions or model size, so lack of a baseline does not necessitate rejection in my opinion.
> >
> > Regarding the related work section, I appreciate that the authors introduce it in the paper, but some important references seem to still be missing. I believe it would be worth discussing classical results on non-input varying linear operators that reviewer 8oZA mentioned there. Furthermore, the references on semiseparable rank that the paper mentions in footnote 3 seem very relevant for the ESS metric, and their relation to current work should be moved to the related work. I found this comment from the reply of the authors to one of the reviewers insightful and relevant here:
> > > Indeed, ESS is connected to 'semiseparable rank,' which we observe can be conceptualized as the max(ESS) across sequence length. However, as presented in previous work [2, 3], semiseparable rank has been applied in a way that is more aligned with the notion of TSS to facilitate efficient algorithm design. This means that prior notions of semiseparable rank do not capture anything beyond model structure (i.e. input data, optimization, etc.). In this study, we provide complete derivations of ESS, clearly distinguish it from TSS, and demonstrate its effectiveness as a proxy for memory utilization in practical settings for modern LIV sequence models. This, in our view, is novel and is the core contribution of our work. Furthermore, applying the ideas of effective rank to produce the entropy-ESS metric has, to the best of our knowledge, never been done before.
> >
> > Finally, the authors cite Kaplan et al 2020 for scaling laws of language models. Kaplan's laws are now obsolete, and practitioners instead follow so-called "Chinchilla scaling laws," given in [1]. It is worth citing those in addition to or instead of Kaplan.
> >
> > [1] Hoffmann, J., Borgeaud, S., Mensch, A., Buchatskaya, E., Cai, T., Rutherford, E., Casas, D.D.L., Hendricks, L.A., Welbl, J., Clark, A. and Hennigan, T., 2022. Training compute-optimal large language models. arXiv preprint arXiv:2203.15556.

---

> > > ### Author Response · Authors · 2024-11-28
> > > **Addressing final concerns from reviewer 6qwH**
> > >
> > > We thank the reviewer for the positive evaluation of our work! We respond to the last couple of points they raised below.
> > >
> > > Regarding the points on TSS, we agree with the observation that attention in practice does not realize an unbounded memory capacity. In particular, we have also observed that the ESS of attention models at initialization grows sublinearly as a function of sequence length, indicating a potentially limited memory capacity. As the reviewer hints at, this suggests that it may be possible to find alternative formulations that take advantage of this observation. However, as mentioned by the reviewer, such recurrent formulations, to the best of our knowledge, do not yet exist; therefore, the reasonable baselines to use are TSS (hidden dimension * sequence length) or model size (both of which we discuss in the paper).
> > >
> > > Regarding the points about the related work, we agree with all them. We have added some discourse on semiseparable rank to the related work, added in the remaining references suggested by reviewer 8oZA and changed the scaling laws reference to the paper suggested by the reviewer.
> > >
> > > We thank the reviewer again for their attention to detail in reviewing our work!

---

### Author Response · Authors · 2024-11-20
**Meta-response to reviewers**

We thank all the reviewers for their thoughtful insights and suggestions.

As our paper aims to cover a broad scope: (1) laying out the theoretical groundwork for interpreting ESS as memory utilization, (2) empirically validating these theoretical interpretations, and (3) garnering insights from applying ESS in a wide range of applications, the majority of the weaknesses and questions were due to the density and presentation of the paper. To address this we made major revisions to the manuscript as follows:
1. We more clearly disambiguate between linear systems and systems with input-varying linear operators, the latter of which we now abbreviate by LIV to conform to classical convention.
2. We move the mid-training results (originally Section 4.2) and hybridization results (originally discussed briefly in Section 4.3), to the appendix in order to better highlight what we think are the central findings of the paper.
3. In Section 1, we modified the contributions section in order to more specifically highlight the core findings of the paper.
4. We have added a concise related works section (Section 2) in order to clarify how our work stands distinct with respect to prior works in the two following ways:
    - We are extending results from classical linear systems theory to modern LIV sequence models
    - We are applying the notions of effective rank to analyze LIVs which has, to the extent of our knowledge, not been done before
5. We have rewritten the theory section (now Section 3) as follows:
    - We clarify that our novel contribution does not lie in the derivation of the theoretical results from minimal realization theory in linear systems, but rather in their extension to LIVs and demonstrating that this extension is empirically justified and can be applied to garner insights that can be used to improve performance in various settings
    - We more thoroughly motivate and define ESS and TSS, discussing their interpretations and how they are distinct from one another
    - We added a section to “Computing Effective State-Size” which clarifies the distinction between average and total ESS
    - We added a section to “Computing Effective State-Size” which discusses the time complexity of computing the ESS metric and ways we can improve it
6. We have reduced the number of references to the appendix to make the work self-contained. As much as possible, we now reserve mentions of the appendix only for ease of reference for the reader.

We hope that our revisions resolve many of the reviewers' concerns and we continue to welcome any additional feedback for further improvement. Thank you again for the constructive feedback.

---

> ### Author Response · Authors · 2024-11-28
> **Meta-response detailing final revisions**
>
> We thank all the reviewers for their repeated engagement during the discussion period. Here, we provide some high level detail regarding the final set of revisions made to the paper.
>
> 1. The related work has been refactored
> - A discussion that contextualizes ESS with respect to concept of semiseparable rank presented in Mamba-2 has been added
> - The second paragraph in related work has been changed to “Quantifying memory utilization in sequence models” in order to emphasize that the quantitative application of ESS lies specifically with respect to it serving as a proxy for memory utilization
> 2. ESS has been formulated more precisely as a quantitative metric
> - We now define ESS for three classes of models in increasing order of specificity: linear systems, SISO linear systems, and multi-layer LIV SISO models (the latter of which are the models explored in the paper)
> - In doing so, we explicitly show how ESS can be viewed as a function of the input, sequence index, channel, and layer in multi-layer LIV SISO models which was previously an implicit assumption
> - We discuss how constructing ESS metrics in multi-layer LIV SISO models can be viewed as selecting a particular aggregation scheme across the input, sequence index, channel, and layer dimensions
> - We motivate average ESS and total ESS as two particular modes of aggregation and justify why these two modes were chosen as the forms of ESS analyzed in the paper; in doing so, we mathematically formulate these metrics to emphasize their quantitative nature
>
> We hope that our final set of revisions resolve the last few points raised by the reviewers, and thank the reviewers again for their insightful comments that have further improved the quality of our work.

---

### Meta-Review · Area_Chair_Yv6x · 2024-12-23

**Metareview:**

## Summary
The paper introduces the effective state size (ESS), a quantitative measure of memory utilization for sequence models. ESS analyzes how models store past information using input-invariant and input-varying linear operators, including attention, convolutions, and recurrences. Unlike prior approaches, ESS offers interpretable metrics that guide initialization, regularization, and model distillation improvements. The authors demonstrate ESS’s utility in enhancing performance and efficiency and analyzing differences in memory usage across architectures.

## Decision

This paper presents interesting ideas and potential contributions; however, in its current form, it is not yet ready for publication at ICLR. As a result, this paper is being rejected due to several significant weaknesses highlighted by the reviewers, including concerns over clarity, presentation, novelty, and depth of analysis. The reviewers gave extensive feedback to the authors to improve their paper; overall, they did a decent job addressing them during the rebuttal period. The average score of the paper was pointing towards borderline. However, the reviewers thought the paper was still not ready yet.  It seems like substantial revisions would be required to make the paper suitable for acceptance at this venue.

**Additional Comments On Reviewer Discussion:**

The main reasons for rejection are summarized below, grouped by similar themes and with attribution to the respective reviewers:



The paper is being rejected due to several significant weaknesses highlighted by the reviewers, including concerns over clarity, presentation, novelty, and depth of analysis. The main reasons for rejection are summarized below, grouped by similar themes and with attribution to the respective reviewers:

Clarity and Presentation Issues:

1.	**Overly Dense Content and Structure**:
Reviewer 6qwH and Reviewer 77Pw noted that the paper attempts to cover too many aspects, making it difficult to follow. Reviewer 6qwH highlighted that related work is relegated to the appendix, and the figures are too small to read. At the same time, Reviewer 77Pw emphasized that key definitions and results require constant referencing of a lengthy appendix. Both reviewers suggested reorganizing the paper to improve readability.

2.	**Ambiguous and Poorly Defined Terminology**:
Reviewer 8oZA and Reviewer 77Pw pointed out that terms such as “state saturation,” “state collapse,” and “total ESS” are not adequately explained in the main text. Reviewer 77Pw also noted that the paper’s presentation of ESS as both an architectural property and an input-dependent metric is unclear.

3.	**Lack of Emphasis on Key Contributions**:
Reviewer o977 mentioned that with nine findings presented, the paper does not highlight which are most significant. This lack of focus diminishes the clarity of its main contributions.

Novelty and Contribution Concerns:

4.	**Limited Originality in Core Results**:
Reviewer 8oZA argued that the theoretical results on input-invariant recurrences rely on well-established concepts in realization theory and do not offer new insights. They also noted that the treatment of input-varying recurrences, which could represent the paper’s key novelty, is limited and lacks theoretical guarantees.

5.	**Insufficient Discussion of Related Work**:
Reviewer 6qwH and Reviewer 8oZA highlighted the lack of a thorough related work section in the main text. Reviewer 8oZA emphasized that the theoretical foundations are not properly contextualized with prior literature.

Experimental and Methodological Weaknesses:

6.	**Lack of Comparison with Baselines**:
Reviewer 6qwH observed that the paper does not adequately compare ESS with model size as a baseline for memory utilization, missing an opportunity to contextualize its contributions.

7.	**Insufficient Analysis of Key Factors:**
Reviewer o977 criticized the lack of analysis on memory utilization for longer sequences, which are critical for state-space and large language models. Similarly, Reviewer 3frp pointed out the absence of ablations on network depth and discussion on computational costs, both of which are critical for assessing scalability.

8.	**Ambiguity in Terminology and Categorization:**
Reviewer o977 mentioned that terms like GLA and WLA are unclear and inconsistently applied, complicating the interpretation of findings.

9.	**Narrow Empirical Findings:**
Reviewer 77Pw and Reviewer 6qwH argued that many findings are narrow in scope and not particularly impactful. Reviewer 77Pw also questioned the broader utility of the empirical results beyond the first finding.

Based on this feedback provided by the reviewers, the paper falls short in its current state in terms of clarity, novelty, and methodological rigor.

---

### Decision · Program_Chairs · 2025-01-22

Reject